

# Bulk-boundary-defect correspondence at disclinations in rotation-symmetric topological insulators and superconductors

**Max Geier[1,2⋆], Ion Cosma Fulga[3] and Alexander Lau[3,4,5]**

**1** Dahlem Center for Complex Quantum Systems and Physics Department,
Freie Universität Berlin, Arnimallee 14, 14195 Berlin, Germany
**2** Center for Quantum Devices, Niels Bohr Institute, University of Copenhagen,
DK-2100 Copenhagen, Denmark
**3** Institute for Theoretical Solid State Physics,
IFW Dresden, 01171 Dresden, Germany
**4** Kavli Institute of Nanoscience, Delft University of Technology,
P.O. Box 4056, 2600 GA Delft, Netherlands
**5** International Research Centre MagTop, Institute of Physics, Polish Academy of Sciences,
Aleja Lotnikòw 32/46, PL-02668 Warsaw, Poland

⋆ geier@nbi.ku.dk

## Abstract

We study a link between the ground-state topology and the topology of the lattice via the presence of anomalous states at disclinations – topological lattice defects that violate a rotation symmetry only locally. We first show the existence of anomalous disclination states, such as Majorana zero-modes or helical electronic states, in second-order topological phases by means of Volterra processes. Using the framework of topological crystals to construct $d$-dimensional crystalline topological phases with rotation and translation symmetry, we then identify all contributions to $(d-2)$-dimensional anomalous disclination states from weak and first-order topological phases. We perform this procedure for all Cartan symmetry classes of topological insulators and superconductors in two and three dimensions and determine whether the correspondence between bulk topology, boundary signatures, and disclination anomaly is unique.

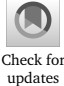

# 1   Introduction

Topological crystalline insulators and superconductors have an excitation gap in the bulk and feature protected gapless or zero-energy modes on their boundaries [1–3]. These boundary modes are anomalous in the sense that they can only be realized in the presence of a topological bulk. Crystalline symmetries, such as rotation or inversion symmetry, may protect higher-order topological phases for which anomalous states are located at corners or hinges of the crystal [4–27]. In particular, a $d$-dimensional topological crystalline phase of order $n$ hosts $(d-n-1)$-dimensional anomalous states at hinges or corners of the corresponding dimension. This correspondence between bulk topology and boundary anomaly is a fundamental aspect of topological insulators and superconductors [12, 25, 28–33].

Topological lattice defects violate a crystalline symmetry locally while the rest of the lattice remains locally indistinguishable from a defect-free lattice. They can be constructed by cutting and gluing symmetry-related sections of the lattice by means of a Volterra process [19, 34–36]. Topological lattice defects are characterized by their holonomy, which is defined as the action on a local coordinate system transported around the defect. Common examples are dislocations and disclinations. The latter violate rotation symmetry locally and carry a rotation holonomy. The association to a holonomy is the property that distinguishes topological lattice defects from other lattice defects. For example, atomic defects such as vacancies, substitutions, or atoms at interstitial positions are not associated to a holonomy, and therefore are not considered topological. For grain boundaries separating regions of different lattice orientations, it has been suggested that they can be described as arrays of dislocations [37–39] or

disclinations [40–45].

Previous works have shown that dislocations carry anomalous states in weak topological phases [31, 46–52]. The label weak indicates that the topological phase is protected by translation symmetry. The existence of anomalous states at disclinations in the absence of weak topological phases has been shown in Refs. [14, 19, 36, 53–55]. Moreover, crystalline topological phases generally have a topological response associated with topological lattice defects [36]. A possible link between second-order topology and anomalous states at disclinations has been put forward in Refs. [14, 19]. Furthermore, a correspondence between a fractional corner charge in two-dimensional topological crystalline insulators [5,6,18,56] and a fractional disclination charge has been shown in Refs. [16,20]. A correspondence between a topological phase realized on a lattice with dislocations and a topological phase realized on a defect-free lattice on a manifold with a larger genus has been suggested in Refs. [57–59]. More generally, symmetry-flux defects have been shown to characterize symmetry-protected phases of matter [60]. In strongly interacting spin-boson models, it has been suggested that the interplay between spontaneous symmetry breaking and symmetry protected topology leads to the appearence of anomalous defect modes at solitons [61,62].

In this study, we establish a precise relation between second-order topological phases protected by rotation symmetry and anomalous states at disclinations. By using both heuristic arguments and the framework of topological crystals [22], we work out for all Cartan classes of spinful fermionic systems the exact conditions under which this *bulk-boundary-defect correspondence* holds. In the cases where it breaks down, the anomaly at the disclination depends on the microscopic properties of the system. Under certain conditions, this obstruction manifests as a domain wall that is connected to the disclination.

Our analysis covers both topological phases defined in the long-wavelength limit where the lattice may be neglected, and topological phases enabled by the presence of the discrete translation symmetry of a lattice. The former shows that the bulk-boundary-defect correspondence does not require an underlying lattice. The latter identifies the contribution of weak topological phases to the anomaly at a dislocation or disclination with non-trivial translation holonomy.

We further go beyond the known result that first-order topological phases can host anomalous states at defects carrying a magnetic flux quantum, for example vortices in a two-dimensional $p$-wave superconductor [63]. In particular, building on the results of Ref. [47], we show that first-order topological phases in symmetry classes that allow for $(d-2)$-dimensional anomalous states host such anomalous states at defects that carry a geometric $\pi$ flux, i.e., the wavefunction of a particle transported around the defect acquires a phase shift of $\pi$. These defects can be viewed as an abstract generalization of vortices in superconductors to other Cartan symmetry classes of quadratic fermionic Hamiltonians.

By collecting all of these contributions, our work contains a unified description of defects in systems described by quadratic fermionic Hamiltonians with translation and rotation symmetries and their anomalous states associated with the topology of the bulk. We provide comprehensive formulas for the defect anomalies in terms of the topological properties of the defect and the topological invariants of the bulk. These formulas apply to disclinations, dislocations, and vortices, as well as combinations and collections thereof. We discuss these defects and their anomalies for all Cartan symmetry classes of quadratic fermionic Hamiltonians in two and three dimensions, which generalizes the previous results for individual symmetry classes or lattice defects of Refs. [14, 31, 36, 46–55].

Our article is organized as follows. Section 2 reviews the construction and the holonomy classification of disclinations in lattice models of fermionic systems. In Sec. 3, we begin by

giving a brief overview of second-order topological phases and their bulk-boundary correspondence. We determine the existence of anomalous states at disclinations for models defined in the long-wavelength limit. In the following Sec. 4, we construct real-space representations of second-order and weak topological phases in the presence of discrete translation symmetry to deduce the existence of anomalous disclination states. This section may be skipped at first reading. In Sec. 5, we cumulate our results to show that each topological property of a disclination, i.e., its translation and rotation holonomies as well as the presence of quantized vortices, is linked to a unique bulk topological invariant determining the existence of anomalous states at the defect. For all symmetry classes we detail whether the bulk-disclination correspondence holds and whether there exist weak and strong first- and second-order topological phases that may contribute $d - 2$ dimensional anomalous states bound to a disclination. In Sec. 6, we apply our construction to a few simple, but physically relevant examples in superconductors in Cartan classes D, DIII, and in insulators in Cartan class AII. We summarize our results and conclude in Sec. 7.

## 2 Disclinations

We begin by providing a brief review of disclinations in two and three dimensions, which are at the core of this work. We recall their definition, their construction from a defect-free lattice through a Volterra process, and their holonomy classification. Finally, we draw a connection from the abstract lattice with disclination to its decoration with physical degrees of freedom. In particular, we show how to construct the hopping terms of a given Hamiltonian around the disclination and identify the arising symmetry constraints on the Hamiltonian terms. In subsequent sections, we will use the latter to show the existence or absence of anomalous disclination states in various symmetry classes.

### 2.1 Topological lattice defects

A lattice is abstractly defined by its space group $G_{\text{lat}}$ that contains all crystalline symmetry elements, e.g., translations, rotations, and inversion symmetry. More precisely, we define the lattice as the charge density $\rho(\mathbf{r})$ of the crystalline system. The charge density breaks the Galilean symmetry group $G_{\text{Galilean}} = O(d) \ltimes \mathbb{R}^d$ of free, $d$-dimensional space into a discrete subgroup $G_{\text{lat}} \subset G_{\text{Galilean}}$.[1] As the lattice is symmetric only with respect to discrete translations $\mathbf{T}$, the charge density in all space can be constructed from the charge density in a minimal volume by applying the translation operators. This minimal volume is called the primitive unit cell [64].

A topological lattice defect breaks an element of the space group of the lattice $G_{\text{lat}}$ locally such that the lattice , i.e. the charge density, remains locally indistinguishable from a defect-free arrangement everywhere else[2]. These defects are topological in the sense that local rearrangements of the lattice can only move, but not remove the lattice defect. This implies that there exists a topological quantity, defined on a closed loop or surface enclosing the defect, that quantifies the lattice defect. This topological quantity can be expressed in terms of the

---

[1] Later in the manuscript, we discuss systems with both magnetic and non-magnetic space groups $G \subset G_{\text{Galilean}} \times \mathbb{Z}_2^{\mathcal{T}}$, where $\mathbb{Z}_2^{\mathcal{T}}$ is the group generated by time-reversal symmetry $\mathcal{T}$ (see also our discussion in section 2.5). As the charge density is a real scalar, it preserves time-reversal symmetry. Therefore, the space group of the lattice $G_{\text{lat}} \subset G_{\text{Galilean}}$ can be constructed from general space groups by selecting the unitary elements in the quotient $G \times \mathbb{Z}_2^{\mathcal{T}} / \mathbb{Z}_2^{\mathcal{T}}$. For non-magnetic space groups $G$, we have simply $G_{\text{lat}} = G$.

[2] A finite volume is said to be locally indistinguishable from a defect-free lattice if the charge density in the volume can be constructed by applying the space group symmetry elements to any primitive unit cell within the volume.

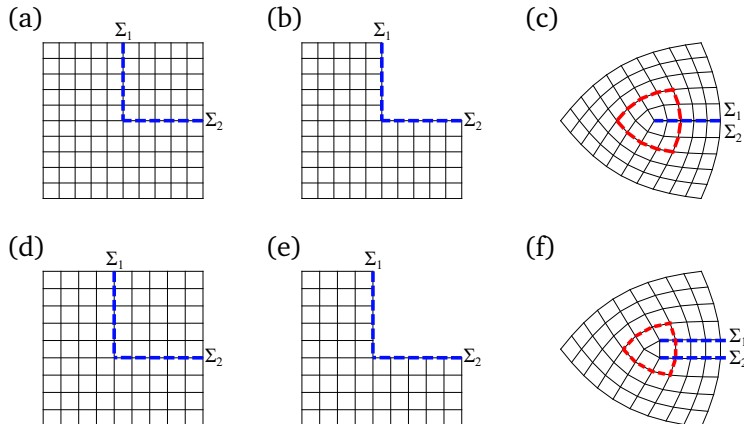

Figure 1: Volterra processes to construct two different $\pi/2$ disclinations in a $C_4$-symmetric lattice. The solid lines outline the boundary of the primitive unit cell. (a)-(c) Type-0 disclination centered at a 3-vertex. (d)-(f) Type-1 disclination centered at a triangular cell. The red dashed lines in (c) and (f) indicate paths encircling the respective disclination.

holonomy associated with the defect. For lattice defects with co-dimension 2, the holonomy is defined as the action on a local coordinate system upon parallel transport along a closed loop around the defect [54]. Common examples of topological lattice defects are dislocations and disclinations, which locally violate translation symmetry and rotation symmetry, respectively. In the following, we focus on disclinations and show how they are constructed using a Volterra process [34,35] for the example of a lattice with $C_4$ symmetry.

## 2.2 Volterra process

To construct a disclination as a topological lattice defect, we require that one chooses a symmetric primitive unit cell that respects the rotation symmetry. The local charge density in the primitive unit cell should respect the rotation symmetry. We consider an example of a two-dimensional lattice with fourfold rotation symmetry, where the symmetric primitive unit cell is a square.

We first cut the crystal along two lines, $\Sigma_1$ and $\Sigma_2$, intersecting in a point $p$ and related by a rotation about an angle $\Omega = \pi/2$ consistent with the lattice symmetry [see Figs. 1(a), and (d)]. The cuts should be performed at the boundary of our choice of symmetric primitive unit cell. We then remove the enclosed segment [see Figs. 1(b), and (e)], deform the crystal such that the lines $\Sigma_1$ and $\Sigma_2$ come together, and finally glue the lattice back together along the cut [see Figs. 1(c) and (f)]. By cutting the sample only along the boundary of symmetric primitive unit cells, we ensure that upon gluing the lattice back together, the charge density with disclination is locally indistinguishable from the defect-free configuration. At the same time, this construction provides a consistent definition of a unit cell in the presence of a topological lattice defect.

This procedure may be used to construct distinct types of $\pi/2$ disclinations depending on the number of additional lattice translations along the direction of the cut: in Fig. 1(c), no extra translation is applied, thereby forming a disclination centered at a vertex with three connections. In Fig. 1(f), one additional translation leads to a disclination centered at a triangular cell. The presence of the disclination strains the lattice close to the defect.

We point out that instead of cutting and removing a segment, one can also cut the crystal along a single line and insert a segment with boundaries related by a $\Omega$ rotation. This process constructs a disclination with a negative Frank angle $-\Omega$ (see below). Furthermore,

disclinations can be constructed as pairs with opposite Frank angle, as we exemplify in App. B.2. In these so-called disclination dipoles, only loops that encircle a single disclination carry a rotation holonomy.

## 2.3 Holonomy of disclinations in two dimensions

Disclinations are classified by their holonomy, which is defined as the amount of excess translation and rotation accumulated by parallel transporting a coordinate system on a closed path around the disclination [35,50,53,54,65,66]. Holonomic quantities are path-independent as long as the starting point is fixed and the path encircles the disclination only once. By considering equivalence classes of holonomies that can be reached by a change of starting point, the holonomic quantities become also independent of the starting point (see Appendix A for details). The rotation holonomy $\Omega$ is called the *Frank angle* and is, by construction, identical to the angle $\Omega$ in the Volterra process defined above. The equivalence classes $\mathrm{Hol}(\Omega)$ of $\Omega$ disclinations in $2\pi/\Omega$-fold rotation symmetric lattices are [53,66]

$$
\begin{aligned}
\mathrm{Hol}(\pi) &= \mathbb{Z}_2 \oplus \mathbb{Z}_2, \\
\mathrm{Hol}(2\pi/3) &= \mathbb{Z}_3, \\
\mathrm{Hol}(\pi/2) &= \mathbb{Z}_2, \\
\mathrm{Hol}(\pi/3) &= 0.
\end{aligned}
\tag{1}
$$

These equivalence classes distinguish disclinations by their translation holonomy. The inequivalent translation holonomies are accompanied by an inequivalent connectivity of unit cells at the disclination center, as illustrated in Fig. 1 (c) and (f) for $\Omega = \pi/2$ and Fig. 2 for $\Omega = \pi/3$, $\pi$ and $2\pi/3$.

For twofold symmetric lattices, there are four types of $\pi$ disclinations. They are distinguished by the parity of the number of translations along the $x$ and $y$ direction of the crystal (see Fig. 2). Threefold rotation-symmetric lattices may host three distinct types of $2\pi/3$ disclinations distinguished by their rotation holonomy modulo three, which is illustrated in Fig. 2. For fourfold symmetric lattices, there are two types of $\pi/2$ disclinations corresponding to whether an even (type 0) or odd (type 1) number of translations by primitive Bravais lattice vectors is required to move around the disclination This is illustrated in Figs. 1(c) and (f), respectively. Finally, sixfold symmetric lattices allow for only a single type of $\frac{\pi}{3}$ disclination (see Fig. 2).

Note that a local rearrangement of the lattice allows to split a topological lattice defect into its elemental components, and vice versa. For example, a $\pi/2$ disclination of type 1 can be split into a $\pi/2$ disclination of type 0 and a dislocation with odd translation holonomy.

## 2.4 Screw disclinations in three dimensions

In three dimensions, a disclination can also carry a translation holonomy $T_z$ in the direction of the rotation axis. These disclinations can be constructed through a Volterra process by translating one of the cut surfaces, $\Sigma_1$ or $\Sigma_2$, along the $z$ direction before they are reconnected. A disclination that carries such a translation holonomy is called a screw disclination.

## 2.5 Decorating a lattice with disclination

In section 2.1, we defined the lattice through the local charge density. The local charge density specifies where the orbitals are located, but not which local degrees of freedom are provided by the orbitals. The physical properties of the local degrees of freedom on the lattice can be described by a tight-binding Hamiltonian $H$. Below, we discuss how to construct a Hamiltonian



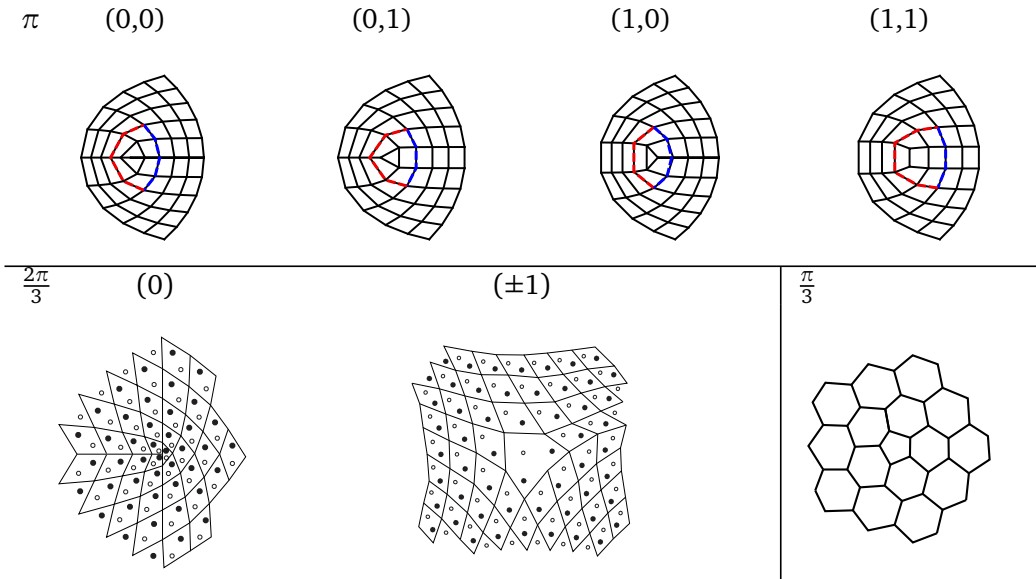

Figure 2: Disclinations in twofold-, threefold-, and sixfold-symmetric lattices: $\pi$ disclinations in twofold-symmetric lattices come in four types ($T_x \bmod 2, T_y \bmod 2$) distinguished by the parity of their translation holonomy $T_i$ ($i = x, y$). The translation holonomy is indicated by the dashed lines, where red (blue) lines are translations in the $x$ ($y$) direction of the local coordinate system. The unit cells of threefold symmetric lattice are parallelograms composed of two equilateral triangles (see Fig. 5). For sixfold symmetric lattices, we choose a hexagonal primitive unit cell. For a threefold-symmetric lattice, there are three types of $2\pi/3$-disclinations. Different threefold rotation centers within the unit cell are denoted by filled and hollow dots. The two types ($\pm 1$) of $2\pi/3$-disclinations differ by exchanging the filled-dot rotation centers with the hollow-dot rotation centers. Finally, in sixfold-symmetric lattices there is only a single type of $\frac{\pi}{3}$ disclinations, which is centered at a five-sided cell.

$H$ on a lattice with disclination from a defect-free Hamiltonian, such that $H$ is locally indistinguishable from the defect-free system everywhere except at the disclination. As a result, we will see that in some symmetry classes, this construction would break some symmetries along the cut line where the system is glued together to form a disclination. We will argue that for these symmetries classes, the cut line can be regarded as a domain wall separating regions distinguishable by a local order parameter.

For our construction, we consider the lattice containing the disclination as the result of a Volterra process [see again Fig. 1(b) and (c)] with the real space positions $r$ and $\mathcal{R}_\Omega r$ along the cut lines identified, where $\mathcal{R}_\Omega$ denotes a rotation by the angle $\Omega$. In this picture, both the coordinate system and the local degrees of freedom of two adjacent unit cells across the cut lines are rotated with respect to each other by the Frank angle $\Omega$. A particle hopping across this branch cut has to respect this local change of basis. Hence, its wavefunction $|\psi(r - \delta r)\rangle$ has to transform to $U(\mathcal{R}_\Omega)|\psi(\mathcal{R}_\Omega r + \delta r)\rangle$ when moving from $r - \delta r$ to $\mathcal{R}_\Omega r + \delta r$ across the branch cut. Here, $U(\mathcal{R}_\Omega)$ is the representation of rotation symmetry acting on the local degrees of freedom within a unit cell and $\delta r$ is a finite but small integer mulitple of the lattice vectors. As mentioned above, the points $r$ and $\mathcal{R}_\Omega r$ are identified. This implies that all hopping terms crossing the branch cut have to incorporate the basis transformation. Requiring that the hopping across the branch cut be indistinguishable from the corresponding hopping in the bulk, the hopping terms $H^{\mathrm{cut}}_{r, r + a_n}$, in mathematically positive direction with respect to

the Frank angle $\Omega$ of the disclination, can be expressed as

$$H^{\text{cut}}_{\boldsymbol{r},\boldsymbol{r}+\boldsymbol{a}_n} = U(\mathcal{R}_\Omega)H_{\boldsymbol{r}_i,\boldsymbol{r}_i+\boldsymbol{a}_n}, \tag{2}$$

where $H_{\boldsymbol{r}_i,\boldsymbol{r}_i+\boldsymbol{a}_n}$ is a corresponding hopping element between unit cells at $\boldsymbol{r}_i$ and $\boldsymbol{r}_i + \boldsymbol{a}_n$ in the translation symmetric bulk. [3]

The hopping terms across the branch cut in Eq. (2) have to respect all *internal symmetries* $g \in G_{\text{int}}$ of the crystal, where $g$ denotes the symmetry element and $G_{\text{int}}$ is the group of internal symmetries. Internal symmetries are global onsite symmetries that act trivially on the real space coordinates. Examples are time-reversal symmetry $\mathcal{T}$, particle-hole antisymmetry $\mathcal{P}$, chiral antisymmetry $\mathcal{C} = \mathcal{PT}$, and $SU(2)$ spin rotation symmetry $\mathcal{S}$. The onsite action of each (crystalline or internal) symmetry element $g \in G \times G_{\text{int}}$ on the Hamiltonian $H$ is expressed by its representation $U(g)$. In the following, we present and discuss the arising constraints on the Hamiltonian terms due to the presence of both unitary and antiunitary symmetries/antisymmetries.

First, for general hopping elements $H_{\boldsymbol{r}_i,\boldsymbol{r}_i+\boldsymbol{a}_n}$, the internal unitary symmetries/antisymmetries $\mathcal{U} = \mathcal{S}, \mathcal{C}$ require

$$U(\mathcal{U})H^{\text{cut}}_{\boldsymbol{r}_i,\boldsymbol{r}_i+\boldsymbol{a}_n}U(\mathcal{U})^\dagger = \pm H^{\text{cut}}_{\boldsymbol{r}_i,\boldsymbol{r}_i+\boldsymbol{a}_n}. \tag{3}$$

This condition can only be fulfilled if the representation of the unitary rotation symmetry commutes with all internal symmetries/antisymmetries of the crystal. If rotation and internal symmetries do not commute, any finite hopping across the branch cut that respects the internal symmetries/antisymmetries necessarily breaks rotation symmetry locally along the branch cut. In this case, the algebraic relations between the symmetry operators obstruct the choice of a hopping across the branch cut that is locally indistinguishable from the bulk hopping. As such, the branch cut can be regarded as a physical domain wall separating regions that are distinguishable by a local order parameter that relates to the local arrangement of the orbitals in the unit cell (see Appendix B.1 for an in-depth discussion).

We point out that this domain wall may become locally unobservable if the sample as a whole breaks at least one of the internal symmetries, or if the translation holonomy of the disclination involves a translation holonomy by a fractional lattice vector, see Appendix B.1. Throughout this paper we aim to make general statements for the topological properties in each symmetry class, and omit model specific details. Therefore, we assume throughout the paper that (i) the sample as a whole obeys all internal symmetries and (ii) that the translation holonomy of the disclination is restricted to integer multiples of the lattice vectors. The latter condition is fulfilled by the topological lattice defects as constructed in this section, because this construction provides a global definition of the unit cell even in the presence of a disclination.

Second, the internal antiunitary symmetries/antisymmetries $\mathcal{A} = \mathcal{T}, \mathcal{P}$ give the constraint

$$U(\mathcal{A})\left(H^{\text{cut}}_{\boldsymbol{r}_i,\boldsymbol{r}_i+\boldsymbol{a}_n}\right)^* U(\mathcal{A})^\dagger = \pm H^{\text{cut}}_{\boldsymbol{r}_i,\boldsymbol{r}_i+\boldsymbol{a}_n}. \tag{4}$$

Note that there is generally a $U(1)$ phase ambiguity in choosing the representation of rotation symmetry: the Hamiltonian is symmetric under $e^{i\phi}U(\mathcal{R}_\Omega)$ for all phases $\phi$. The value of $\phi$ enters in the commutation relations of $e^{i\phi}U(\mathcal{R}_\Omega)$ with antiunitary time-reversal symmetry $\mathcal{T}$ and particle-hole antisymmetry $\mathcal{P}$ as $U(\mathcal{R}_\Omega)U(\mathcal{A}) = e^{-2i\phi}U(\mathcal{A})U(\mathcal{R}_\Omega)^*$. The condition in Eq. (4) therefore fixes the phase factor $e^{i\phi}$ up to a sign, but does not otherwise obstruct the formation of disclinations which are indistinguishable away from their core.

If a system is symmetric under the combined action of rotation and time-reversal symmetry $\mathcal{RT}$, but neither under the action of rotation nor time-reversal symmetry separately, the system is said to have *magnetic* rotation symmetry. When constructing a $2\pi/n$ disclination in a lattice

---

[3]For concreteness, the element $H_{\boldsymbol{r}_i,\boldsymbol{r}_i+\boldsymbol{a}_n}$ can be taken from the translation symmetric system without disclination and without boundary whose lattice positions are denoted by $\boldsymbol{r}_i$.

with an $n$-fold magnetic rotation axis using a Volterra process, we have to connect two parts of the lattice that are mapped onto each other under magnetic rotation symmetry. Since the disclination cannot involve the time-reversal operation, any finite hopping across the branch cut necessarily breaks magnetic rotation symmetry. Thus, the branch cut forms a domain wall separating regions distinguishable by a local order parameter that is odd under time-reversal symmetry, see Appendix B.3 for an explicit example.

In summary, a necessary condition for the application of a *bulk-equivalent* hopping across the branch cut [see Eq. (2)] is a unitary rotation symmetry that commutes with all unitary internal symmetries and antisymmetries of the system. In the absence of additional crystalline symmetries, this condition is also sufficient. If this condition is violated, the Volterra process leads to a domain wall emanating from the disclination. This insight will be crucial in subsequent sections to show for which symmetry classes anomalous disclination states exist.

### 2.6 Rotation holonomy for spinful fermions

At the end of this section, we want to briefly remark on a peculiarity regarding systems with half-integer spins. Rotating a particle with half-integer spin by $2\pi$ shifts the phase of its wavefunction by $\pi$. As a consequence, it seems as if the rotation holonomy of disclinations for particles with half-integer spin should be defined modulo $4\pi$ [54]. However, when transporting a half-integer spinful particle around a $2\pi$ disclination [4], there are two effects contributing a $\pi$ phase to its wavefunction: (i) the rotation of the real space coordinate system and (ii) the basis rotation $U(\mathcal{R}_{2\pi}) = -1$ of the local degrees of freedom upon applying the gluing prescription Eq. (2) when forming the disclination. [5] The total phase acquired is thus $\pi + \pi = 2\pi$.

The geometric phase shift $\alpha$ obtained upon parallel transport of a particle along a closed loop can be quantized to multiples of $\pi$ both by time-reversal, as well as by particle-hole symmetry. In the presence of time-reversal symmetry, the magnetic flux enclosed by a closed loop is quantized to multiples of the magnetic flux quantum $\phi_0 = hc/2e$. Parallel transporting a charged particle around a magnetic flux quantum leads to a $\pi$ phase shift of its wavefunction. Particle-hole symmetry also quantizes this phase shift to integer multiples of $\pi$, as can be best seen in superconductors, where the integer corresponds to the number of superconducting vortices which are encircled by the particle's path. Throughout the paper we say a defect carries a geometric $\pi$-flux if the geometric phase shift $\alpha$ is equal to $\pi$ mod $2\pi$.

As by the above argument parallel transporting a half-integer spin particle around a $2\pi$ disclination does not cause a $\pi$ phase shift, we distinguish between disclinations on the one hand and point defects binding geometric $\pi$-flux quanta on the other hand. Throughout the paper, we therefore assume that a disclination does not bind a geometric $\pi$-flux unless otherwise stated.

## 3 Strong second-order topology and disclinations

In this section, we set out to investigate anomalous states bound to disclinations in second-order topological phases with rotational symmetry. More specifically, we focus on *strong* topological phases where the presence of the underlying lattice can be neglected. The effect of translational symmetries will be considered in a later section (see Sec. 4).

We first review properties of rotation-symmetry protected topological phases, define the notion of topological charge for anomalous bound states, and present constraints on these

---

[4] A $2\pi$ disclination may also be formed through a Volterra process, for example by inserting a segment.

[5] By insisting on the gluing prescription Eq. (2) also for $2\pi$ disclinations, we ensure that they can be combined from disclinations with smaller Frank angle and that disclinations are clearly distinguished from geometric $\pi$ fluxes.

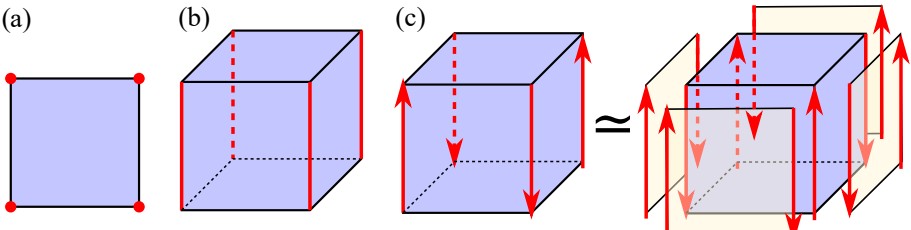

Figure 3: Examples of strong topological phases protected by a fourfold rotation symmetry: (a) with Majorana corner states in two dimensions, and (b) with helical hinge modes in three dimensions. In (c), a three-dimensional insulator with chiral hinge modes consistent with a magnetic rotation symmetry $C_4\mathcal{T}$ is depicted. A symmetry-allowed decoration as indicated changes the propagation direction of the chiral modes after hybridization.

charges imposed by rotational symmetry. We then consider these phases in the presence of an isolated disclination breaking the protecting rotational symmetry locally. Using Volterra processes as introduced in Sec 2.2, we show under which conditions strong second-order topological phases host protected anomalous states at disclinations. Finally, we provide a brief summary of the main results of this section.

## 3.1 Strong rotation-symmetry protected second-order topological phases

A *strong* topological phase does not rely on the microscopic translation symmetry for its topological protection. In particular, it is independent of the size of the unit cell.[6] This allows to coarse-grain the lattice and switch to an effective, continuous description (*c.f.* also Ref. [36] for a detailed discussion). For strong topological phases, we only need to require that the topological properties are realized when the system size is much larger than any microscopic length scale associated with the Hamiltonian. Consequently, during a Volterra process of a topological crystalline phase, we can assume that also the cut-out part is in the same topological phase.

A second-order topological phase protected by rotation symmetry has $(d-2)$ dimensional anomalous boundary states, for example isolated Majorana bound states at the corners of a two-dimensional crystal or chiral/helical modes at the hinges of a three-dimensional crystal. This is illustrated in Figs. 3(a) and (b). We require that the anomalous boundary excitations are *intrinsic* [11, 12, 25], i.e., we allow any changes of the crystal termination consistent with the rotation symmetry, for instance a decoration of the boundary with lower-dimensional topological phases. This property ensures that the anomalous boundary excitations are truly attributed to the topology of the $d$-dimensional bulk. Furthermore, throughout this article we focus on tenfold-way anomalous boundary states appearing in systems described by quadratic fermionic Hamiltonians.

## 3.2 Topological charge

The topological charge associated with an anomalous boundary state quantifies the anomaly. For topological insulators, helical hinge modes are characterized by a $\mathbb{Z}_2$ topological charge $Q \in \{0, 1\}$ measuring their existence. Chiral hinge modes are quantified by a $\mathbb{Z}$ topological charge $Q = n_+ - n_-$ defined as the difference of the number of forward-propagating ($n_+$) and

---

[6]In topological band theory, such as developed in Ref. [1], strong topological phases can be described in the long-wavelength expansion of the single-particle Hamiltonian. To connect to this approach, one needs to require that the momentum is well-defined in the long wavelength limit. As only the long-wavelength limit is required, the size of the unit cell can be chosen arbitrarily large.

backward-propagating ($n_-$) chiral modes. The Abelian groups $\mathbb{Z}_2$ and $\mathbb{Z}$ determine how the anomalous boundary states hybridize (fusion rules). For topological superconductors, Majorana corner modes and helical Majorana hinge modes have a $\mathbb{Z}_2$ topological charge, while chiral Majorana modes have a $\mathbb{Z}$ topological charge. Zero-energy eigenstates in Cartan classes AIII, BDI and CII are simultaneous eigenstates of the unitary chiral antisymmetry $\mathcal{C} = \mathcal{P}\mathcal{T}$ with eigenvalue $c = \pm 1$. This symmetry prohibits to hybridize and gap out zero-modes with the same eigenvalue $c$. Therefore, a $\mathbb{Z}$ topological charge is obtained by counting the number of zero-energy eigenstates weighted with their eigenvalue $c$.

Anomalous states always appear in pairs with canceling anomaly at the boundary or at defects of a topological bulk [31]. Consequently, in a closed system, isolated Majorana bound states, or Kramers pairs thereof, always come in pairs. The zero-dimensional anomalous states with $\mathbb{Z}$ topological charge occur in pairs with opposite eigenvalue under chiral antisymmetry. One-dimensional anomalous states form closed loops at the boundary or along defect lines of a topological bulk. For a three-dimensional system with anomalous hinge states, the number of inward and outward propagating modes intersecting any closed (or infinite open) surface needs to be equal and the associated topological charge needs to cancel.

In summary, for anomalous states with $\mathbb{Z}_2$ topological charge $Q_i$ the total topological charge $Q_{\text{tot}}$ needs to be even

$$Q_{\text{tot}} = \sum_i Q_i \quad \mod 2 = 0, \tag{5}$$

where for zero-dimensional anomalous states we sum over all anomalous states in the system, and for one-dimensional anomalous states we sum over all states intersecting an arbitrary closed (or infinite open) surface. Similarly, for zero- and one-dimensional anomalous states with $\mathbb{Z}$ topological charge $Q_i$, the total topological charge $Q_{\text{tot}}$ must vanish

$$Q_{\text{tot}} = \sum_i Q_i = 0. \tag{6}$$

## 3.3 Boundary-signature constraints from rotation symmetry

The presence of rotational symmetries leads to constraints on the possible boundary signatures. As we explain in the following, this can be seen by invoking the topological charge introduced above.

A rotation-symmetric sample can be divided into asymmetric sections. An asymmetric section is the maximal volume such that no two points in the volume are related by rotation symmetry. The rotation symmetry then relates the topological charge in symmetry-related sections. Because anomalous states always come in pairs ($Q_{\text{tot}} = 0$), asymmetric sections with non-zero topological charge can only exist in systems with even order of rotation symmetry, i.e., $C_2$, $C_4$ and $C_6$. The anomalous boundary signatures in rotation symmetric topological phases have also been discussed in Refs. [4–6, 8, 9, 11–27, 55].

For asymmetric sections exhibiting a non-zero $\mathbb{Z}$ topological charge, the internal action of rotation must invert the topological charge of the anomalous states to satisfy the anomaly cancellation criterion in Eq. (6). In particular, a rotation-symmetry protected second-order topological phase hosting anomalous zero-energy corner states in Cartan classes AIII, BDI and CII can exist only if the representation of rotation symmetry anticommutes with chiral antisymmetry. In this case, the chiral eigenvalue $c = \pm 1$ of states related by rotation symmetry alternates. Similarly, a second-order topological phase with chiral (Majorana) hinge modes may exist only in the presence of magnetic rotation symmetry. The reason is that the time-reversal operation is required to invert the propagation direction of modes related by symmetry.

For second-order anomalous states with $\mathbb{Z}$ topological charge protected by rotation symmetry, only a $\mathbb{Z}_2$ factor can be attributed to the bulk topology as an intrinsic boundary sig-

nature [4, 11, 12, 25]. This factor merely measures the existence of anomalous states but not their number. To illustrate this, consider a cubic crystal with chiral hinge modes, as depicted in Fig. 3(c), as an example of a second-order topological phase protected by magnetic fourfold rotation symmetry $C_4 \mathcal{T}$. A symmetry-allowed decoration with Chern insulators reverses the propagation direction of the chiral hinge modes, thereby changing their $\mathbb{Z}$ topological charge by an even number.

## 3.4 Volterra process with a second-order topological phase

In the following, we establish the existence of anomalous disclination states in a given second-order topological phase by performing a Volterra process. Recall that, in this section, we consider strong topological phases that do not rely on the presence of translation symmetries. Therefore, we allow to coarse-grain the lattice or break the translation symmetries. In such cases, disclinations are characterized only by their Frank angle $\Omega$. We lay out our arguments for two-dimensional systems. Nevertheless, they are generalized straightforwardly to $d > 2$ dimensions by considering a symmetric-pillar geometry and by applying the anomaly cancellation criterion to the $(d-2)$-dimensional hinge modes with respect to a plane perpendicular to the rotation axis.

A unique correspondence between bulk topology and disclination anomaly exists only if the system can be made locally indistinguishable from the bulk everywhere away from the disclination as a result of the Volterra process. This requires us to connect the lines/surfaces at the branch cut of the Volterra process using the appropriate hybridization terms given by the conditions imposed by Eq. (2) derived in the previous section. Below, we establish the correspondence for symmetry classes in which these conditions can be fulfilled.

In some symmetry classes, however, these conditions can not be satisfied. For instance, we showed in the previous subsection above that second-order topological phases with zero-energy states of $\mathbb{Z}$ topological charge exist only if rotation symmetry anticommutes with chiral antisymmetry. However, a chiral antisymmetry anticommuting with rotation symmetry forbids to construct bulk-equivalent hopping terms across the branch cut in the Volterra process, which follows from the central insight of our discussion in Sec. 2.5 above. Similarly, second-order topological phases hosting one-dimensional chiral hinge states require magnetic rotation symmetry, for which bulk-equivalent hopping terms across the branch cut are also not allowed (see again Sec. 2.5). These arguments can be generalized to second-order topological phases protected by rotation symmetry with $\mathbb{Z}$ anomalous boundary states in any dimension $d \geq 2$ (see Appendix B.4). Hence, only nonmagnetic second-order phases in symmetry classes with $\mathbb{Z}$ topological charge may allow for a bulk-boundary-defect correspondence. A detailed discussion on other symmetry classes where the branch-cut hopping condition in Eq. (2) cannot be satisfied is provided in Appendix B.

**Twofold rotation symmetry**

Twofold rotation-symmetry protected second-order topological phases host anomalous states on symmetry-related points of their boundary. The Volterra process to construct a $\pi$ disclination is illustrated in Fig. 4(a). The first step is to cut the sample into two symmetric halves. We require that the cutting process preserve the bulk and surface Hamiltonians except for the breaking of bonds along the cut. Therefore, the two symmetric halves host anomalous boundary states at corners with the same orientation as the original sample. Note that this requires the energy gap to close and reopen along the cut. Upon deforming the sample and hybridizing the bonds across the cut to complete the Volterra process, the two upper corners connect to form a smooth boundary. In the resulting sample, the bulk and all $(d-1)$-dimensional bound-

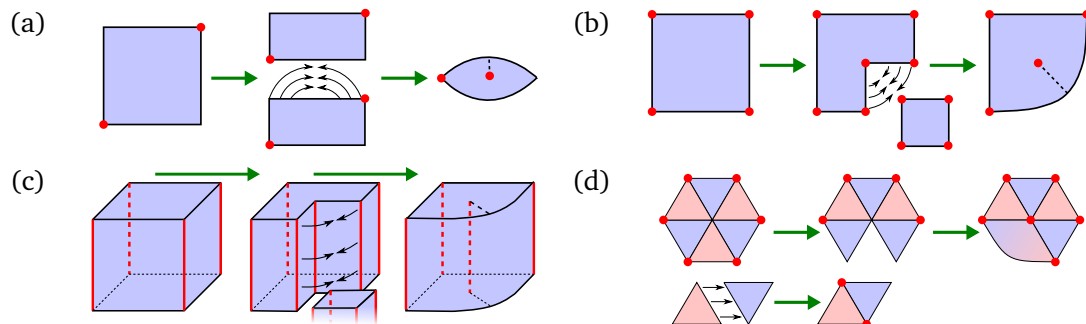

Figure 4: Volterra processes to construct disclinations in systems with second-order topology and different rotation symmetries: (a) twofold symmetry in two dimensions, (b) four-fold symmetry in two dimensions, (c) four-fold symmetry in three dimensions, (d) sixfold symmetry in two dimensions. Red dots and red lines indicate anomalous corner and hinge modes, respectively. In (d), adjacent triangles in red and purple are related by a sixfold rotation.

aries are gapped by construction. [7] Consequently, the anomalous state formerly located at one of the upper corners moves to the disclination in the process. An equivalent way to see this is to note that, upon gluing the surface, the boundary gap closes and reopens in the same way as when cutting the sample.

**Fourfold rotation symmetry**

In Fig. 4(b), we show the Volterra process for a fourfold rotation-symmetric system in two dimensions along with its boundary signatures. The removed segment in this process is itself fourfold symmetric and has, by the anomaly cancellation criterion, second-order boundary signatures at its corners. After deforming and gluing the other part, the resulting lattice has four singular points that may host zero-dimensional anomalous states: the three corners and the disclination. As the three corners remain unaffected during the deformation, the disclination has to host an anomalous state to satisfy the anomaly cancellation criterion. Fig. 4(c) shows the same process for a three dimensional system.

**Sixfold rotation symmetry**

We consider a sixfold symmetric sample in the shape of a hexagon. It can be divided into six equilateral triangles as demonstrated in Fig. 4(d). A two-dimensional second-order topological phase protected by sixfold rotation symmetry hosts anomalous boundary states on symmetry-related corners of a hexagonal sample. Since each triangle has only threefold rotation symmetry, there are two types of triangles related by a sixfold rotation. The anomaly cancellation criterion together with threefold rotation symmetry requires that the topological charge at each corner of the triangle must cancel, i.e., the topological charge at each corner, if present, must be even. In order for the hexagonal sample to exhibit its anomalous corner states, hybridizing two triangles along a shared boundary needs to close and reopen the excitation gap to create a pair of anomalous states [see Fig. 4(d)]. Conversely, breaking the bonds between two triangles closes and reopens the gap along the shared boundary, thereby removing the anomalous corner states. Putting all triangles together results in six anomalous states at the center of the hexagon, which gap out upon hybridization.

---

[7]Note that the corner state at the corner that is connected to its partner during the Volterra process can not move to the other, unrelated corner because the initial $(d-1)$-dimensional boundaries (not created by the cut) remain gapped during the process.

In the first step of the Volterra process, we remove a triangle from the hexagon. This requires to break the bonds between adjacent triangles of opposite orientation. By the arguments above, the excitation gap closes and reopens along both of the cut lines, thereby removing the anomalous corner states. In the second step, we deform one triangle adjacent to the cut to glue the sample back together. This process rotates the part of the deformed triangle close to the cut by $\pi/3$. As the type of triangle is determined by the orientation of the triangle in space, the deformation smoothly interpolates between the two types as defined above. Thus, hybridizing the deformed sample across the cut creates a pair of anomalous boundary states, one at the disclination and one at the corner. The disclination state appears because the gap closes and reopens an odd number of times during the Volterra process: once along each of the two cut lines when removing a triangle, and once when gluing the edges back together.

## 3.5 Summary of results

In this section, we have shown that *strong* second-order topological phases protected by $C_2$, $C_4$, or $C_6$ symmetries host anomalous disclinations states provided that the symmetry conditions summarized at the end of Sec. 2.5 are satisfied. In particular, we have argued that only nonmagnetic phases in symmetry classes with $\mathbb{Z}_2$ topological charge may allow for such a bulk-boundary-defect correspondence.

In the following section, we will generalize these results to include effects coming from the presence of translational symmetries. In Section 6, we will apply these concepts to a few examples of superconductors hosting Majorana bound states or Kramers pairs thereof at disclinations, and to time-reversal symmetric insulators hosting a helical disclination mode. An example for the appearance of a domain wall at a disclination in a magnetic topological insulator hosting chiral hinge states is given in Appendix B.3.

## 4 Disclinations in topological crystals

Our considerations thus far have been independent of translational symmetries. In this section, we extend our arguments to lattice models with discrete translation symmetries. As discussed in Sec. 2, disclinations in lattices are classified into topological equivalence classes according to their rotation *and* translation holonomy. Real-space representations of topological crystalline phases naturally including translation symmetries can be constructed using the framework of *topological crystals* [22]. Below, we briefly review and discuss the essential steps of the topological-crystal construction applied to lattices with rotation symmetries. Moreover, we extend the recipe developed in Ref. [22] by showing how to relate the constructed real-space representation to weak and higher-order topological phases obtained from other classification schemes [12, 25, 67]. We then apply the topological-crystal construction to determine the existence of anomalous states at disclinations of all types. As the topological crystal construction was originally performed for periodic samples, we briefly comment in App. C.4 on the validity of the approach for finite size samples or in the presence of inhomogeneities. Finally, we provide a summary of the main results of this section.

### 4.1 Cell decomposition

The first step in the topological-crystal construction is the covering of the system's lattice with specific types of cells as detailed below. Consider a $d$-dimensional space $\mathbb{R}^d$ subject to the symmetry group $G \times G_{int}$. As only topological crystalline phases protected by translation and/or rotation symmetry can contribute to the anomaly at a disclination (see Sec. 5.4 below), we

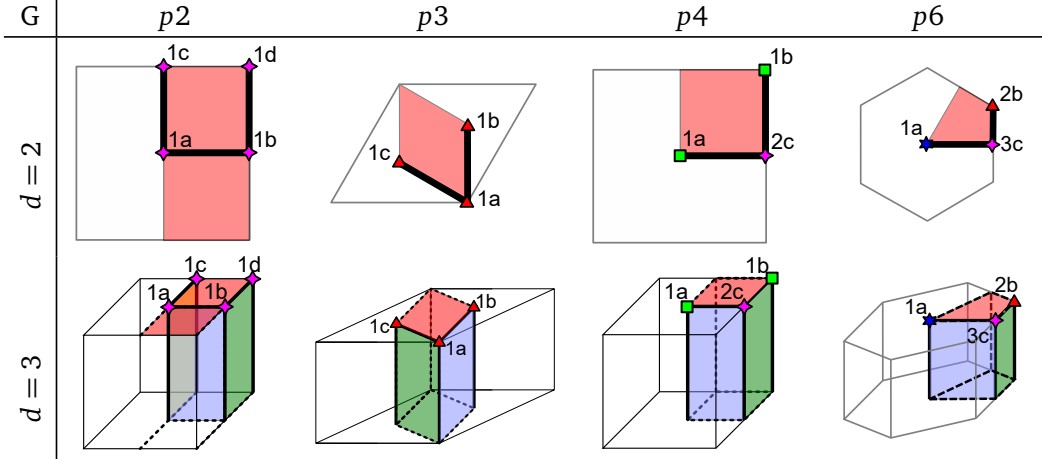

Figure 5: Cell decompositions of unit cells in space group $G$ in dimensions $d = 2, 3$: Pink stars, red triangles, green squares and blue stars denote inequivalent twofold, threefold, fourfold and sixfold rotation axes, respectively. Colored areas and bold lines denote inequivalent 2-cells and 1-cells, respectively. In two dimensions, the 2-cell is the asymmetric unit. The 0-cells coincide with the rotation axis. In three dimensions, the asymmetric unit is a 3-cell whose hinges are denoted by dotted lines. The 0-cells lie at the end of the rotation axes. We use the standard labels for Wyckoff positions.

focus on the (magnetic) space groups $G = pn$ ($G = pn'$) generated by $n$-fold (magnetic) rotation symmetry and translations. Further note that only ($n \in \{2, 3, 4, 6\}$)-fold rotations are compatible with translation symmetry. First, one defines an *asymmetric unit* (AU) as the interior of the largest region in $\mathbb{R}^d$ such that no two distinct points in this region are related by a crystalline symmetry $g \in G$. A cell complex structure is generated by copying the AU throughout $\mathbb{R}^d$ using all elements of the space group $G$. Next, one places cells of dimension $(d-1)$ on faces where adjacent AUs meet. Throughout the following, cells of spatial dimension $d_b$ are denoted as $d_b$-cells. These cells are chosen as large as possible, such that no two distinct points in the same cell are related by a crystalline symmetry. Furthermore, cells are not allowed to extend over corners or hinges of the AUs. In the same way, one continues iteratively by placing $(d-n-1)$-cells on faces where $(d-n)$-cells coincide. We present the resulting cell complex structures for $p2$, $p3$, $p4$ and $p6$ in two and three dimensions in Fig. 5 (see Appendix C.1 for details on their construction).

## 4.2 Decoration of cells with topological phases

The considered space is filled with matter by decorating the $d_b$-cells with $d_b$-dimensional topological phases. The topological phases have to satisfy all internal symmetries of the cell. Furthermore, a cell located on a mirror plane or on a rotation axis can only be decorated with topological phases satisfying the crystalline symmetries that leave the cell invariant. As one aims to construct only phases with an excitation gap in the bulk, one also requires that gapless modes on adjacent faces or edges of the decorated cells gap out mutually.

The tenfold-way topological phases have an Abelian group structure where the group operation is the direct sum "$\oplus$" of two Hamiltonians [1,3,31]. Topological crystals constructed as decorations with tenfold-way topological phases inherit this Abelian group structure. This allows to choose a set of generators from which all topological crystals can be constructed using the direct sum and symmetry-allowed deformations of the generating topological crystals.

The labels *weak* and *strong* for topological crystalline phases refer to the behavior of the topological crystalline phase under breaking of translation symmetry. A topological crystalline phase is called weak if its topological invariant can be changed by a redefinition of the unit cell, thereby breaking the translation symmetry of the original crystal. If this is not possible, the topological crystalline phase is termed strong. For a topological crystal, we determine whether it is weak or strong using the following procedure: we first double the unit cell by combining two adjacent unit cells of the original crystal. After this redefinition we allow for symmetric deformations to express the result in terms of generating topological crystals. A topological crystal that remains invariant during this procedure corresponds to a strong topological crystalline phase.

Furthermore, we identify the order of the topological crystal from its boundary signature. A topological crystal corresponding to a decoration of $d_b$ cells has a $(d_b - 1)$-dimensional boundary signature. This is because its anomalous boundary states are inherited from the decoration. Hence, it represents a topological phase of order $(d - d_b - 1)$.

## 4.3 Decorations for rotation-symmetric lattices

Having reviewed the general procedure and nomenclature of the topological-crystal construction, we now focus on the special case of space groups with rotational symmetries. In particular, we determine all possible weak and second-order topological phases using the topological-crystal construction.

Without specifying the set of internal symmetries, we first work out for each space group the set of generating topological crystal decorations. For these generators, we then check whether they describe a weak or a strong topological phase, and finally determine the order of this phase. The group of internal symmetries together with the space group and the algebraic relations of their representations finally decides whether the decoration is valid. In other words, we determine whether the $d$-dimensional topological phases required to decorate the $d$-cells of the system exist for a given set of internal symmetries, and whether the anomalous states at the cell interfaces can be gapped out. In particular, to decide whether the asymmetric unit can be decorated with a topological phase, we have to ensure that all boundaries, corners, and hinges are gapped at the rotation axis.

For three-dimensional systems we omit decorations of 1-cells parallel to the rotation axis as they cannot give rise to anomalous states with the same dimension as the disclination line. Moreover, the plane perpendicular to the rotation axis corresponds to a two-dimensional rotation-symmetric system. The decorations in this plane acquire the label weak because their topological invariant can be changed by a redefinition of the unit cell in the $z$ direction.

We defer the detailed derivation of the generating 1-cell (2-cell) decorations for rotation-symmetric lattices in two (three) dimensions to Appendix C.2. Below, we present the generating sets together with their properties for decorations with $\mathbb{Z}$ topological phases. The results for decorations with $\mathbb{Z}_2$ phases are straightforwardly obtained by taking the topological charge of the decorations modulo two.

*Twofold rotation symmetry.* With twofold rotation symmetry, there exist two distinct weak topological phases and one strong second-order topological phase, which we depict in Figs. 6(a)-(c).

*Threefold rotation symmetry.* For threefold symmetry, there is no valid 1-cell decoration (2-cell decoration parallel to the rotation axis). The reason is that each 1-cell (2-cell) ends at a threefold rotation axis at which the anomaly cancellation criteria (5) and (6) cannot be satisfied locally. Thus, there are neither weak nor second-order topological phases with threefold rotation symmetry.

*Fourfold rotation symmetry.* In a two (three) dimensional lattice with fourfold rotation symmetry, all 1-cell decorations (2-cell decorations parallel to the rotation axis) are generated

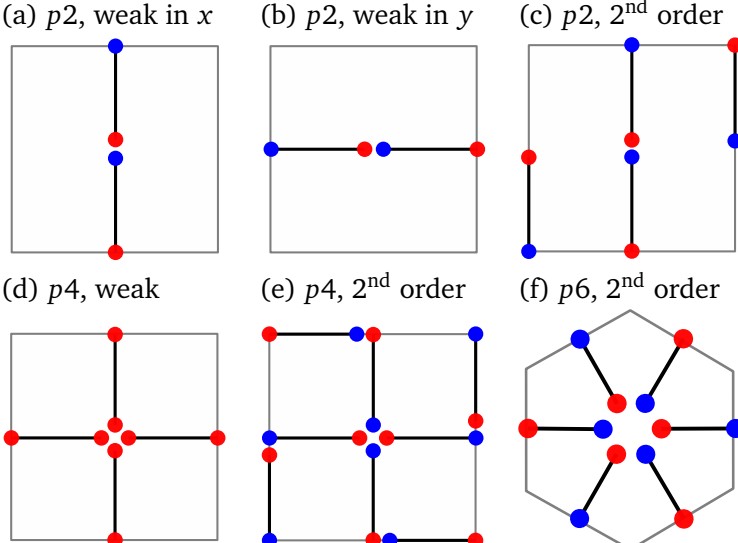

(a) $p2$, weak in $x$   (b) $p2$, weak in $y$   (c) $p2$, 2$^{\text{nd}}$ order

(d) $p4$, weak   (e) $p4$, 2$^{\text{nd}}$ order   (f) $p6$, 2$^{\text{nd}}$ order

Figure 6: Generating sets of valid 1-cell decorations (cross sections of 2-cell decorations parallel to the rotation axis) of two (three) dimensional lattices with twofold, fourfold, and sixfold rotation axes. Black lines denote one (two) dimensional topological phases with anomalous boundary states depicted as dots: for $\mathbb{Z}$ topological phases, the red and blue dots distinguish between topological charges $q = +1$ and $q = -1$, respectively. For $\mathbb{Z}_2$ topological phases with topological charges $q \mod 2$, the two colors are equivalent.

from one weak and one strong second-order topological phase, which we show in Figs. 6(d) and (e). The weak phase exists only for 1-cell (2-cell) decorations with $\mathbb{Z}_2$ topological phases.

*Sixfold rotation symmetry.* With sixfold rotation symmetry, the only valid 1-cell decoration (2-cell decoration parallel to the rotation axis), which we depict in Fig. 6(f), corresponds to a strong second-order topological phase.

Furthermore, in a symmetry class in which the $(d-2)$-dimensional anomalous states have $\mathbb{Z}$ topological charge, one can show that the direct sum of a strong second order topological phase with itself can be adiabatically deformed such that no $(d-2)$-dimensional anomalous states remain in the system (see Appendix C.2). This holds for all $(n = 2, 4, 6)$-fold symmetric systems. It confirms our statement from the end of Sec. 3.3 that only a $\mathbb{Z}_2$ factor of the topological charge of anomalous boundary states in these systems is an intrinsic property of the topological bulk.

## 4.4 Weak and second-order topological phases with disclinations

We are now equipped to study disclinations in the weak and second-order topological phases constructed above. In particular, we determine for each generator (see Fig. 6) whether it hosts topological disclination states. We realize this by decorating a lattice with disclination, as constructed through a Volterra process, with its topological-crystal limit. The disclination breaks rotation symmetry locally, thus only internal symmetries constrain the hybridization of disclination states.

We require that the system is locally indistinguishable from the bulk along the branch cut. As we have shown in Sec. 3.4, this is not possible in symmetry classes that host rotation-symmetry protected second-order topological phases with $\mathbb{Z}$ topological charge. In fact, in these symmetry classes a decoration with weak or second-order topological phases represents an obstruction to forming a lattice with an isolated disclination (see Appendix C.3). We there-

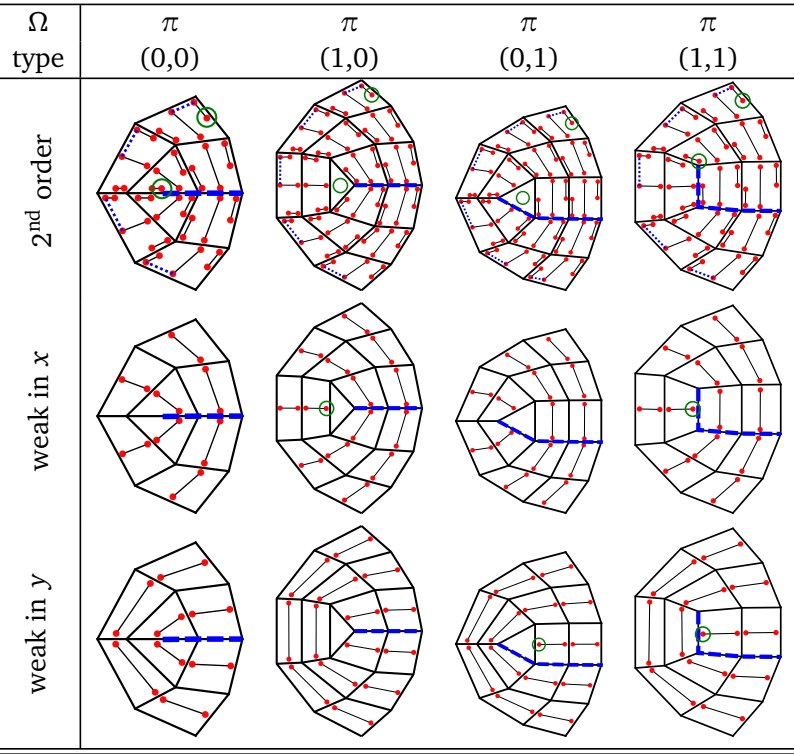

Figure 7: Decorations of twofold rotation-symmetric lattices with $\pi$ disclinations of all types as defined in Fig. 2: decorations with second-order topological phases protected by twofold rotation symmetry and decorations with weak topological phases as stacks in the $x$ and $y$ directions. Red dots represent $d-2$ dimensional anomalous states with $\mathbb{Z}_2$ topological charge. Dashed blue lines indicate the branch cut in the Volterra process across which anomalous states hybridize. Green circles denote locations where unpaired anomalous states remain. Note that for weak phases hosting an anomalous disclination state, there is also an odd number of anomalous states on the boundary.

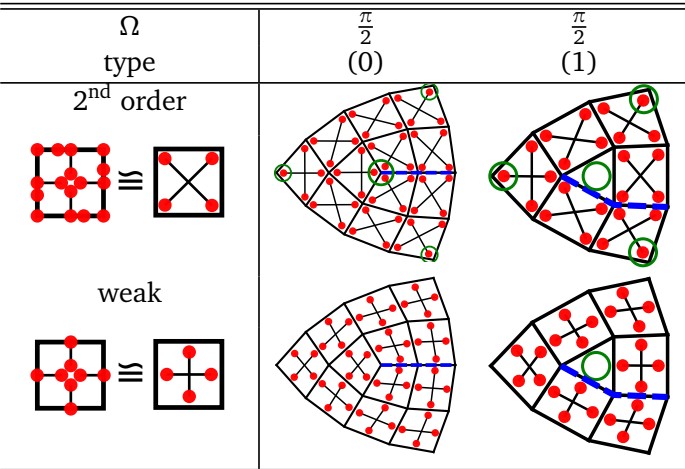

Figure 8: Decorations of fourfold-symmetric lattices with $\pi/2$ disclinations of both types: decorations with second-order topological phases and decorations with weak topological phases. The left column depicts the corresponding topological crystals. For simplicity, anomalous bound states hybridizing within a unit cell are not shown. We use the same symbols and colors as in Fig. 7.

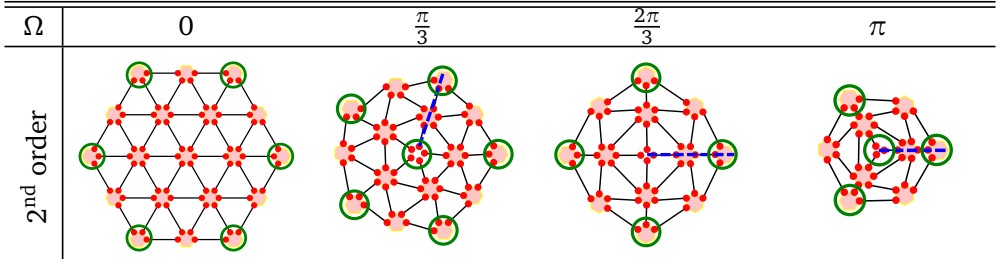

Figure 9: Unique decoration patterns of a sixfold rotation symmetric lattice without disclination ($\Omega = 0$) and with disclination of Frank angle $\Omega \neq 0$: decorations with second-order topological phases of $\mathbb{Z}_2$ topological charge. Symbols and colors are as in Fig. 7.

fore restrict our discussion to symmetry classes whose $d-2$ dimensional anomalous states have $\mathbb{Z}_2$ topological charge.

Our results for the corresponding decorations of twofold, fourfold, and sixfold symmetric lattices are presented in Figs. 7, 8, and 9. The decoration pattern is unique up to an arbitrary decoration at the disclination itself, which cannot change the total topological disclination charge due to the anomaly cancellation criterion of Sec. 3.2. Therefore, the unique bulk decoration pattern determines the existence of anomalous states at the topological lattice defect.

In three dimensions, also screw disclinations with a translation holonomy $T_z$ along the rotation axis may occur. However, by a local rearrangement of the lattice a screw disclination can be separated into a disclination with trivial translation holonomy $T_z = 0$ and a screw dislocation carrying the translation holonomy $T_z$. The topological charge bound to the topological lattice defects depends only on its holonomies defined on a loop enclosing the defects. These defects can be pulled apart arbitrarily. Hence, the topological charge at a defect with multiple non-trivial holonomies can be determined from the sum of the topological charge at the individual defects with a single non-trivial holonomy. The arguments of Ref. [46] show that the weak topological phases obtained by stacking 2D first-order phases along the rotation axis contribute $(d-2)$-dimensional anomalous states to screw dislocations with odd translation holonomy $T_z$. While Ref. [46] considered explicitly weak topological insulators in class AII, their arguments generalize straightforwardly to other symmetry classes as well [8] [47].

Using the decorations constructed above, we deduce that the contributions of weak and second-order topological phases with topological invariants $G_v = \left( v_x, v_y, v_z \right)^T$ and $v_{2\pi/n}$ [9],

---

[8] In particular, Ref. [46] considers a finite sample with screw dislocation. The screw dislocation is the termination of two step edges on the two opposite surfaces perpendicular to the screw dislocation. In a corresponding weak topological phase, these step edges host one-dimensional anomalous states. As the one-dimensional anomalous state cannot terminate at the screw dislocation, it must continue along the screw disclination and connect to the opposite surface. This argument holds for all symmetry classes hosting one-dimensional anomalous states.

[9] The topological invariants $v_i$, $i = x, y, z$ and $v_{2\pi/n}$ for the weak and second order topological phases, respectively, can be abstractly defined through the isomorphism $v$ from the abelian group of stable topological equivalence classes of Hamiltonians [1] to its isomorphic abelian group of integers $\mathbb{Z}$ or $\mathbb{Z}_2$, with addition as group operation and generator "1". For second-order topological phases, we argued in section 3 that their classifying group is isomorphic to $\mathbb{Z}_2$. In this case, the quantity $v_{2\pi/n}$ can be abstractly defined as a quantity that takes the value "0" in the trivial phase and "1" in the topological phase. For weak topological phases, the classifying group can be isomorphic to $\mathbb{Z}_2$ or $\mathbb{Z} \simeq 2\mathbb{Z}$. In the latter case, the quantity $v_i$ should take the value "1" or "2" for the generator of the abelian group of topological equivalence classes of Hamiltonians and respect the abelian group property in its domain and image. Here, we distinguish the cases where the symmetries constrain a topological invariant to be even, see also the discussion in Sec. 5. This isomorphism is typically expressed for a periodic system as a quantity of the reciprocal space Hamiltonian, such as the Chern number [31], or a symmetry-based indicator [68]. Explicit expressions for the topological invariants for topological superconductors in Cartan class D using symmetry-based indicators are presented in Appendix E. In general, explicit expressions for topological invariants of higher-order

respectively, to the number of $(d-2)$-dimensional anomalous states $\theta_n^{\text{disc}}$ at a disclination with rotation holonomy $\Omega$ and translation holonomy $\boldsymbol{T}$ are

$$\theta_2^{\text{disc}} = \frac{\Omega}{\pi}\nu_\pi + \boldsymbol{T} \cdot \boldsymbol{G}_\nu \quad \text{mod } 2, \tag{7}$$

$$\theta_3^{\text{disc}} = T_z \nu_z, \tag{8}$$

$$\theta_4^{\text{disc}} = \frac{2\Omega}{\pi}\nu_{\pi/2} + \boldsymbol{T} \cdot \boldsymbol{G}_\nu \quad \text{mod } 2, \tag{9}$$

$$\theta_6^{\text{disc}} = \frac{3\Omega}{\pi}\nu_{\pi/3} + T_z \nu_z \quad \text{mod } 2. \tag{10}$$

In two dimensions, the dimension spanned by the $z$ direction is absent such that $T_z$ and $\nu_z$ are absent. Furthermore, recall that in Eq. (9), fourfold rotation symmetry requires that $\nu_x = \nu_y$ such that for a $\pi/2$ disclination of type 1, the two equivalent translation holonomies $\boldsymbol{T} = (0,1)$ and $\boldsymbol{T} = (1,0)$ always yield the same result.

For symmetry classes whose $(d-2)$-dimensional anomalous states have $\mathbb{Z}_2$ topological charge, these equations not only predict the total parity of anomalous states at disclinations, but also at dislocations and at collections thereof. First, Eqs. (7) to (10) are also valid for dislocations: for zero Frank angle, the equations agree with the familiar result for dislocations [50]. Second, Eqs (7) to (10) depend only on the holonomical quantities of a loop around the defect. This allows to perform local rearrangements of the lattice at the defect, which in particular allows to split the defect into multiple defects. By regarding the holonomy of a loop around each of these defects, one can apply Eqs. (7) to (10) to each defect individually. This shows that Eqs. (7) to (10) determine the parity of anomalous states of collections of defects from the holonomy of an enclosing loop. As a consequence, Eqs. (7) to (10) also determine the fate of the defect anomalies upon splitting and, conversely, fusion of lattice defects.

## 4.5 Summary of results

Using the topological crystal construction, we have shown that $d = 2$ and $d = 3$ dimensional weak and second-order topological phases with $\mathbb{Z}_2$ topological charge host anomalous states at disclinations. We have summarized these results in Eqs. (7)–(10) relating the number of states bound to a disclination to the weak and second-order invariants of the bulk. Recall that the first-order (or ten-fold way) invariants of weak and higher-order topological phases are zero. A complete picture of the bulk-boundary-defect correspondence in rotation-symmetric systems, however, also has to contain the first-order contributions to the number of disclination states. This will be done in the next section.

# 5 Bulk-boundary-defect correspondence

The bulk-boundary correspondence and the bulk-defect correspondence link the bulk topological invariant of a sample to the existence of anomalous states at its boundaries and its defects, respectively. Cumulating our results from previous sections, we now determine the precise relationship between the topological charge at point defects and bulk topological invariants in rotation-symmetric systems.

First, we work out the relation between disclinations and anomalous states for first-order topological phases to derive a general formula relating the number of anomalous states at disclinatinons to the first-order, second-order, and weak topological invariants of a rotation-symmetric system with $\mathbb{Z}_2$ topological charge. This represents the central result of our work.

---

topological phases can be derived using the method outlined in Refs. [67, 69].

After that, we discuss implications of our findings, such as the presence of domain walls in systems where the bulk-boundary-defect correspondence fails. We further argue that, in certain symmetry classes, a bulk-boundary-defect correspondence also holds for systems with $\mathbb{Z}$ topological charge. We continue with a recipe on how to apply our results to systems with additional symmetries besides rotational and translational symmetry. Finally, we present complete classification tables for all Cartan classes in 2D and 3D showing the possible first-order, second-order, and weak topological crystalline phases and whether the bulk-boundary-defect correspondence holds.

## 5.1 Topological invariants and topological charge at disclinations

In Sec. 2, we argued that the rotation and translation holonomies of disclinations on the one hand and bound geometric $\pi$-flux quanta on the other hand are distinct topological properties of point defects. In the previous Sec. 4, we worked out the contributions of weak and second-order topological phases to the topological charge at a disclination. Conversely, to determine the bulk topological invariants from the topological charge at the lattice defect, we first need to investigate under which conditions a *first-order* topological phase hosts anomalous states at a point defect.

Our detailed analysis is presented in Appendix D. We find that tenfold-way (first-order) topological phases, which are independent of crystalline or internal unitary symmetries for their protection, host anomalous states at disclinations only if bound to a geometric $\pi$-flux quantum. This general result is in agreement with case studies in the literature indicating that these phases may bind $(d-2)$-dimensional anomalous states to point and line defects with geometric $\pi$-flux quanta [31, 47, 48, 63, 70–72] if allowed in the respective symmetry class. Furthermore, we find that this statement can be generalized to all dimensions $d \geq 2$ (see Appendix D.2). As a consequence, a tenfold-way topological phase in a symmetry class that allows for $(d-2)$-dimensional anomalous defect states hosts an anomalous state at a disclination if and only if the disclination also binds a geometric $\pi$-flux quantum.

Hence, we can express the total number of $(d-2)$-dimensional anomalous state $\theta_n$ at a disclination in an $n$-fold rotation-symmetric system, in the presence of a unitary rotation symmetry whose representation commutes with all internal unitary symmetries and antisymmetries (see Sec. 2.5), as

$$\theta_n = \theta_n^{\text{disc}} + \frac{\alpha}{\pi} \nu_1 \mod 2, \tag{11}$$

where $\theta_n^{\text{disc}}$ is defined as in Eqs. (7)–(10), $\alpha = l\pi$, $l \in \mathbb{Z}$, is the geometric phase obtained by parallel transporting a particle around the defect, and $\nu_1$ is the tenfold-way strong first-order topological invariant of the system's bulk Hamiltonian. In the presence of time-reversal symmetry, the geometric phase $\alpha = \frac{\phi}{\phi_0}$ is given by the quantized magnetic flux $\phi = l\phi_0$ bound to the defect, where $\phi_0 = \frac{hc}{2e}$ is the magnetic flux quantum. [10]

One can determine the second-order topological invariant $\nu_{2\pi/n}$ and the weak topological invariants $\mathbf{G}_\nu = (\nu_x, \nu_y, \nu_z)$ by probing disclinations with different translation holonomies. The parity of the first-order topological invariant is determined from the existence of an anomalous state at a $\pi$-flux point defect. For phases of matter that obey rotation symmetry but no

---

[10]The flux quantization in time-reversal symmetric systems is a result of the following Gedankenexperiment: If a defect of codimension 2 binds a magnetic flux that is a multiple of $\phi_0 = \frac{hc}{2e}$, then any electron transported around that path acquires a geometric phase that is a multiple of $\pi$. Under time-reversal symmetry, both the flux and the corresponding geometric phase are reverted. As the phase of the wavefunction is defined only modulo $2\pi$, a time-reversal symmetric system with defect with flux a multiple of $\phi_0 = \frac{hc}{2e}$ still satisfies a time-reversal symmetry. As all other values of the flux break time-reversal symmetry, the requirement of time-reversal symmetry quantizes the flux to multiples of $\phi_0 = \frac{hc}{2e}$.

translation symmetry, Eqs. (7)–(11) apply with trivial translation holonomy $T = 0$, in agreement with our arguments in Sec. 3. Equations (7)–(11) are the central result of this paper.

Putting our results into perspective, for lattices with rotation symmetry we have shown: (i) as any anomaly at a disclination has to be canceled somewhere else in the system, any crystalline topological phase with an anomalous state at a disclination also hosts anomalous boundary states. This provides a sufficient condition for the existence of anomalous boundary states based on the topological properties of the lattice defect alone. (ii) Each crystalline-symmetry protected topological phase contributes anomalous states only to defects that carry a holonomy of the protecting crystalline symmetry. As summarized in Eqs. (7) to (10), the rotation holonomy only contributes to disclination states in the presence of a nontrivial second-order topological invariant, whereas the translation holonomy only contributes if there are nonzero weak topological invariants. This establishes a bridge between two distinct phenomenologies of topological crystalline phases: their higher-order bulk-boundary correspondence on the one hand and their topological response with respect to topological lattice defects [36] on the other hand.

## 5.2 Implications and remarks

Having presented our central result, we now discuss consequences that follow from our findings.

If the representation of the unitary rotation symmetry does not commute with all internal unitary symmetries and antisymmetries, or if the symmetry group does not contain a unitary rotation symmetry, the cut in the Volterra process remains *distinguishable* from the bulk (see Sec 2.5). In these cases, the disclination must be the end point of a domain wall between regions distinguishable by their local arrangements of orbitals in the unit cell (see Appendix B.1 for details). Moreover, the hopping across the domain wall is not restricted to a unique pattern. Thus, we cannot determine the topological charge at the disclination from the bulk and the lattice topology alone. We can only make a statement about the parity of the topological charge along the domain wall, for which we refer the interested reader to Appendix B.2.

In three dimensions, a disclination may also host a one-dimensional stack of zero-dimensional anomalous states whose pairwise annihilation is prohibited by translation symmetry along the lattice defect. These states exist in the presence of weak topological phases obtained by stacking two-dimensional topological phases along the defect direction. The contributions of stacks of two-dimensional first-order, second-order and weak topological phases are determined by similar equations as Eqs. (7) to (11), where the topological invariants $\{\nu_1, \nu_{2\pi/n}, \nu_x, \nu_y, \nu_z\}$ must be replaced by $\{\nu_z, \nu_{2\pi/n,z}, \nu_{x,z}, \nu_{y,z}, 0\}$ measuring the presence of weak topological phases obtained by a stack of two-dimensional strong first-order ($\nu_z$), second-order ($\nu_{2\pi/n,z}$) and two-dimensional weak topological phases ($\nu_{x,z}, \nu_{y,z}$) along the defect direction. There are no contributions proportional to the translation holonomy $T_z$. Therefore, $\nu_z$ can be replaced by 0.

## 5.3 Disclinations in systems with $\mathbb{Z}$ topological charge

Eqs. (7) to (11) were derived for symmetry classes whose $(d-2)$-dimensional anomalous states have $\mathbb{Z}_2$ topological charge. Nevertheless, also for symmetry classes with $\mathbb{Z}$ topological charges we can make a couple of statements. For this purpose, we note that these symmetry classes can be divided into two subsets (which are contained in Tables 1 and 2 below):

(i) In symmetry classes that, at the same time, allow for strong, rotation-symmetry protected second-order topological phases, our arguments from Sec. 3 show that the anomaly at the disclination depends on the microscopic properties of the system. This situation gives again rise to the appearance of a domain wall, as discussed in detail in Appendix B.

(ii) In the remaining subset of symmetry classes where strong, rotation-symmetry protected second-order topological phases are forbidden, the anomaly at the disclination may still be determined from the bulk topology alone. In particular, in three-dimensions these symmetry classes may allow for two-dimensional first-order topological phases stacked along the rotation axis. Their contribution to the number of one-dimensional anomalous states at a disclination or dislocation is $\theta = \nu_z T_z$. This is in fact the only contribution, as both first-order topological phases as well as weak topological phases obtained by stacking two-dimensional first-order topological phases in the $x$- or $y$-direction are forbidden in these symmetry classes. The former statement follows from the structure of the tenfold-way classification. The latter is a consequence of the topological crystal construction (see Appendix C). Tables 1 and 2 below contain a list of the corresponding symmetry classes in two and three dimensions, respectively.

### 5.4 Presence of additional symmetries

Disclinations may exist in all space groups with rotation symmetry. For space groups with additional symmetry elements other than translations and rotations, our findings apply as follows: a strong crystalline topological phase hosts anomalous disclination states if it realizes a strong second-order topological phase after lifting all symmetry constraints except for rotation symmetries. This statement holds because we have shown the existence of anomalous disclination states for all Hamiltonians realizing second-order topological phases, which may also satisfy additional symmetries. This allows us to identify the anomaly at disclinations by considering only topological crystalline phases with translation, rotation and internal symmetries. Note, however, that we also have to respect any additional crystalline symmetries when we construct the hopping terms across the branch cut of the Volterra process, as defined in Eq. (2).

Furthermore, the presence of additional internal unitary symmetries $\mathcal{U}$ may protect anomalous states at points with $\mathcal{U}$-symmetry flux in first-order topological phases (see Appendix D.3).

### 5.5 Application to all Cartan classes

To complete our discussion, we summarize our results in Tables 1 and 2. We present a classification of first-order, second-order, and weak topological crystalline phases with real-space limits as in Fig. 6 for all Cartan classes describing spinful fermions with magnetic and non-magnetic rotation symmetry in two and three dimensions. Those cover all topological phases that give rise to $d-2$ dimensional anomalous states at disclinations in rotation-symmetric lattices. For each symmetry class we determine whether the disclinations have to be connected to a domain wall as discussed in section 2.5. This criterion determines whether the bulk-boundary-defect correspondence holds (the disclination is not associated to a domain wall), or whether it does not hold (domain wall necessarily exists).

A few remarks are in order. For spinful fermions, rotation symmetry satisfies $U(\mathcal{R}_{2\pi/n})^n = -1$ and commutes with time-reversal symmetry $U(\mathcal{R}_{2\pi/n})U(\mathcal{T})K = U(\mathcal{T})U(\mathcal{R}_{2\pi/n})^*K$, where $K$ denotes complex conjugation. Spin-rotation symmetry, if present, can be combined with rotation symmetry such that the representation of rotation symmetry within a spin subspace satisfies $U(R_{2\pi/n})^n = 1$. For superconductors, our discussion covers all possible pairing symmetries for which $u(R_{2\pi/n})\Delta(R_{2\pi/n}\mathbf{k})u^\dagger(R_{2\pi/n}) = e^{i\phi}\Delta(\mathbf{k})$, where $u(R_{2\pi/n})$ is the representation of $n$-fold rotation symmetry acting on the normal-state Hamiltonian (see Appendix F.1 for details). We interpret Cartan class AIII as a time-reversal symmetric superconductor with $U(1)$ spin-rotation symmetry. Cartan class BDI is interpreted as a superconductor of spinless fermions, in which time-reversal symmetry and particle-hole antisymmetry square to 1 and the crystalline symmetry operations satisfy $U(R_{2\pi/n})^n = 1$ and commute with time-reversal symmetry. We interpret Cartan class CII as a superconductor of spinful fermions with spin-rotation symmetry and an emergent time-reversal symmetry $\mathcal{T}^2 = -1$ in each spin subspace. Also in

Table 1: Classification of two-dimenional $n$-fold rotation-symmetric weak, second-order, and first-order topological superconductors that give rise to anomalous boundary states. $Q$ indicates the topological charge of zero-dimensional anomalous states in the given Cartan class. $G$ denotes the space group $pn$ ($pn'$) with $n$-fold unitary (magnetic) rotation symmetry. $\phi$ is the $U(1)$ phase determining the superconducting pairing symmetry. The three central columns show the structure of the topological invariants. For systems with first-order invariant $2\mathbb{Z}$, the rotation symmetry constrains the Chern number to be even. The last column indicates whether the bulk-boundary-defect correspondence (BBDC) holds at a disclination ($\checkmark$). The remaining (non-superconducting) Cartan classes do not allow for zero-dimensional anomalous states, i.e., they have $Q = 0$.

| Cartan class | $Q$ | $G$ | $\phi$ | Weak in $x$ / $y$ | $2^{\text{nd}}$ order | $1^{\text{st}}$ order | BBDC |
|---|---|---|---|---|---|---|---|
| D | $\mathbb{Z}_2$ | $p2$ | $0$ | $\mathbb{Z}_2^2$ | $\mathbb{Z}_2$ | $\mathbb{Z}$ | $\checkmark$ |
| | | | $\pi$ | - | - | $2\mathbb{Z}$ | $\checkmark$ |
| | | $p4$ | $0,\pi$ | $\mathbb{Z}_2$ | $\mathbb{Z}_2$ | $\mathbb{Z}$ | $\checkmark$ |
| | | | $\frac{\pi}{2},\frac{3\pi}{2}$ | - | - | $2\mathbb{Z}$ | $\checkmark$ |
| | | $p6$ | $0,\frac{2\pi}{3},\frac{4\pi}{3}$ | - | $\mathbb{Z}_2$ | $\mathbb{Z}$ | $\checkmark$ |
| | | | $\pi,\frac{\pi}{3},\frac{5\pi}{3}$ | - | - | $2\mathbb{Z}$ | $\checkmark$ |
| | | $p2'$ | | $\mathbb{Z}_2^2$ | $\mathbb{Z}_2$ | - | - |
| | | $p4'$ | | $\mathbb{Z}_2$ | $\mathbb{Z}_2$ | - | - |
| | | $p6'$ | | - | $\mathbb{Z}_2$ | - | - |
| DIII | $\mathbb{Z}_2$ | $p2$ | $0$ | $\mathbb{Z}_2^2$ | $\mathbb{Z}_2$ | $\mathbb{Z}_2$ | $\checkmark$ |
| | | | $\pi$ | $\mathbb{Z}_2^2$ | $\mathbb{Z}_2$ | - | - |
| | | $p4$ | $0$ | $\mathbb{Z}_2$ | $\mathbb{Z}_2$ | $\mathbb{Z}_2$ | $\checkmark$ |
| | | | $\pi$ | $\mathbb{Z}_2$ | $\mathbb{Z}_2$ | - | - |
| | | $p6$ | $0$ | - | $\mathbb{Z}_2$ | $\mathbb{Z}_2$ | $\checkmark$ |
| | | | $\pi$ | - | $\mathbb{Z}_2$ | - | - |
| AIII, BDI, CII | $\mathbb{Z}$ | $p2$ | $0$ | - | - | - | $\checkmark$ |
| | | | $\pi$ | $\mathbb{Z}^2$ | $\mathbb{Z}_2$ | - | - |
| | | $p4$ | $0$ | - | - | - | $\checkmark$ |
| | | | $\pi$ | - | $\mathbb{Z}_2$ | - | - |
| | | $p6$ | $0$ | - | - | - | $\checkmark$ |
| | | | $\pi$ | - | $\mathbb{Z}_2$ | - | - |

this case, the rotation symmetry operators satisfy $U(R_{2\pi/n})^n = 1$ and are assumed to commute with time-reversal symmetry. We present the detailed results for each symmetry class of our classification in Appendix F.

Finally, we point out that our results for anomalies at disclinations respect the Abelian group structure of topological crystalline phases given by the direct sum [11] In particular, in symmetry classes for which the bulk-boundary-defect correspondence holds, the direct sum of a first-order topological phase with itself cannot lead to a second-order topological phase. In symmetry classes in which a disclination is connected to a domain wall, meaning that the bulk-boundary-defect classification does not hold, the direct sum of a first-order topological phase with itself may result in a second-order topological phase. The latter scenario is absent for all cases discussed here. Our classification results are consistent with corresponding results from Refs. [12, 67, 69].

---

[11]Note that the direct sum operation in the Abelian group structure of topological crystalline phases is applied to both the Hamiltonians $H_1 \oplus H_2$ and the representations of its symmetries $U_1(g) \oplus U_2(g)$. Here, the Hamiltonians $H_i$, $i = 1, 2$, are elements of the same symmetry class, with corresponding representation $U_i(g)$ of all symmetry elements $g \in G \times G_{\text{int}}$.

Table 2: Classification of three-dimenional $n$-fold rotation-symmetric weak, second-order, and first-order topological insulators and superconductors that give rise to one-dimensional anomalous defect states. We use the same notations as in Table 1. Here, $Q$ denotes the topological charge of the one-dimensional anomalous states in the given symmetry class. Since non-magnetic rotation symmetry preserves the propagation direction of chiral modes, there are neither weak nor higher-order phases in Cartan classes A, D, and C with non-magnetic rotation symmetry.

| Cartan class | $Q$ | $G$ | $\phi$ | Weak in $x$ / $y$ | Weak in $z$ | $2^{\text{nd}}$ order | $1^{\text{st}}$ order | BBDC |
|---|---|---|---|---|---|---|---|---|
| A | $\mathbb{Z}$ | $p2$ | | - | $\mathbb{Z}$ | - | - | ✓ |
| | | $p4$ | | - | $\mathbb{Z}$ | - | - | ✓ |
| | | $p6$ | | - | $\mathbb{Z}$ | - | - | ✓ |
| | | $p2'$ | | $\mathbb{Z}^2$ | - | $\mathbb{Z}_2$ | - | - |
| | | $p4'$ | | - | - | $\mathbb{Z}_2$ | - | - |
| | | $p6'$ | | - | - | $\mathbb{Z}_2$ | - | - |
| D | $\mathbb{Z}$ | $p2$ | $0$ | - | $\mathbb{Z}$ | - | - | ✓ |
| | | | $\pi$ | - | $2\mathbb{Z}$ | - | - | ✓ |
| | | $p4$ | $0, \pi$ | - | $\mathbb{Z}$ | - | - | ✓ |
| | | | $\frac{\pi}{2}, \frac{3\pi}{2}$ | - | $2\mathbb{Z}$ | - | - | ✓ |
| | | $p6$ | $0, \frac{2\pi}{3}, \frac{4\pi}{3}$ | - | $\mathbb{Z}$ | - | - | ✓ |
| | | | $\pi, \frac{\pi}{3}, \frac{5\pi}{3}$ | - | $2\mathbb{Z}$ | - | - | ✓ |
| | | $p2'$ | | $\mathbb{Z}^2$ | - | $\mathbb{Z}_2$ | - | - |
| | | $p4'$ | | - | - | $\mathbb{Z}_2$ | - | - |
| | | $p6'$ | | - | - | $\mathbb{Z}_2$ | - | - |
| DIII | $\mathbb{Z}_2$ | $p2$ | $0$ | $\mathbb{Z}_2^2$ | $\mathbb{Z}_2$ | $\mathbb{Z}_2$ | $\mathbb{Z}$ | ✓ |
| | | | $\pi$ | $\mathbb{Z}_2^2$ | - | $\mathbb{Z}_2$ | - | - |
| | | $p4$ | $0$ | $\mathbb{Z}_2$ | $\mathbb{Z}_2$ | $\mathbb{Z}_2$ | $\mathbb{Z}$ | ✓ |
| | | | $\pi$ | $\mathbb{Z}_2$ | - | $\mathbb{Z}_2$ | - | - |
| | | $p6$ | $0$ | - | $\mathbb{Z}_2$ | $\mathbb{Z}_2$ | $\mathbb{Z}$ | ✓ |
| | | | $\pi$ | - | - | $\mathbb{Z}_2$ | - | - |
| AII | $\mathbb{Z}_2$ | $p2$ | | $\mathbb{Z}_2^2$ | $\mathbb{Z}_2$ | $\mathbb{Z}_2$ | $\mathbb{Z}_2$ | ✓ |
| | | $p4$ | | $\mathbb{Z}_2$ | $\mathbb{Z}_2$ | $\mathbb{Z}_2$ | $\mathbb{Z}_2$ | ✓ |
| | | $p6$ | | - | $\mathbb{Z}_2$ | $\mathbb{Z}_2$ | $\mathbb{Z}_2$ | ✓ |
| C | $\mathbb{Z}$ | $p2$ | $0$ | - | $2\mathbb{Z}$ | - | - | ✓ |
| | | | $\pi$ | - | $2\mathbb{Z}$ | - | - | ✓ |
| | | $p4$ | $0, \pi$ | - | $2\mathbb{Z}$ | - | - | ✓ |
| | | | $\frac{\pi}{2}, \frac{3\pi}{2}$ | - | $2\mathbb{Z}$ | - | - | ✓ |
| | | $p6$ | $0, \frac{2\pi}{3}, \frac{4\pi}{3}$ | - | $2\mathbb{Z}$ | - | - | ✓ |
| | | | $\pi, \frac{\pi}{3}, \frac{5\pi}{3}$ | - | $2\mathbb{Z}$ | - | - | ✓ |
| | | $p2'$ | | $\mathbb{Z}^2$ | - | $\mathbb{Z}_2$ | - | - |
| | | $p4'$ | | - | - | $\mathbb{Z}_2$ | - | - |
| | | $p6'$ | | - | - | $\mathbb{Z}_2$ | - | - |

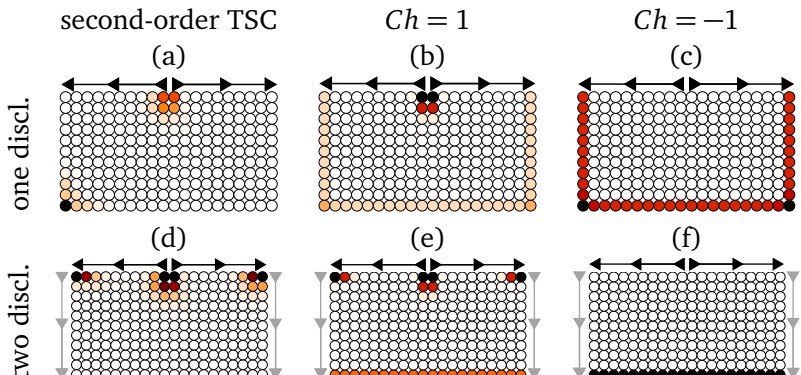

Figure 10: Model of a 2D second-order topological phase in class D protected by twofold rotation symmetry, as defined in Eq. (12), on a lattice with one and two disclinations: real-space weights for the wavefunctions of the two eigenstates with lowest absolute energy. Darker colors denote larger weights. The boundary conditions forming the disclinations are indicated by black and gray arrows: sites along corresponding lines of arrows are connected respecting the arrow orientation. The model is based on a two-layer stack of Chern superconductors with opposite Chern numbers $Ch = \pm 1$. The model parameters are $m = -1$ and $b_1 = b_2 = 0.4$. The first column shows the full model with both layers coupled. The second and third columns correspond to the individual, decoupled layers when we set $b_1 = b_2 = 0$. In (e), we plot four instead of two lowest eigenstates to also indicate the presence of the chiral edge modes.

# 6 Examples

Having laid out the general theory, we now turn to demonstrating our statements for specific models of second-order topological phases. In particular, we illustrate how to carry out the Volterra process explicitly and how anomalous disclination states arise in these phases. Throughout this section, we use $\tau_i, \rho_i, \eta_i, \sigma_i$, $i \in \{0, 1, 2, 3\}$, to denote Pauli matrices acting in different subspaces. For the computations and visualizations we have used the Python software package Kwant [73].

## 6.1 Class D in two dimensions

Cartan class D allows for topological phases in one and two dimensions. In one dimension, this phase corresponds to a Kitaev chain with zero-energy Majorana end states. In two dimensions, it is a superconductor with nontrivial Chern number and chiral Majorana edge states. We consider superconductors whose superconducting order parameter is even under rotation ($\phi = 0$ as in Table 1). In this case the representation of $n$-fold rotation symmetry satisfies $U(\mathcal{R}_{2\pi/n})^n = -1$ and commutes with particle-hole antisymmetry.

Using symmetry-based indicators, we derive in Appendix E an explicit expression for the number of Majorana bound states at a disclination in terms of the Chern number and rotation invariants. This specific result for class D is in agreement with previous literature [54]. Moreover, our method is applicable to other symmetry classes as well.

### 6.1.1 Twofold rotation symmetry

A model for a second-order topological superconductor protected by twofold rotation symmetry is

$$
\begin{aligned}
H_{\text{D},2}(\boldsymbol{k}) = \; & \tau_2\rho_3(m+2-\cos k_x - \cos k_y) \\
& + \tau_3\rho_3 \sin k_x + \tau_1\rho_3 \sin k_y + b_1\tau_1\rho_2 + b_2\tau_3\rho_2,
\end{aligned}
\tag{12}
$$

with particle-hole symmetry $\mathcal{P} = K$ in the Majorana basis and rotation symmetry $U(\mathcal{R}_\pi) = i\tau_2\rho_3$. The $4 \times 4$ matrices $\tau_i\rho_j$ are Kronecker products of Pauli matrices acting on the four sublattice degrees of freedom. For $-2 < m < 0$ and $b_1 = b_2 = 0$, this model corresponds to a stack of two Chern superconductors in the $\tau$ subspace with opposite Chern numbers hosting counterpropagating chiral edge modes. The terms proportional to $b_1$ and $b_2$ hybridize the two layers. The hybridization gaps the counterpropagating chiral edge modes such that a pair of Majorana bound states appears on corners related by twofold rotation.

Exact diagonalization of the Hamiltonian on a lattice with disclination shows that the model hosts one Majorana bound state at the disclination [see Fig. 10(a)]. This is in agreement with our general results from Eqs. (7) and (11) where, in this case, only the nontrivial second-order invariant contributes to the topological charge at the disclination. We observe that the Majorana bound state persists upon decoupling the layers by setting $b_1 = b_2 = 0$. As there is only a single Majorana at the disclination, our general results suggest that one of the two decoupled Chern superconductors must therefore bind a $\pi$-flux to the disclination. This is indeed the case as we confirm by investigating the hybridization across the branch cut in the two layers. According to Eq. (2), the nearest-neighbor hopping along the surface normal $\boldsymbol{n}$ is constructed from the bulk hopping $H_{\boldsymbol{r},\boldsymbol{r}+\boldsymbol{n}}$ in the direction of $\boldsymbol{n}$ as $H^{\text{cut}}_{\boldsymbol{r},\boldsymbol{r}+\boldsymbol{n}} = U(\mathcal{R}_\pi)H_{\boldsymbol{r},\boldsymbol{r}+\boldsymbol{n}}$. As the representation of twofold rotation differs by a $\pi$ phase between the subspaces of the two layers, the disclination binds a $\pi$-flux in only one of the layers. We confirm this picture by exactly diagonalizing the two layers in the decoupled limit separately, as shown in Figs. 10(b) and (c).

We also construct a lattice with two $\pi$ disclinations by connecting the surfaces as shown in Fig. 10(d). The exact diagonalization results confirm that the second-order topological phase hosts one anomalous state at each of the two disclinations. Moreover, recall that the total phase shift of a particle after transporting it around a $2\pi$ disclination is $2\pi$ (see Sec. 2). As a consequence, each of the two decoupled layers of Chern superconductors hosts an even number of Majorana bound states, zero or two in this case, distributed over the two disclinations [see Figs. 10(e) and (f)].

### 6.1.2 Fourfold rotation symmetry

To construct a model for a second-order topological superconductor protected by fourfold rotation symmetry [5, 74], we consider a square lattice in the $xy$ plane and place at each lattice site a Majorana mode $\gamma_i = \gamma_i^\dagger$. We add imaginary nearest-neighbor hopping between the Majorana modes, which we dimerize in the $x$ and in the $y$ direction. Finally, we thread a magnetic flux quantum through each lattice plaquette. The model is illustrated in Fig. 11(a). It can be viewed as an array of alternatingly coupled Kitaev chains in the presence of a gauge field. The Bloch Hamiltonian of the model reads

$$
\begin{aligned}
H_{\text{D},4}(\mathbf{k}) = \; & (t+\delta t)[\tau_3\rho_2 + \tau_2\rho_0] \\
& - (t-\delta t)[\cos(k_x)\tau_3\rho_2 + \sin(k_x)\tau_3\rho_1] \\
& - (t-\delta t)[\cos(k_y)\tau_2\rho_0 + \sin(k_y)\tau_1\rho_0],
\end{aligned}
\tag{13}
$$

with the real hopping parameters $t$ and $\delta t$. The model is written in the Majorana basis $\{\gamma_{a,\boldsymbol{k}}, \gamma_{b,\boldsymbol{k}}, \gamma_{c,\boldsymbol{k}}, \gamma_{d,\boldsymbol{k}}\}$, as indicated in Fig. 11(a). It satisfies particle-hole antisymmetry

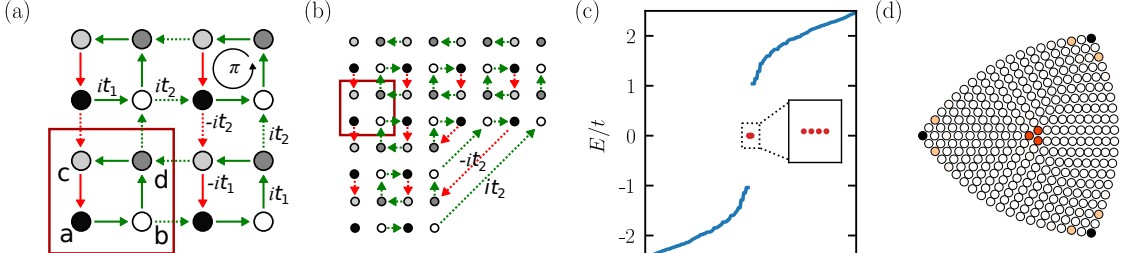

Figure 11: Model of a 2D second-order topological superconductor in class D protected by fourfold rotation symmetry as defined in Eq. (13): (a) depiction of the model Hamiltonian with $t_{1,2} = t \pm \delta t$. Majoranas hopping along red arrows pick up an additional $\pi$ phase giving rise to a total magnetic flux of $\phi_0$ per lattice plaquette. The red square denotes the unit cell of the model. (b) Illustration of how to connect sites across the cuts in the Volterra process for the fully dimerized limit ($t_1 = 0$), which leads to a $\pi/2$ disclination at the center of the lattice. Finally, on a lattice with $\pi/2$ disclination and model parameters $\delta t = -0.5t$ we show in (c) the spectrum close to $E = 0$ and in (d) the real-space weights of the zero modes. In (c), the four zero modes in the spectrum are highlighted in red.

$H^*(-\boldsymbol{k}) = -H(\boldsymbol{k})$ and fourfold rotation symmetry $U_{\frac{\pi}{2}} H(k_y, -k_x) U_{\frac{\pi}{2}}^\dagger = H(\boldsymbol{k})$. The representation of fourfold rotation $U_{\frac{\pi}{2}}$ describes a counterclockwise rotation of the Majorana particles in the unit cell including a sign change $\gamma_a \to -\gamma_c$.

For $\delta t < 0$, the model realizes a second-order topological superconductor with Majorana corner states. Figure 11(b) shows explicitly for the fully dimerized limit ($\delta t = -t$) how unit cells are connected across the cuts in the Volterra process to form a $\pi/2$ disclination. In this limit, it is apparent that unpaired Majorana bound states occur only at the three corners and at the disclination. Nevertheless, exact diagonalization of the model confirms that the Majorana bound states persist also away from the completely dimerized limit [see Fig. 11(c) and (d)]. This is in agreement with our general results from Eqs. (9) and (11) with only the second-order invariant contributing to the topological charge at the disclination.

### 6.1.3 Sixfold rotation symmetry

A model Hamiltonian for a second-order topological superconductor protected by sixfold rotation symmetry can similarly be constructed from a sixfold-symmetric arrangement of Majorana fermions in the unit cell [54]. Our model is composed of two hybridization patterns thereby interpolating between trivial and topological phase,

$$H_{\mathrm{D},6} = t_1 H_{\mathrm{triv}} + t_2 H_{\mathrm{topo}}. \tag{14}$$

$H_{\mathrm{triv}}$ describes the trivial phase [see Fig. 12(a)] as all Majorana fermions are recombined within the unit cell. $H_{\mathrm{topo}}$, on the other hand, corresponds to a second-order topological phase [see Fig. 12(b)] where Majorana bound states hybridize across unit cells. A Bloch Hamiltonian for the topological phase is given in Ref. [54].

In the topological phase, the model features a Majorana zero mode bound to a $\pi/3$ disclination, as illustrated in Figs. 12(c) and (d), in accordance with the general result for the topological charge at a disclination presented in Eq. (10). Moreover, in Fig. 13 we explicitly show that a $2\pi/3$ disclination does not give rise to anomalous disclination states.

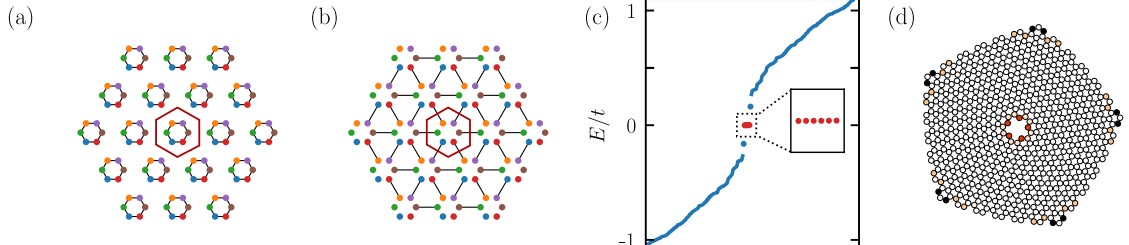

Figure 12: Model of a 2D second-order topological superconductor in class D protected by sixfold rotation symmetry: (a), (b) depiction of the model in the two fully dimerized limits, where (a) describes the trivial phase $H_{\text{triv}}$ and (b) the second-order topological phase $H_{\text{topo}}$. The unit cell (red hexagon) consists of six Majorana fermions depicted by different colors. For a lattice with $\pi/3$ disclination and model parameters $t_1 = 1.5$ and $t_2 = 0.5$, we show in (c) the spectrum close to $E = 0$ and in (d) the real-space weights of the six zero modes. In our model, there is another pair of in-gap modes localized at the disclination. These modes are not anomalous and can be pushed into the bulk spectrum by a local deformation at the disclination.

## 6.2 Class DIII in two dimensions

Cartan class DIII describes time-reversal symmetric superconductors in the absence of spin-rotation symmetry. As before, we assume a pairing symmetry of the form $U(\mathcal{R})\Delta(\mathcal{R}\boldsymbol{k})U^{\dagger}(\mathcal{R}) = \Delta(\boldsymbol{k})$. We can straight-forwardly construct a class-DIII second-order topological superconductor from a corresponding model $H_D(\mathbf{k})$ in class D by augmenting it with its time-reversed copy $H_D^*(-\mathbf{k})$. We do this explicitly for the fourfold-symmetric model defined in Eq. (13) Note that models for the other rotation symmetries can be constructed in the same way.

The resulting model $H_D(\mathbf{k}) \oplus H_D^*(-\mathbf{k})$ is time-reversal symmetric with the anitunitary operator $\mathcal{T} = i\tau_0\rho_0\sigma_2 K$ and $\mathbf{k} \to -\mathbf{k}$, where $\sigma_i$ are Pauli matrices acting on the pseudo-spin degree of freedom. The model is particle-hole symmetric with the antiunitary operator $\mathcal{P} = K$, $\mathbf{k} \to -\mathbf{k}$. The representation of fourfold rotation symmetry is $\tilde{R}_{\frac{\pi}{2}} = U_{\frac{\pi}{2}}\sigma_0$, $(k_x, k_y) \to (k_y, -k_x)$, where $U_{\frac{\pi}{2}}$ is the same as for the class-D model. In our model we allow for symmetry-preserving nearest-neighbour coupling between the pseudo-spins. The Bloch Hamiltonian takes the form,

$$H_{\text{DIII},4}(\mathbf{k}) = \begin{pmatrix} H_{D,4}(k_x, k_y) & \lambda A(k_x, k_y) \\ \lambda A(k_x, k_y) & H_{D,4}^*(-k_x, -k_y) \end{pmatrix}, \tag{15}$$

with the term

$$\begin{aligned} A(\mathbf{k}) &= [1 - \cos(k_x)]\tau_3\rho_2 - \sin(k_x)\tau_3\rho_1 \\ &\quad + [1 - \cos(k_y)]\tau_2\rho_0 - \sin(k_y)\tau_1\rho_0, \end{aligned}$$

coupling the two pseudo-spin subblocks.

In the second-order topological phase, this system features one Majorana Kramers pair at each of the four corners of a square-shaped sample. After carrying out the Volterra process, the resulting $\pi/2$ disclination binds one Majorana Kramers pair, as illustrated in Fig. 14. This again confirms our general results from Eqs. (9) and (11).

## 6.3 Class DIII in three dimensions

In Cartan class DIII, second-order topological superconductors exist both in two and three dimensions. In three dimensions, those come in two different variants: First, a strong second-order topological phase is constructed from a corresponding two-dimensional model in class

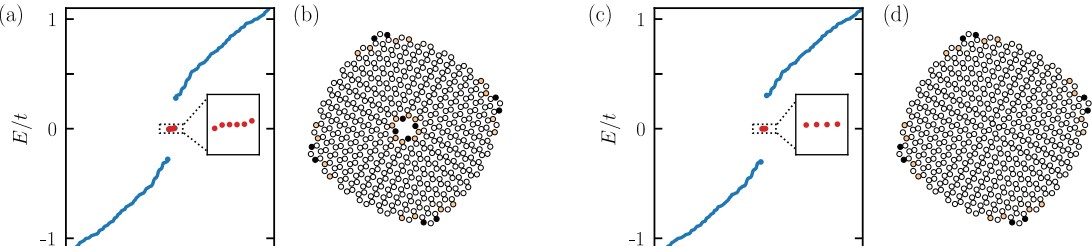

Figure 13: 2D second-order class D model from Eq. (14) with a $2\pi/3$ disclination: model parameters $t_1 = 1.5$ and $t_2 = 0.5$. (a) shows the spectrum close to $E = 0$ for a disclination for which the central sites have been removed [similar to Fig. 12(c) and (d)]. We find four zero modes localized to the four corners of the lattice and two finite-energy mid-gap states at the disclination (see inset). In (b), we visualize the real-space weights of these six modes. The two finite-energy modes are not anomalous and can be pushed into the bulk spectrum by a local deformation at the disclination. This is explicitly demonstrated in (c) and (d) by recovering the four central sites at the disclination.

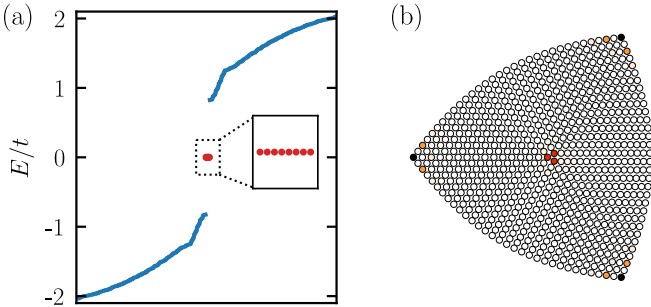

Figure 14: Model of a 2D second-order topological superconductor in class DIII on a square lattice with disclination as defined in Eq. (15): the model parameters are $\delta t = -0.5t$, $\lambda = 0.5t$. (a) Low-energy spectrum with four Kramers pairs of Majorana zero modes highlighted in red. (b) Real-space weights of the Majorana zero modes.

D by using the dimensional raising map [47,75,76]. Alternatively, since class DIII already features second-order topological phases in 2D, these nontrivial class-DIII models can be stacked along the rotation axis to produce a weak second-order topological superconductor. In the following, we demonstrate these two cases explicitly for fourfold symmetric systems, but the same procedures are applicable also to models with other rotation symmetries.

### 6.3.1 Strong second-order topological superconductor

Applying the dimensional raising map to the Hamiltonian $H_{D,4}(\mathbf{k})$ from Eq. (13) (see Appendix F.3.3) results in the following Bloch Hamiltonian,

$$
\begin{aligned}
H_{\text{DIII}}^{\text{st}}(\mathbf{k}) &= H_{D,4}[k_x, k_y; \delta t \to \delta t \cos(k_z)]\sigma_3 \\
&\quad + \sin(k_z)\tau_0\rho_0\sigma_1,
\end{aligned} \tag{16}
$$

where we replace $\delta t$ in the definition of $H_{D,4}(\mathbf{k})$ by $\delta t \cos(k_z)$. The model is defined on a cubic lattice and has particle-hole, time-reversal, and fourfold rotation symmetry around an axis parallel to the $z$ axis. The respective operators have the same structure as for the two-dimensional model in class DIII discussed at the end of the previous section: $\mathcal{P} = K$, $\mathcal{T} = i\tau_0\rho_0\sigma_2 K$, and $\tilde{R}_{\frac{\pi}{2}} = U_{\frac{\pi}{2}}\sigma_0$ with the $C_4$ operator $U_{\frac{\pi}{2}}$ of the underlying model in class D.

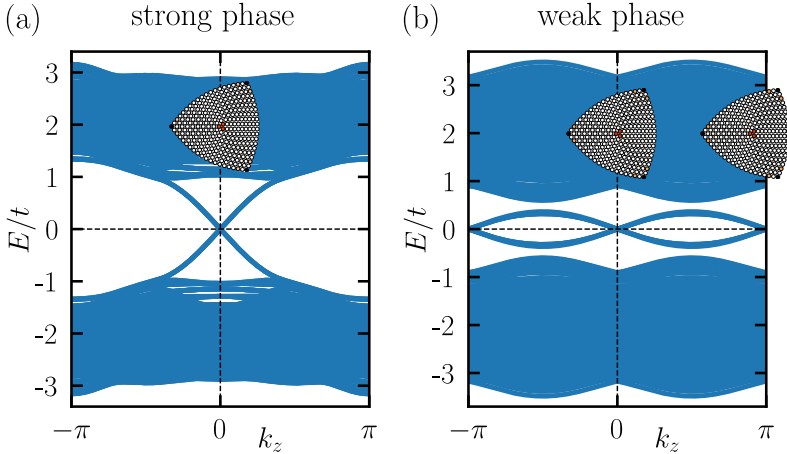

Figure 15: Bandstructures of 3D second-order topological superconductors in class DIII protected by fourfold symmetry: (a) strong topological phase and (b) weak topological phase, with model Hamiltonians as defined in Eqs. (16) and (17), respectively. The corresponding models are realized in a pillar geometry with a line disclination parallel to the $z$ axis. The insets show the real-space weights of the zero-energy states at the respective momenta.

We consider this model in a pillar geometry, infinite along the $z$ direction and having a finite, square-shaped cross section with open boundary conditions in the $x$ and $y$ directions. We carry out a Volterra process, as depicted in Fig. 4(c), resulting in an infinitely long $\pi/2$ line disclination parallel to the $z$ axis. The bandstructure shows that the disclination binds one helical Majorana mode. We observe the same features at the three hinges of the lattice [see Fig. 15(a)]. This strong topological phase is robust against translation symmetry breaking.

### 6.3.2 Weak second-order topological superconductor

A stack of the two-dimensional systems $H_{\mathrm{DIII}}(k_x, k_y)$ as defined in Eq. (15) is described by a Hamiltonian of the form

$$H_{\mathrm{DIII}}^{\mathrm{w}}(k_x, k_y, k_z) = H_{\mathrm{DIII}}(k_x, k_y) + t_z \sin(k_z)\tau_0 \rho_0 \sigma_z, \tag{17}$$

where we have included a symmetry-allowed hybridization between adjacent layers proportional to $t_z$. The symmetries of this system and their representations are identical to the two-dimensional class-DIII model.

We consider this model in the same pillar geometry as for the strong second-order phase above and apply the Volterra process accordingly. We show the bandstructure of the lattice with disclination in Fig. 15(b). For each of the three hinges and for the disclination line, there is a pair of particle-hole symmetric bands within the bulk energy gap. Most notably, these bands are detached from the bulk continuum but pinned to the momenta $k_z = 0$ and $\pi$ at $E = 0$ by symmetry. The hinge and disclination spectra correspond to chains of Majorana Kramers pairs. This is a weak topological phase because it can be trivialized by breaking translation symmetry such that states at $k = 0$ and $\pi$ hybridize.

### 6.4 Class AII in three dimensions.

A class-AII model can formally be constructed from a Hamiltonian $H_{\mathrm{DIII}}^{\mathrm{st}}$ in class DIII [see Eq. (16)] by breaking its particle-hole antisymmetry while preserving time-reversal and ro-

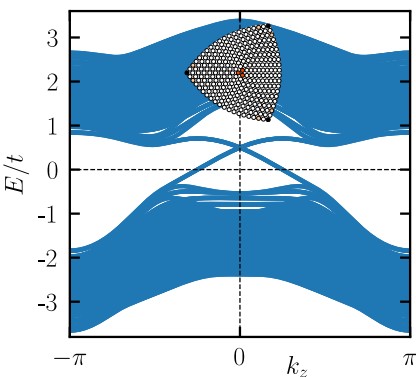

Figure 16: Bandstructure of a 3D second-order topological insulator in class AII protected by fourfold symmetry as defined in Eq. (18): the model is realized in a pillar geometry with a line disclination parallel to the $z$ axis. The inset shows the real-space weight of the electronic in-gap states at $k_z = 0$.

tation symmetry. In addition, this requires that the corresponding single-particle Hamiltonian is defined in terms of electronic operators $d^\dagger_{\mathbf{k}a\sigma}$ instead of Majorana operators $\gamma_{\mathbf{k}a\sigma}$, where $a$ and $\sigma$ denote the sublattice and pseudo-spin degrees of freedom, respectively.

With the above transformation, a suitable model in class AII is given by

$$H_{\mathrm{AII}}(\mathbf{k}) = H^{\mathrm{st}}_{\mathrm{DIII}}(\mathbf{k}) + t_b \cos(k_z)\tau_0\rho_0\sigma_0\,, \tag{18}$$

where the term proportional to $t_b$ breaks particle-hole antisymmetry. The symmetry operators are $\mathcal{T} = i\tau_0\rho_0\sigma_2 K$ with $\mathcal{T}^2 = -1$, and $\tilde{R}_{\frac{\pi}{2}} = U_{\frac{\pi}{2}}\sigma_0$ with $(\tilde{U}_{\frac{\pi}{2}})^4 = -1$.

The spectrum for the same pillar geometry with disclination as above reveals the presence of helical electronic states running along the line disclination (see Fig. 16). The energy bands corresponding to the helical states traverse the bulk-energy gap. As opposed to the model in class DIII, the crossing point of the helical bands is no longer at zero energy due to the absence of particle-hole antisymmetry.

We point out that models for second-order topological insulators in class AII protected by twofold or sixfold rotation symmetries can be constructed in the same way.

# 7   Conclusion

*Results.* By combining an exhaustive holonomy classification of lattice defects in rotation symmetric two- and three-dimensional lattices with an exhaustive classification of topological phases of free fermions in such lattices, we have determined the precise relation between bulk topology, and boundary and defect anomaly. This relation is captured by Eqs. (7) to (11). Our result shows that topological phases protected by a crystalline symmetry contribute anomalous states only at lattice defects that carry a holonomy of the protecting crystalline symmetry. In particular, second-order topological phases contribute to the anomaly of disclinations, and weak topological phases contribute to the anomaly at dislocations and disclinations with non-trivial translation holonomy. Tenfold-way first-order topological phases contribute anomalous states only at defects that bind a $\pi$-flux, such as a superconducting vortex. These results apply both to individual defects as well as to the total anomaly of a collection of defects determined from the holonomy of a loop around the collection.

These results formalize the concept of *bulk-boundary-defect-correspondence* for rotation and translation symmetric topological phases of free fermions: the topological protection of

anomalous states at a translation or rotation symmetric boundary and at a defect that carries a translation or rotation holonomy is tied to the same bulk crystalline topological phase, whose bulk topology in turn is protected by the respective crystalline symmetry.

Furthermore, we have identified the set of symmetry classes in which the disclination as constructed from the Volterra process is the edge of a domain wall. The domain wall is present in the absence of a unitary rotation symmetry that commutes with all internal unitary symmetries and antisymmetries of the system's Hamiltonian. In case the disclination is connected to a domain wall, the anomaly at the disclination depends on the microscopic properties of the domain wall. Therefore, in these symmetry classes the presence of the domain wall prohibits a unique determination of the disclination anomaly from the bulk topology. We note that the domain wall may become locally unobservable if the disclination breaks some internal symmetries otherwise present in the bulk, or if the disclination involves a translation holonomy by a fractional lattice vector.

Finally, we have determined for all Cartan classes of free fermions with a physically meaningful representation of rotation symmetry in two and three dimensions: (i) all possible contributions to $(d-2)$-dimensional anomalous states at a disclination, dislocation or geometric $\pi$ flux defect and (ii) whether a domain wall emerges at a disclination in the given symmetry class. These results are summarized in tables 1 and 2. We ensure to capture all contributions to (i) by comparing to a complete classification of first and second-order strong topological phases and the relevant weak topological phases. We have confirmed our construction for a few physical examples of superconductors with and without time-reversal symmetry in Cartan classes D and DIII and for a time-reversal symmetric insulator in Cartan class AII.

*Discussion.* Due to the large elastic stress associated with a disclination in ordered media, disclinations typically appear in pairs with canceling Frank angle or at the edges of grain boundaries [35]. More specifically, in the disclination model of grain boundaries, they come in bound pairs [40, 44]. Therefore, anomalous disclination states along a grain boundary may gap out pairwise through hybridization. On the contrary, *isolated* disclinations may be realized at the center of nanowires.

Possible platforms for the study of anomalous disclination states are SnTe and antiperovskite materials. These materials classes have been put forward as candidates for second-order topological insulators protected by rotation symmetry [4, 15]. Our findings suggest that disclinations in these materials may bind helical modes. Moreover, SnTe nanowires with a pentagonal crossection have been succesfully fabricated [77]. Their unusual shape hints at the presence of an isolated disclination at their core, rendering them a promising experimental platform for the study of anomalous disclination states.

On the other hand, second-order topological superconductors protected by rotation symmetry may be realized in the superconducting phases of certain topological crystalline insulators [26] or in iron-based superconductors [78]. In these materials, disclinations may bind helical Majorana modes.

Disclinations can also appear in mesomorphic phases [35, 79–81]. Our arguments of Sec. 3 show that the bulk-boundary-disclination correspondence also holds for these partially ordered phases. Furthermore, a bulk-disclination correspondence has been observed in photonic crystals [82].

Going beyond our results in this work, we conjecture that relations similar to Eqs. (7)–(10), relating the topological charge at a defect to the higher-order topology of the bulk also exist for crystalline symmetries other than rotations. Topological lattice defects can be defined for all space-group symmetries and it has been shown that crystalline topological phases generally exhibit a topological response with respect to a corresponding topological lattice defect [36]. Hence, we expect that higher-order topological phases host anomalous states at topological

lattice defects whose holonomy corresponds to an action of the protecting crystalline symmetry.

Our results have extended the higher-order bulk-boundary correspondence of topological crystalline phases to disclinations in rotation-symmetric systems. We have thereby established a link to the topological response theory for defects familiar from the study of interacting symmetry-protected topological phases [36, 83].

All files and data used in this study are available in the repository at Ref. [84].

# Acknowledgements

We thank Andreas Bauer, Piet Brouwer, Luka Trifunovic, Piotr Dziawa and Jakub Polaczyński for enlightening discussions.

**Funding information**   MG acknowledges support by project A03 of the CRC-TR 183 "Entangled States of Matter" and by the European Research Council (ERC) under the European Union's Horizon 2020 research and innovation program under grant agreement No. 856526. ICF was supported by the Deutsche Forschungsgemeinschaft (DFG, German Research Foundation) under Germany's Excellence Strategy through the Würzburg-Dresden Cluster of Excellence on Complexity and Topology in Quantum Matter – *ct.qmat* (EXC 2147, project-id 390858490). AL was supported by the International Centre for Interfacing Magnetism and Superconductivity with Topological Matter project, carried out within the International Research Agendas program of the Foundation for Polish Science co-financed by the European Union under the European Regional Development Fund.

**Author contributions.**   ICF initiated the project with the idea of considering the Volterra process in 2d and 3d second order topological phases. AL supervised the project. MG performed the topological crystal calculations and obtained the classifications presented in our tables. AL and MG performed the numerical simulations. MG and AL produced the figures of the manuscript. All authors contributed in developing and understanding the results and in writing the manuscript.

# Appendix

The Appendix is organized as follows. In Appendix A, we review in more detail how to construct equivalence classes of holonomies for disclinations. Appendix B contains a detailed discussion on the occurrence of domain walls bound to disclinations in certain certain symmetry classes. In Appendix C, we provide supplementary information on the derivation of the topological crystal construction and on the presence of the domain wall as an obstruction to a symmetric decoration in the topological crystal construction. In Appendix D, we present an argument that the contribution of first-order topological phases to the number of anomalous states at disclinations is independent of their rotation holonomy, but only depends on the presence of $\pi$-fluxes for tenfold-way topological phases. Appendix E contains an example on how to apply symmetry-based indicators to determine the presence of anomalous states at a disclination. Finally, Appendix F presents the details on how to derive the classification of anomalous disclination states in all symmetry classes in two and three dimensions as summarized in Tables 1 and 2 of the main text.

# A  Holonomy equivalence classes of disclinations

We briefly review the construction of the equivalence classes of holonomies $\text{Hol}(\Omega)$ of disclinations with Frank angle $\Omega$ as presented in Ref. [50]. The lattice density $\rho(r) = \rho(r + t)$ breaks the Galilean symmetry group $G_{\text{Galilean}} = SO(2) \ltimes \mathbb{R}^2$ in two dimensions into a discrete space group of the lattice $G_{\text{lat}}$ with lattice vector $t$. These groups are constructed from the semidirect product $\ltimes$ of rotations $r(\phi) \in SO(2)$ and translations $t$ because these elements in general do not commute. Disclinations can be described in the simplest case where $G_{\text{lat}} = C_n \ltimes T$ is symmorphic and is generated by $n$-fold rotations $r(2\pi/n)$ and translations $T \simeq \mathbb{Z}^2$. Therefore, the lattice density is an element of the order parameter space $G_{\text{Galilean}}/G_{\text{lat}} = SO(2) \ltimes \mathbb{R}^2/C_n \ltimes T$. Point defects in this order parameter space are described by the fundamental group $\pi_1(G_{\text{Galilean}}/G_{\text{lat}}) \simeq \frac{2\pi}{n}\mathbb{Z} \ltimes T$ whose elements $(\Omega, t)$ describe the rotation $\Omega$ and translation holonomy $t$ of the defect. Because translations and rotations in $G$ do not commute, for point defects with $\Omega \neq 0$ (*i.e.* disclinations), the holonomy of a loop depends on the initial point and orientation of the coordinate system that is transported around the system. As the classification should not depend on this arbitrary choice, one takes the equivalence relation over all initial conditions obtained by a translation or rotation of the initial coordinate system. As a result, each lattice defect is characterized by an element $(\Omega, [t])$ where $[t]$ are the equivalent translation holonomies obtained by a variation of the initial condition. The number of inequivalent translation holonomies depends on the Frank angle $\Omega$ of the defect. The resulting classification $\text{Hol}(\Omega)$ of disclinations with Frank angle $\Omega$ distinguished by their inequivalent translation holonomies $t$ is summarized in Eq. (1) of the main text. A detailed example on how the equivalence classes are computed can be found in the Supplementary Material of Ref. [66].

# B  Symmetry classes hosting disclinations binding domain walls

We argued in Sec. 2.5 of the main text that in certain symmetry classes it is impossible to apply the bulk hopping across the cut during the Volterra process without breaking some internal symmetries. In Sec. B.1 below, we illustrate that for these symmetry classes the Volterra process leads to a domain wall bound to the disclination. The domain wall separates regions that are distinguishable by the local arrangement of orbitals in the unit cell. In particular, the presence of a domain wall implies that there is no unique hybridization pattern across the cut line. As a consequence, the topological charge at the disclination is not uniquely determined from rotation and translation holonomies and bulk topological invariants alone. However, we show in Sec. B.2 that the parity of the topological charge along the cut line can be related to the bulk topology. These facts are illustrated with an example of a magnetic topological insulator in Sec. B.3. Furthermore, we argue in Sec. B.4 that for all symmetry classes with $d-2$ dimensional anomalous states with $\mathbb{Z}$ topological charge and ($n \in 2, 3, 4, 6$)-fold rotation symmetry the following holds: either (i) no strong second-order or weak topological phase with topological crystal limit as shown in Fig. 6 of the main text exists, or (ii) their symmetry group does not contain a unitary rotation symmetry, i.e., it contains only an antiunitary rotation symmetry or a rotation antisymmetry. This provides a *no-go theorem* for a unique correspondence between strong bulk topology and $(d-2)$ dimensional disclination anomaly in these symmetry classes.

## B.1  Domain wall interpretation

In Sec. 2.5 of the main text, we showed that in certain symmetry classes the disclination necessarily connects to a line along which some symmetries are broken. These are precisely those symmetry classes that either do not contain a unitary rotation symmetry (for instance in

**SciPost**        SciPost Phys. 10, 092 (2021)

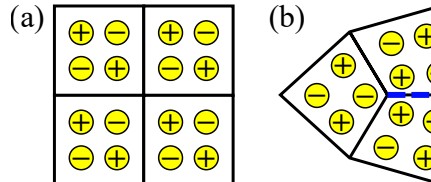

Figure B.1: An internal unitary symmetry or antisymmetry $\mathcal{U}$ with $\mathcal{U}^2 = 1$ allows to label the physical degrees of freedom in the unit cell, indicated by the signs $\pm$ on the orbitals depicted by yellow circles. In (a), we show a fourfold rotation symmetric lattice in which the representation of fourfold rotation symmetry has to anticommute with the internal unitary symmetry or antisymmetry because the rotation permutes the labels. In (b), we show a disclination that is connected to a domain wall (blue dashed line) between two regions related by a permutation of the labels.

magnetic space groups) or whose unitary rotation symmetry anticommutes with some internal unitary symmetries or antisymmetries. Here, we show that in these cases one can define labels or an order parameter that allow to distinguish the two regions bordered by the line. This shows that the line is in fact a domain wall.

Internal unitary symmetries or antisymmetries provide labels for the physical degrees of freedom on the lattice. These labels are defined in the diagonal basis of the internal unitary symmetry/antisymmetry. The representation of unitary rotation symmetry describes the action on the physical degrees of freedom that needs to be performed such that the system is invariant under the rotation. In case the representation of unitary rotation symmetry anticommutes with an internal unitary symmetry or antisymmetry, the rotation symmetry permutes the labelled degrees of freedom, as illustrated for a fourfold rotation symmetric sample in Fig. B.1(a). Thus, two patches that are rotated with respected to each other without applying the representation of internal symmetry are distinguishable by their configuration of labels. In this case, a disclination is therefore the end of a *domain wall* bordering two regions with permuted labelling, see Fig. B.1(b).

For magnetic space groups, where only the product of rotation and time reversal is preserved but not individually, one can define a vectorial order parameter, such as a local magnetization, that is odd under time reversal. A disclination with Frank angle corresponding to the magnetic rotation symmetry is thus the end point of a domain wall separating regions that are related by time-reversal symmetry. This is further illustrated with an example in Sec. B.3 below.

In the presence of translation symmetry, the local order parameter can be expressed in terms of the labels under the unitary internal symmetry or antisymmetry $\mathcal{U}^2 = \pm 1$ (or the local magnetization for magnetic rotation symmetry) in the asymmetric unit within the unit cell, see Fig. 5 in the main text. The unit cell then consists of asymmetric sections with labels related by rotation symmetry. In case rotation symmetry anticommutes with the unitary internal symmetry or antisymmetry (in case of magnetic rotation symmetry), the labels (local magnetization) in symmetry related asymmetric sections are opposite. Here, despite rotations, a translation by half a lattice vector may interchange the labels. Therefore, the domain wall may become locally unobservable if the disclination contains a translation holonomy by a fraction of a lattice vector. Note that in this case, the sample does not allow for a global and consistent definition of the unit cell. Throughout the paper, we restrict ourselves to lattice defects with a translation holonomy that is an integer multiple of the lattice vectors. These lattice defects can be constructed and analyzed with the methods from Sec. 2.2 in the main text. By construction, the lattice with the defect contains a consistent definition of the unit cell. The interplay of topological phases and screw dislocations with fractional translation holonomy has been

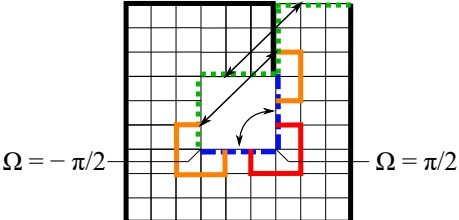

Figure B.2: A disclination dipole consisting of two disclinations with opposite Frank angle $\Omega = \pm\pi/2$ can be constructed in a fourfold rotation symmetric lattice by removing a square from the sample and connecting the boundaries as indicated by the blue dashed and green dotted lines. The blue dashed lines indicate boundary conditions that involve a rotation by $\pi/2$. The green dotted lines indicate boundary conditions that do not involve any rotation. A path around the disclination with Frank angle $\Omega = \pi/2$ ($\Omega = -\pi/2$) is indicated by the red (orange) bold line. For clarity, the sample boundary is highlighted by the black bold line.

investigated in Ref. [85].

Furthermore, we point out that if the sample with disclination as a whole does not respect all internal symmetries of the bulk system, then the domain wall may become unobservable. In particular, this may be the case when constructing nearest-neighbor lattice models with an artificial 'sublattice' antisymmetry $\Gamma^2 = 1$ which indicates the absence of hopping terms between different sublattices and the equality of the chemical potential on both sublattices. In case $n$-fold rotation symmetry anticommutes with the sublattice antisymmetry, the sublattices are interchanged at adjacent unit cells across the branch cut attached to a disclination with Frank angle $\Omega$ equal to an odd integer multiple of $2\pi/n$. If the sublattices are indistinguishable up to their label, then applying the same nearest-neighbor hopping across the branch cut yields a branch cut indistinguishable from the bulk lattice. This breaks the sublattice antisymmetry along the branch cut as sites with same sublattice label are connected by a hopping term. As a consequence, also the disclination breaks the sublattice antisymmetry and a potentially bound state may acquire a finite energy and is not anomalous. For consistency, we assume throughout the paper that the sample with disclination as a whole respects the internal symmetries of the bulk system.

## B.2 Topological charge at the domain wall and a disclination dipole

In case the cut line forms a domain wall and connects to the boundary, the intersection of the line with the boundary forms another point defect. Then, the parity of the total topological charge along the domain wall is determined by identical expressions as in Eqs. (7) to (11) in the main text. This is because of the anomaly cancellation criterion: anomalous boundary states associated to the domain wall can only be created pairwise.

However, in case of a disclination dipole, the domain wall may connect between the two partners of the dipole. A disclination dipole consists of two disclinations with opposite Frank angle that are connected by a cut line, as depicted in Fig. B.2. In case none of the two disclinations involves a translation holonomy, the topological charge at the pair cancels. Thus, the topological charge at the disclinations of the dipole cannot be predicted from the bulk topology if the cut line forms a domain wall.

## B.3 Example: Magnetic topological insulator

A magnetic topological insulator breaks time-reversal symmetry, but preserves the product of time-reversal symmetry and a crystalline symmetry operation. In the following, we consider

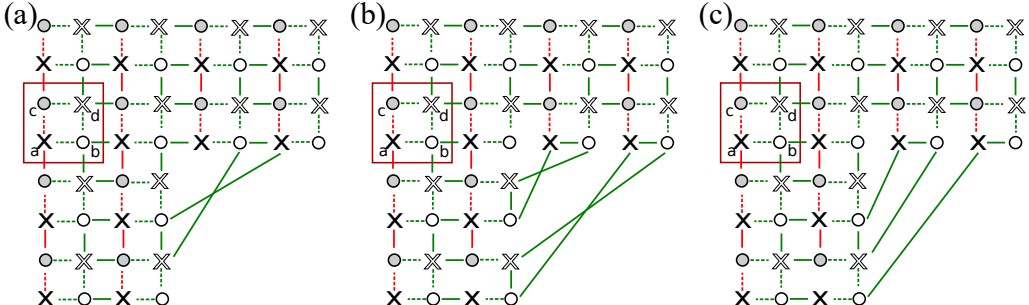

Figure B.3: Depiction of the model Hamiltonian for the magnetic second-order topological insulator defined in Eq. (20). Crosses and dots denote chiral modes with opposite chirality as defined from an expansion of the Hamiltonian around $k_z = 0$ and $k_z = \pi$. Full (dashed) lines denote the hybridization at $k_z = 0$ ($k_z = \pi$). Red lines include a $\pi$ phase. The unit cell is highlighted by a red square. In (a)–(c), we show three different possibilities of connecting the hybridizations at $k_z = 0$ across the cut in the Volterra process to construct a $\frac{\pi}{2}$ disclination.

a three-dimensional magnetic topological insulator where the product of fourfold rotation and time-reversal symmetry is preserved. In this case, there is only a strong second-order phase. Weak phases corresponding to arrangements of Chern insulators parallel to the rotation axis are forbidden as the copropagating chiral modes at twofold rotation symmetric momenta in this decoration cannot gap out. Furthermore Chern insulators are not compatible with a perpendicular magnetic rotation axis.

A model for a second-order topological insulator with magnetic fourfold rotation symmetry can be constructed similar to the class D model in Eq. (13) of the main text. The Hamiltonian of our model is given by

$$
\begin{aligned}
H_A(\boldsymbol{k}) = \ & t(1 - \cos k_z)[\tau_3\sigma_2 - \tau_2\sigma_0] \\
& - t(1 + \cos k_z)[\cos(k_x)\tau_3\sigma_2 + \sin(k_x)\tau_3\sigma_1] \\
& + t(1 + \cos k_z)[\cos(k_y)\tau_2\sigma_0 - \sin(k_y)\tau_1\sigma_0] \\
& + t_z \sin k_z \tau_3\sigma_3.
\end{aligned}
\tag{19}
$$

As in the previous section, the hopping parameter $t$ is real, $\tau_i\sigma_j$ are four-by-four matrices composed of Pauli matrices acting on the four degrees of freedom in the unit cell. Here, our basis consists of four fermionic operators $c_{\boldsymbol{k},a}, c_{\boldsymbol{k},b}, c_{\boldsymbol{k},c}, c_{\boldsymbol{k},d}$ arranged as depicted in Fig. B.3(a). Magnetic fourfold rotation exchanges the fermions in the unit cell counter-clockwise including a phase $c_{\boldsymbol{k},a} \to -c_{\boldsymbol{k},c}$ and a time-reversal operation implemented by complex conjugation. This model can be adiabatically deformed to the chiral higher-order topological insulator of Ref. [4].

To understand the topology of this model, consider an expansion in small $k_z$ around $k_z = 0$ and $k_z = \pi$ separately. Both cases around $k_z = 0$ and $k_z = \pi$ correspond to a lattice of chiral modes whose chirality is determined by the term $t_z \sin k_z \tau_3\sigma_3$. Around $k_z = 0$ and for positive $t_z$ the "a" and "d" ("b" and "c") lattice sites have positive (negative) chirality. The chiral modes are hybridized across unit cells such that there is a chiral mode remaining at each corner, as depicted by the dashed lines in Fig. B.3. Around the $k_z = \pi$ plane the chiralities are reversed and all chiral modes are hybridized within the unit cells such that no corner modes are remaining, as depicted by the full lines in Fig. B.3. The Hamiltonian interpolates between the two hybridizations and remains gapped for every $k_z$. As a consequence, the model realizes chiral hinge states in agreement with fourfold magnetic rotation symmetry.

In the Volterra process we need to determine the fate of the hybridizations of the chiral modes around $k_z = 0$ across the cut. As depicted in Fig. B.3 there are two choices realizing

completely dimerized limits along the cut line. Both limits require a breaking of fourfold magnetic rotation symmetry along the cut line, in agreement with our results from Sec. 4 of the main text. The configuration in Fig. B.3(a) can be realized without closing the excitation gap along the cut line. It realizes a chiral disclination mode propagating into the plane and two chiral modes propagating out of the plane at the end of the cut line. In order to obtain the configuration in Fig. B.3(b) one needs to change the hybridization pattern along both cut lines which requires a closure of the excitation gap along these lines. This configuration has a disclination mode propagating out of the plane and gapped boundaries. Finally, Fig. B.3(c) depicts a hybridization pattern where the excitation gap needs to close only on one of the two cut lines. This pattern hosts no anomalous disclination state, but a single chiral mode at the end of the cut line.

**Domain wall interpretation**

For an antiferromagnetic insulator with magnetic space group $p4'$, one can define a vectorial order parameter as the magnetization within a quarter of the unit cell. Thus, with a consistent definition of a unit cell as provided by the Volterra process in Sec. 2 of the main text, the order parameter distinguishes regions that are rotated by $\pi/2$ with respect to each other. A disclination with Frank angle $\Omega = \pi/2$ is thus the edge of a domain wall.

### B.4 No-go theorem in symmetry classes whose $d-2$ dimensional anomalous states have $\mathbb{Z}$ topological charge

The purpose of this section is to show that for symmetry classes whose $d-2$ dimensional anomalous states have $\mathbb{Z}$ topological charge, either (i) no strong second-order topological phase exists, or (ii) the symmetry group does not contain a unitary rotation symmetry that commutes with all internal unitary symmetries or antisymmetries. We presented an argument that shows the correctness of this statement in two and three dimensions in Sec. 3.4 of the main text. Here, we show that the argument can be generalized to any dimension $d \geq 2$ using the dimensional raising map [47,75,76].

The dimensional raising map provides an isomorphism between the classifying groups of the strong topological phases in different symmetry classes and dimensions. It has been extended to be applied in the presence of crystalline symmetries [75,76]. Here, we apply the dimensional raising map such that rotation symmetry acts trivially in the added dimensions. Below we are going to review how the Hamiltonians and symmetry operators are mapped under the dimensional raising maps following Ref. [76]. We refer to Ref. [76] for the derivation of the expressions and proof of the isomorphism property.

First, we introduce the $\gamma$-matrices used in the expressions of the images of the Hamiltonian and representations under the dimensional raising map,

$$\gamma_{2n-1}^{(k)} = \left(\bigotimes^{n-1}\sigma_0\right) \otimes \sigma_2 \otimes \left(\bigotimes^{k-n}\sigma_3\right), \tag{20}$$

$$\gamma_{2n}^{(k)} = \left(\bigotimes^{n-1}\sigma_0\right) \otimes -\sigma_1 \otimes \left(\bigotimes^{k-n}\sigma_3\right), \tag{21}$$

for $1 \leq n \leq k$ and $\gamma_{2k+1}^{(k)} = \bigotimes^k \sigma_3, \gamma_1^{(0)} = 1$ where $\bigotimes^n \sigma_j = \sigma_j \otimes ... \otimes \sigma_j$ describes the $n$-fold Kronecker product of the Pauli matrix $\sigma_j$. The $\gamma$-matrices satisfy $\{\gamma_i^{(k)}, \gamma_i^{(j)}\} = 2\delta_{i,j}$. We furthermore define

$$\begin{aligned}\epsilon_j^{(+)} &= \mathbb{1} \otimes \gamma_{2j}^{(r)} \\ \epsilon_j^{(-)} &= \mathbb{1} \otimes i\gamma_{2j-1}^{(r)}\end{aligned}. \tag{22}$$

### B.4.1 Dimensional raising from a symmetry class with chiral antisymmetry

The dimensional raising map is expressed in terms of a map of a representative Hamiltonian $H(\mathbf{k})$ defined on the base space $\mathbf{k} \in X$ and representations $U(g)$ of the symmetry operators $g \in \mathcal{G} \times \mathcal{C}$. Here, $\mathcal{G}$ is the magnetic space group that includes all unitary or antiunitary symmetries of system, such as time-reversal and crystalline symmetry operations. The chiral antisymmetry $\mathcal{C}$ plays a special role in the construction of the dimensional raising map. In particular, one defines the subgroup $\mathcal{G}_{\mathcal{C}} \subset \mathcal{G} \times \mathcal{C}$ compatible with chiral antisymmetry $\mathcal{C}$ that consist of all elements $g \in \mathcal{G} \times \mathcal{C}$ that satisfy

$$\begin{aligned} U(g)H(g\mathbf{k})U^{\dagger}(g) &= c(g)H(\mathbf{k}) \\ U(g)U(\mathcal{C})U^{\dagger}(g) &= c(g)U(\mathcal{C}) \end{aligned}, \tag{23}$$

for $g$ unitary and

$$\begin{aligned} U(g)H^*(-g\mathbf{k})U^{\dagger}(g) &= c(g)H(\mathbf{k}) \\ U(g)U^*(\mathcal{C})U^{\dagger}(g) &= c(g)U(\mathcal{C}) \end{aligned}, \tag{24}$$

for $g$ antiunitary with the same value $c(g) \in \{-1, 1\}$. For $c(g) = 1$ ($c(g) = -1$) the element $g \in \mathcal{G}_{\mathcal{C}}$ is a symmetry (antisymmetry). Note that $\mathcal{G}_{\mathcal{C}}$ is a normal subgroup of $\mathcal{G} \times \mathcal{C}$ and $\mathcal{G}_{\mathcal{C}} \times \mathcal{C} = \mathcal{G} \times \mathcal{C}$. The elements of $\mathcal{G}_{\mathcal{C}}$ are going to be used to construct the symmetry elements in the image of the dimensional raising map.

To define the dimensional raising map for a representative Hamiltonian $H_1(\mathbf{k})$ of a given (nontrivial) topological equivalence class, one considers a parameter family of Hamiltonians $H_{10}(\mathbf{k}, m)$ with $m \in [m_0, m_1]$ such that $H(\mathbf{k}, m_0) = H_0(\mathbf{k})$ is a representative Hamiltonian of a trivial topological equivalence class and $H(\mathbf{k}, m_1) = H_1(\mathbf{k})$. As $H_0(\mathbf{k})$ and $H_1(\mathbf{k})$ are in distinct topological equivalence classes, the gap needs to close for some finite value of the parameter $m$.

For two-dimensional $n$-fold rotation symmetry protected second-order topological superconductors in a Cartan class with chiral antisymmetry, the symmetry group $G$ is generated by rotations $\mathcal{R}_{2\pi/n}$ and, if present, time-reversal symmetry $\mathcal{T} = \mathcal{PC}$. In the presence of spin-rotation symmetry (or other internal unitary symmetries), the Hamiltonian is block-diagonal such that the analysis can be restricted to separate blocks.

*Raising by an odd number of dimensions* – First, we show how to construct a Hamiltonian in dimension $d = 2 + 2r + 1$ from a second-order topological phase in $d = 2$. Without loss of generality, we assume that $H_{10}(\mathbf{k}, m)$ interpolates between the second-order topological phase for $-2 < m < 0$ and a trivial phase for $m > 0$. We denote the momentum directions of the two dimensional second-order topological phase by $\mathbf{k} = (k_x, k_y)^T$ and the newly added momentum directions by $\mathbf{k}_{\perp}$. The dimensional raising map is given by defining the $(d = 2 + 2r + 1)$-dimensional Hamiltonian $H(\mathbf{k}, \mathbf{k}_{\perp})$ inheriting its topological invariants from the family of two-dimensional Hamiltonians $H_{10}(\mathbf{k}, m)$ as

$$\begin{aligned} H(\mathbf{k}, \mathbf{k}_{\perp}) &= \mathbb{H}_{10}(\mathbf{k}, m(\mathbf{k}_{\perp})) \\ &+ \sum_{j=1}^{r} (i\epsilon_j^{(-)} \sin k_{\perp,j} + \epsilon_j^{(+)} \sin k_{\perp,r+j}) \\ &+ \epsilon_{\mathcal{C}} \sin k_{\perp,2r+1}, \end{aligned} \tag{25}$$

with $\mathbb{H}_{10}(\mathbf{k}, m(\mathbf{k}_{\perp})) = H_{10}(\mathbf{k}, m(\mathbf{k}_{\perp})) \otimes \gamma_{2r+1}^{(r)}$, $\epsilon_{\mathcal{C}} = U(\mathcal{C}) \otimes \gamma_{2r+1}^{(r)}$ and $m(\mathbf{k}_{\perp}) = -1 + \sum_{j}^{2r+1}(1 - \cos k_{\perp,j})$. To express the corresponding representations of the symmetry elements $U(g, \mathbf{k}, \mathbf{k}_{\perp})$ we first introduce

$$\mathbb{U}(g, \mathbf{k}) = \begin{cases} U(g, \mathbf{k}) \otimes \mathbb{1} & \text{for } c(g) = 1 \\ U(g, \mathbf{k}) \otimes \gamma_{2r+1}^{(r)} & \text{for } c(g) = -1. \end{cases} \tag{26}$$

This allows us to express $U(g, \boldsymbol{k}, \boldsymbol{k}_\perp)$ for $g \in \mathcal{G}_\mathcal{C}$ unitary as

$$U(g, \boldsymbol{k}, \boldsymbol{k}_\perp) = \mathbb{U}(g, \boldsymbol{k}) \tag{27}$$

and for $g \in \mathcal{G}_\mathcal{C}$ antiunitary as

$$U(g, \boldsymbol{k}, \boldsymbol{k}_\perp) = \left(\prod_{j=1}^{r} \epsilon_j^{(+)}\right) \epsilon_\mathcal{C} \mathbb{U}(g, \boldsymbol{k}). \tag{28}$$

The mapped Hamiltonian (25) satisfies for $g \in \mathcal{G}_\mathcal{C}$ unitary

$$U(g, \boldsymbol{k}, \boldsymbol{k}_\perp) H(g\boldsymbol{k}, \boldsymbol{k}_\perp) U^\dagger(g, \boldsymbol{k}, \boldsymbol{k}_\perp) = c(g) H(\boldsymbol{k}, \boldsymbol{k}_\perp) \tag{29}$$

and for $g \in \mathcal{G}_\mathcal{C}$ antiunitary

$$\begin{aligned} &U(g, \boldsymbol{k}, \boldsymbol{k}_\perp) H^*(-g\boldsymbol{k}, -\boldsymbol{k}_\perp) U^\dagger(g, \boldsymbol{k}, \boldsymbol{k}_\perp) \\ &= (-1)^{r+1} c(g) H(\boldsymbol{k}, \boldsymbol{k}_\perp). \end{aligned} \tag{30}$$

*Raising by an even number of dimensions*– Under the dimensional raising map a $d = 2 + 2r$ dimensional Hamiltonian inheriting its topological invariants from the family of two-dimensional Hamiltonians $H_{10}(\boldsymbol{k}, m)$ is constructed as

$$\begin{aligned} H(\boldsymbol{k}, \boldsymbol{k}_\perp) = {}&\mathbb{H}_{10}(\boldsymbol{k}, m(\boldsymbol{k}_\perp)) \\ &+ \sum_{j=1}^{r-1} (i\epsilon_j^{(-)} \sin k_{\perp,j} + \epsilon_j^{(+)} \sin k_{\perp,r+j}) \\ &+ \epsilon_r^{(-)} \sin k_{\perp,r} + \epsilon_\mathcal{C} \sin k_{\perp,2r}, \end{aligned} \tag{31}$$

where the representations $U(g, \boldsymbol{k}, \boldsymbol{k}_\perp)$ are given for $g \in \mathcal{G}_\mathcal{C}$ unitary as

$$U(g, \boldsymbol{k}, \boldsymbol{k}_\perp) = \mathbb{U}(g, \boldsymbol{k}) \tag{32}$$

and for $g \in \mathcal{G}_\mathcal{C}$ antiunitary as

$$U(g, \boldsymbol{k}, \boldsymbol{k}_\perp) = \left(\prod_{j=1}^{r-1} \epsilon_j^{(+)}\right) \epsilon_\mathcal{C} \mathbb{U}(g, \boldsymbol{k}). \tag{33}$$

In addition, the Hamiltonian (31) satisfies the unitary antisymmetry $U(\mathcal{C}, \boldsymbol{k}, \boldsymbol{k}_\perp) = \epsilon_r^{(+)}$,

$$\epsilon_r^{(+)} H(\boldsymbol{k}, \boldsymbol{k}_\perp) \epsilon_r^{(+)} = -H(\boldsymbol{k}, \boldsymbol{k}_\perp). \tag{34}$$

The mapped Hamiltonian (31) satisfies for $g \in \mathcal{G}_\mathcal{C}$ unitary

$$U(g, \boldsymbol{k}, \boldsymbol{k}_\perp) H(g\boldsymbol{k}, \boldsymbol{k}_\perp) U^\dagger(g, \boldsymbol{k}, \boldsymbol{k}_\perp) = c(g) H(\boldsymbol{k}, \boldsymbol{k}_\perp) \tag{35}$$

and for $g \in \mathcal{G}_\mathcal{C}$ antiunitary

$$\begin{aligned} &U(g, \boldsymbol{k}, \boldsymbol{k}_\perp) H^*(-g\boldsymbol{k}, -\boldsymbol{k}_\perp) U^\dagger(g, \boldsymbol{k}, \boldsymbol{k}_\perp) \\ &= (-1)^{r} c(g) H(\boldsymbol{k}, \boldsymbol{k}_\perp). \end{aligned} \tag{36}$$

The same commutation relations hold for the representations of the symmetry elements and the chiral antisymmetry $U(\mathcal{C}, \boldsymbol{k}, \boldsymbol{k}_\perp) = \epsilon_r^{(+)}$. In particular, they are for $g \in \mathcal{G}_\mathcal{C}$ unitary

$$U(g, \boldsymbol{k}, \boldsymbol{k}_\perp) U(\mathcal{C}, \boldsymbol{k}, \boldsymbol{k}_\perp) U^\dagger(g, \boldsymbol{k}, \boldsymbol{k}_\perp) = c(g) U(\mathcal{C}, \boldsymbol{k}, \boldsymbol{k}_\perp) \tag{37}$$

and for $g \in \mathcal{G}_\mathcal{C}$ antiunitary

$$\begin{aligned} &U(g, \boldsymbol{k}, \boldsymbol{k}_\perp) U^*(\mathcal{C}, \boldsymbol{k}, \boldsymbol{k}_\perp) U^\dagger(g, \boldsymbol{k}, \boldsymbol{k}_\perp) \\ &= (-1)^{r} c(g) U(\mathcal{C}, \boldsymbol{k}, \boldsymbol{k}_\perp). \end{aligned} \tag{38}$$

### B.4.2 No-go theorem

Two-dimensional $n$-fold rotation symmetry protected second-order topological phases in Cartan classes AIII, BDI and CII (whose 0-dimensional anomalous states have $\mathbb{Z}$ topological charge) require that the representations of chiral antisymmetry $U(\mathcal{C})$ and $n$-fold rotation symmetry $U(\mathcal{R}_{2\pi/n})$ anticommute. In this case, the conditions (23), (24) imply that the group of symmetry elements compatible with chiral antisymmetry $\mathcal{G}_{\mathcal{C}}$ is generated by a rotation *antisymmetry* $\mathcal{R}_{2\pi/n}\mathcal{C}$ and, depending on the Cartan class, either time-reversal symmetry or particle-hole antisymmetry. Thus $\mathcal{G}_{\mathcal{C}}$ does not contain any unitary rotation symmetries.

Upon raising the dimension by an odd number, Eqs. (29) and (30) also show that the rotation elements in the image of the dimensional raising map cannot be both unitary and commute with the Hamiltonian. Upon raising the dimension by an even number, Eq. (37) shows that the anticommutation of the chiral antisymmetry with the representation of (unitary) rotation symmetry remains preserved. Antiunitary symmetry elements remain antiunitary under the dimensional raising map.

Finally, any $d > 2$ dimensional rotation symmetry protected second-order topological phase can be constructed from a $d = 2$ dimensional second-order topological phase where the corresponding model can be found by using the inverse dimensional reduction map of the isomorphism. The dimensional reduction map can be explicitly expressed in terms of a continuous deformation of the Hamiltonian [47] or in terms of the scattering matrix of a symmetric boundary [11, 86].

Therefore, the criterion from Sec. 2.5 of the main text for the application of the unique bulk hybridization (2) is violated for all symmetry classes and dimensions $d \geq 2$ that host $n$-fold rotation symmetry protected topological phases with $\mathbb{Z}$ anomalous boundary states. This proves the absence of a unique correspondence between $(d-2)$ dimensional disclination anomaly and strong bulk topology in these symmetry classes.

## C  Details on the topological crystal construction

We present additional details on the topological crystal construction. Section C.1 shows in detail how to perform the cell decomposition with space group $p2$. Section C.2 contains details on how to determine the valid decorations as well as their properties in terms of weak and strong topological phases as well as their Abelian group property. Finally, in Sec. C.3 we show that an obstruction exists to decorate the $d-1$ cells of lattices with disclination with $\mathbb{Z}$ topological phases.

### C.1  Cell decomposition with space group $p2$

The cell decomposition of a cubic lattice with space group $p2$ in two and three dimensions is shown in Fig. 5 in the main text.

*Two dimensions.* In two dimensions, we choose the asymmetric unit to be bounded by the lines $(x,0)$ with $x \in [0, \frac{a}{2}]$, $(0,y)$ and $(\frac{a}{2},y)$ with $y \in [0,a]$. This choice is made such that a corner of the asymmetric unit lies at the unit cell center and its edges are parallel to the lattices vectors. Note that with this choice, the asymmetric unit extends over two adjacent unit cells. In particular, the boundary of the unit cell at $y = \frac{a}{2}$ is not a boundary of the asymmetric unit as chosen here.

There are three symmetry inequivalent 1-cells which together cover the complete boundary of the 2-cell upon translating them with the crystalline translation and twofold rotation symmetries. They are given by the lines: i) $(x,0)$ with $x \in [0, \frac{a}{2}]$, ii) $(0,y)$ with $y \in [0, \frac{a}{2}]$,

and iii) $(\frac{a}{2}, y)$ with $y \in [0, \frac{a}{2}]$.

The symmetry-inequivalent 0-cells are the boundaries of the 1-cells. They coincide with maximal Wyckoff positions, which are labeled in standard notation as 1a at $\boldsymbol{x} = (0, 0)$, 1b at $\boldsymbol{x} = (\frac{a}{2}, 0)$, 1c at $\boldsymbol{x} = (0, \frac{a}{2})$ and 1d at $\boldsymbol{x} = (\frac{a}{2}, \frac{a}{2})$.

*Three dimensions.* In three dimensions, see Fig. 5 in the main text, the asymmetric unit is a cuboid where every $x$-$y$ plane cut reproduces the two-dimensional asymmetric unit. Without loss of generality and for simplicity, we set the boundaries of the cube at the planes $z = \pm\frac{a}{2}$.

There are four symmetry-inequivalent 2-cells:

- (red) the $x$-$y$ plane bounded between the lines $(x, 0, \frac{a}{2})$ with $x \in [0, \frac{a}{2}]$, $(0, y, \frac{a}{2})$ and $(\frac{a}{2}, y, \frac{a}{2})$ with $y \in [0, a]$,

- (blue) an $x$-$z$ plane bounded between the lines $(x, 0, \pm\frac{a}{2})$ with $x \in [0, \frac{a}{2}]$, $(0, 0, z)$ (Wyckoff position 1a) and $(\frac{a}{2}, 0, z)$ (Wyckoff position 1b) with $z \in [-\frac{a}{2}, \frac{a}{2}]$,

- (yellow) a $y$-$z$ plane bounded by the lines $(0, y, \pm\frac{a}{2})$ with $y \in [0, \frac{a}{2}]$, $(0, 0, z)$ (Wyckoff position 1a) and $(0, \frac{a}{2}, z)$ (Wyckoff position 1c) with $z \in [-\frac{a}{2}, \frac{a}{2}]$ and

- (green) another $y$-$z$ plane bounded by the lines $(\frac{a}{2}, y, \pm\frac{a}{2})$ with $y \in [0, \frac{a}{2}]$, $(\frac{a}{2}, 0, z)$ (Wyckoff position 1b) and $(\frac{a}{2}, \frac{a}{2}, z)$ (Wyckoff position 1d) with $z \in [-\frac{a}{2}, \frac{a}{2}]$.

There are seven symmetry inequivalent 1-cells, three in the $x$-$y$-plane similar to the two dimensional cell decomposition, and four at the four maximal Wyckoff positions 1a at $\boldsymbol{x} = (0, 0, z)$, 1b at $\boldsymbol{x} = (\frac{a}{2}, 0, z)$, 1c at $\boldsymbol{x} = (0, \frac{a}{2}, z)$ and 1d at $\boldsymbol{x} = (\frac{a}{2}, \frac{a}{2}, z)$, $z \in [-\frac{a}{2}, \frac{a}{2}]$.

Finally, there are four symmetry inequivalent 0-cells at $\boldsymbol{x} = (0, 0, \frac{a}{2})$, $\boldsymbol{x} = (\frac{a}{2}, 0, \frac{a}{2})$, $\boldsymbol{x} = (0, \frac{a}{2}, \frac{a}{2})$ and $\boldsymbol{x} = (\frac{a}{2}, \frac{a}{2}, \frac{a}{2})$.

## C.2 Decorations of topological crystals with rotation symmetry

In this section we present in detail the decoration of the 1-cells (2-cells parallel to the rotation axis) in two (three) dimensional rotation symmetric lattices with $\mathbb{Z}$ topological phases and the properties of the resulting decorations in terms of weak, strong and higher-order topological phases. The results for decorations with topological phases with $\mathbb{Z}_2$ anomalous edge states can be straightforwardly obtained by taking the fusion rules modulo two. Below we present the derivation for two dimensional lattices and comment on the straightforward extension to three dimensions. At the end of the section we show some criteria that simplify the determination of the existence of a decoration.

*Twofold rotation symmetry in two dimensions.* With twofold rotation symmetry, the unit cell and cell decomposition is shown in Fig. 5 in the main text.

The asymmetric unit is a 2-cell that can be decorated with a two dimensional topological phase. The decoration describes a gapped topological phase if the anomalous state along its boundaries can be gapped by hybridization with anomalous boundary states of adjacent decorated 2-cells. Note that the hybridization of edge states along the 1-cells may create anomalous states at the twofold rotation axis.

There are in total $\mathbb{Z}^3$ topological crystals that can be constructed from decorations of the three distinct 1-cells with $\mathbb{Z}$ topological phases. A given 1-cell decoration can be identified by the vector $\boldsymbol{\nu} = (\nu_{1a|1b}, \nu_{1a|1c}, \nu_{1b|1d})$ where $\nu_{i|j}$ is the topological invariant characterizing the topological phase occupying the 1-cell between Wyckoff positions $i$ and $j$ where $i, j \in \{1a, 1b, 1c, 1d\}$. As we show in Fig. C.4 (d) some topological crystals are topologically equivalent in the sense that they can be adiabatically deformed into each other: A topological crystal describing the element $(0, 2, 0)$ can be written as a direct sum $H_{(0,2,0)} = H_{(0,1,0)} \oplus_\mu H_{(0,1,0)}$ where $H_{\boldsymbol{\nu}}$ is a Hamiltonian describing the topological crystal with topological invariants $\boldsymbol{\nu}$ and

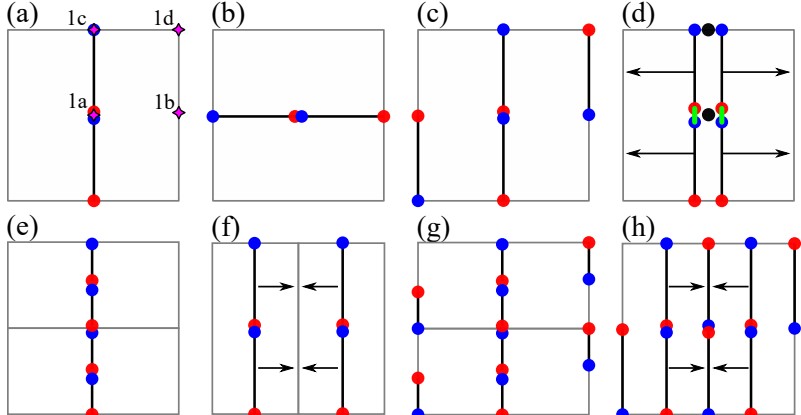

Figure C.4: Panels (a), (b) and (c) repeat for convenience the generating set of valid 1-cell decorations for two dimensional twofold rotation symmetric lattices with topological invariants as in Fig. 6 in the main text. In panel (a) we included the labels of the maximal Wyckoff positions at the twofold rotation axes denoted by the violet stars. The decorations (a), (b), (c) have the topological invariants $\nu = (\nu_{1a|1b}, \nu_{1a|1c}, \nu_{1b|1d}) = (0,1,0)$, $(1,0,0)$ and $(0,1,-1)$ as defined in the text, respectively. Figure (d) shows that a topological crystal $\nu = (0,2,0)$ can be adiabatically and symmetrically deformed to a topological crystal $\nu = (0,0,2)$ up to additional $d-2$ dimensional topological phases at and parallel to the rotation axis (black dots) by deforming the hybridization of the anomalous edge states (green bars) such that it allows a symmetric and adiabatic movement of the decoration. (e), (f) Doubling of the unit cell in $y$, $x$-direction with a decoration with the topological crystal shown in (a). The topological crystal shown in (a) is invariant under a doubling of the unit cell in $y$ direction after hybridization of the anomalous edge states. A doubling of the unit cell in $x$ direction takes the Hamiltonian $H$ describing the topological crystal shown in (a) to $H \oplus H$ after a movement of the symmetry related pairs of 1-cells as shown in (f). In contrast, the topological crystal shown in (c) is invariant under a doubling of the unit cell in $y$, $x$ direction shown in (g), (h) using hybridization and symmetry allowed deformations.

the subscript $\mu$ indicates that Pauli matrices denoted by $\mu_j$ act in the space of the two systems. A simple check directly shows that if $H_{(0,1,0)}$ describes a gapped system whose anomalous edge states at the maximal Wyckoff positions hybridize and whose hybridization is given by $h$, then $h \otimes \mu_1$ is a possible hybridization of $H_{(0,2,0)}$. This hybridization is shown by the green bars in Fig. C.4 (d) and allows to symmetrically move the decoration to the 1-cell pointing from 1b to 1d. [12]

Note that in order to adiabatically deform the hybridization $h \otimes \mu_0$ obtained from the direct sum $H_{(0,1,0)} \oplus_\mu H_{(0,1,0)}$ to the required $h \otimes \mu_1$ one may need to add extra gapped degrees of freedom at the rotation axis. In two dimensions, these additional degrees of freedom are a 0-cell describing a gapped orbital. In higher dimensions, the additional degrees of freedom can be a $d-2$-dimensional topological phase that decorates the $d-2$-cell that coincides with the rotation center. The additional degrees of freedom remain as $d-1$-cells are moved. As

---

[12]The block-off-diagonal coupling $h \otimes \mu_1$ is required for the following reason. Twofold rotation symmetry exchanges the $(d-1)$-cell decorations within each block. Upon symmetrically moving the decorations away from the $(d-1)$-cells at the center of the unit cell, any diagonal hopping $h \otimes \mu_0$ or $h \otimes \mu_3$ would become non-local because the twofold rotation symmetry related decorations within a single block move in opposite directions. Only a block-off-diagonal coupling $h \otimes \mu_1$ remains local upon moving the decorations of each block symmetrically in opposite directions, as shown in Fig. C.4 (d).

in this article we restrict the analysis to the properties of the topological crystals obtained from $d-2$-cells, we do not discuss under which conditions the addition of gapped degrees of freedoms is necessary, nor the properties of the resulting topological crystals.

This shows that the element $(0, 2, 0)$ is equivalent to the element $(0, 0, 2)$ up to an atomic limit or decorated $d-2$ cells parallel to the rotation axis. Note that a similar deformation is not allowed for a decoration $(0, 1, 0)$ as this would require either a hybridization of non-overlapping anomalous edge states of the 1-cells or an absence of a hybridization of the anomalous edge states which corresponds to a bulk gap closure.

A complete set of 1-cell decorations from which all topological crystals can be constructed using the direct sum operation is given by $\nu = (0, 1, 0)$, $(1, 0, 0)$ and $(0, 1, -1)$ shown in Fig. C.4 (a), (b) and (c), respectively. In order for the decorations to be valid, i.e., to describe a gapped topological phase, all anomalous edge states of the decorations need to gap out with overlapping anomalous states at the same location. Each 1-cell ends at a twofold rotation axis and thus has a partner under twofold rotation. A sufficient criterion for the validity of all decorations $\nu = (0, 1, 0)$, $(1, 0, 0)$ and $(0, 1, -1)$ is that the anomalous states at the rotation axis gap out with their partner under twofold rotation. This requires that the topological charge at each rotation axis is balanced. For this space group, this criterion is also necessary as Wyckoff positions 1c and 1d border only to a twofold rotation symmetry related pairs of 1-cells.

The topological crystals shown in Fig. C.4 (a), (b) are real space limits of weak topological phases corresponding to stacks of twofold rotation symmetric one dimensional topological phases. Figure C.4 (f) shows that the topological crystal shown in Fig. C.4 (a) can be trivialized by a doubling of the unit cell in $x$ direction. However, it is invariant under a doubling of the unit cell in $y$ direction as shown in Figure C.4 (e). A similar argument holds for the topological crystal shown in Fig. C.4 (b). The topological crystal shown in Fig. C.4 (d) is the real space limit of a strong second-order topological superconductor. It is invariant under a redefinition of the unit cell in both $x$ and $y$ direction, as seen in Fig. C.4 (g) and (h), respectively. Due to the equivalence relation, the strong second-order topological phase has $\mathbb{Z}_2$ character, i.e. the topological crystal with topological invariants $(0, 2, -2)$ can be adiabatically deformed to the trivial crystal $(0, 0, 0)$ up to an atomic limit or decorated $d-2$ cells parallel to the rotation axis. In case decorated $d-2$ cells remain, the resulting topological crystal may be a strong third order topological phase, as expected from the $K$-theoretic results from Ref. [12].

*Fourfold rotation symmetry, two dimensions.* With fourfold rotation symmetry, the unit cell and cell decomposition is shown in Fig. 5 in the main text. The decoration of the 2-cell with a two dimensional topological phase is valid if all 1-cells and 0-cells of its boundaries gap out upon hybridizing the anomalous edge states of adjacent 2-cells.

There are two distinct 1-cell decorations shown in Fig. 6 (d) and (e) in the main text. Similar arguments as for twofold rotatation symmetry show that (a) is a weak topological phase and (b) is a strong topological second-order topological phase with $\mathbb{Z}_2$ character due to an equivalence relation invovling an adiabatic and symmetric deformation moving the 1-cell decorations between the two 1-cells. By construction, every fourfold rotation symmetric Wyckoff position is the edge of four 1-cell decorations and therefore hosts four zero-dimensional anomalous edge states of the 1-cells. As every fourfold rotation symmetric Wyckoff position also satisfies twofold rotation symmetry, gapping of twofold rotation symmetry related pairs of anomalous edge modes of 1-cells is a sufficient criterion for the existence of both 1-cell decorations. This criterion is also necessary for the weak topological phase shown in Fig. 6 (d) in the main text as the twofold rotation symmetric Wyckoff position 2c border only to a twofold rotation symmetry related pair of 1-cells. However, this criterion is not necessary for the strong second-order topological phase shown in Fig. 6 (e) in the main text. In fact, the 1-cells of the weak topological crystal cannot be decorated with $\mathbb{Z}$ topological phases as the anomaly cancellation criterion cannot be satisfied both at twofold and fourfold rotation axis: In order

to gap the anomalous states at fourfold rotation axis, fourfold rotation needs to invert the $\mathbb{Z}$ topological charge. As the action of twofold rotation is given by a double action of fourfold rotation, twofold rotation leaves the topological charge invariant. As in the weak phase, the twofold rotation axes are occupied only by two anomalous states related by twofold rotation, their topological charge needs to be equal. This prohibits anomalous states with $\mathbb{Z}$ topological charge to gap at the twofold rotation axis of the weak topological crystal. An example where the weak topological phase is forbidden while the strong second-order topological phase exists is a magnetic topological insulator as discussed in Sec. B.3.

*Sixfold rotation symmetry, two dimensions.* With sixfold rotation symmetry, the unit cell and cell decomposition is shown in Fig. 5. As before, the decoration of the 2-cell with a two dimensional topological phase is valid if all 1-cells and 0-cells of its boundaries gap out upon hybridizing the anomalous edge states of adjacent 2-cells. There is only a single decoration of 1-cells shown in Fig. 6 (f) that is consistent with the anomaly cancellation criterion at each rotation axis. A decoration of 1-cells ending at a threefold rotation symmetric momentum cannot be consistent with the anomaly cancellation criterion for tenfold-way topological insulators and superconductors. The valid 1-cell decoration is a strong second-order topological phase. As every sixfold rotation symmetric Wyckoff position also satisfies twofold rotation symmetry and Wyckoff position 3c borders only to a twofold rotation symmetry related pair of 1-cells, gapping of twofold rotation symmetry related pairs of anomalous edge modes of 1-cells is a sufficient and necessary criterion for the existence of both 1-cell decorations.

*Extension to three dimensions.* In three dimensions, the cell decompositions for space groups $p2$, $p4$ and $p6$ are shown in figures Fig. 5 in the main text. For the three dimensional asymmetric unit, all 2-cells and the 1-cells perpendicular to the rotation axis similar arguments as in two dimensions apply. Decorations of the 1-cells parallel to the rotation axis would give rise to a weak topological phase and a third order topological phase hosting an anomalous state at a rotation symmetric corner of the crystal. As in this article we focus on topological crytalline phases that may host second-order anomalous states at disclinations, we omit the construction of topological crystals corresponding to decorations of 1-cells parallel to the rotation axis.

*A necessary criterion on a strong first order topological phase for fourfold and sixfold rotation symmetry.* A first order topological phase in a given topological crystal exists if the decoration of the asymmetric unit with the first order topological phase is valid. As both the fourfold and sixfold rotation symmetric lattices contain a separate twofold rotation axis, the decoration with the first order topological phase in fourfold and sixfold rotation symmetric lattices is possible only if the corresponding decoration is possible in the twofold rotation symmetric lattice.

*Connection to K-theoretic classification schemes.* The existence of a mass term gapping anomalous states related by $n$-fold rotation symmetry can be determined from the existence of a strong second-order phase protected by $n$-fold rotation symmetry in the respective symmetry class and dimension as determined from $K$-theoretic methods (see Refs. [11,12,25,87]). This follows as it has been shown in these articles that a strong second-order topological phase can be deformed into symmetry related $d-1$ dimensional building blocks decorated with $d-1$ dimensional first order topological phases. This limit is identical to the topological crystal limit locally around a rotation axis. This draws a one-to-one correspondence between the existence of mass terms hybridizing symmetry related $d-2$ dimensional anomalous states at a rotation axis and the existence of a second-order topological phase protected by the same rotation symmetry.

## C.3 Obstruction to decorate a lattice with disclination with $\mathbb{Z}$ topological phases

We show that the topological crystal limit of second-order or weak topological phases in symmetry classes with $\mathbb{Z}$ anomalous boundary modes reveals an obstruction to realize these phases

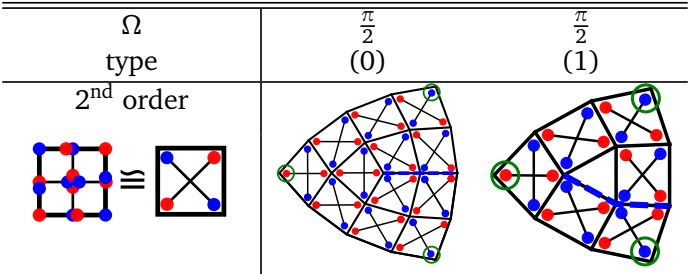

Figure C.5: Decoration of a fourfold symmetric lattice with $\pi/2$ disclinations of both types with strong second-order topological phases with $\mathbb{Z}$ anomalous boundary state. The first row shows the corresponding topological crystals. For simplicity, the anomalous bound states that hybridize within the unit cell are not shown.

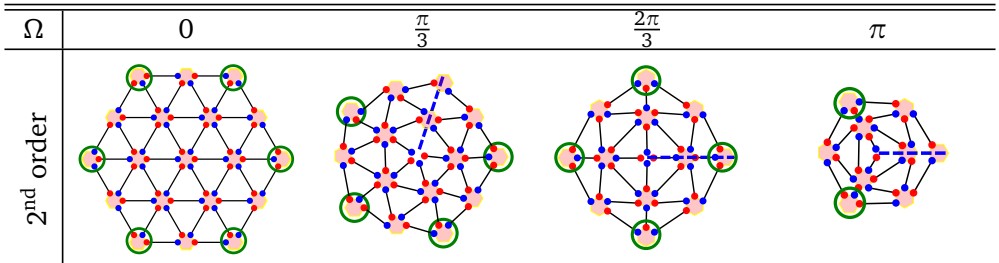

Figure C.6: Sixfold rotation symmetric lattice with disclination with Frank angle $\Omega$ decorated with a second-order topological phase whose $d-2$ dimensional anomalous states have $\mathbb{Z}$ topological charge. Note for the lattice with $\pi/3$ or $\pi$ disclination that it is impossible to choose a decoration along the cut line that preserve the sixfold rotation symmetry. In these cases we omit the decoration of the $(d-1)$-cells along the cut line. For the $2\pi/3$ disclination the unique decoration pattern of the second-order topological phase is shown.

on a lattice with disclination such that the system is locally indistinguishable from the bulk everywhere except at the disclination.

Upon forming a $2\pi/n$ disclination by folding the lattice, the unit cells are rotated by $\frac{2\pi}{n}$ in real space without applying any interal action on the degrees of freedom within the unit cell. For $(d-1)$-cell decorations with edge states with $\mathbb{Z}$ topological charge, the onsite action is resposible for inverting the topological charge of symmetry related anomalous states within the unit cell. Due to the absence of the interal action in the rotation of the lattice during the Volterra process, unit cells whose configuration of anomalous states is rotated by $2\pi/n$ are brought next to each other. These unit cells form a continuous line connecting the disclination to the boundary or to another disclination. At each point along the line, the Hamiltonian is locally *distinguishable* from the bulk. Local rotations of unit cells can move, but not remove this line.

*Fourfold rotation symmetry*. For example, consider the fourfold rotation symmetric lattice with $\pi/2$ disclination depicted in the left column of Fig. C.5. The cut line over which the system was folded is visible as the decorations of adjacent unit cells are rotated with respect to each other. Along this line, the overlapping anomalous states are located in a way that is inconsistent with fourfold rotation symmetry as it is defined in the translation-invariant bulk. It is impossible to apply the same hybridization term as in the bulk.

*Sixfold rotation symmetry*. With sixfold rotation symmetry, the topological crystal construction of the second-order topological phase dictates that every line connecting nearest sixfold

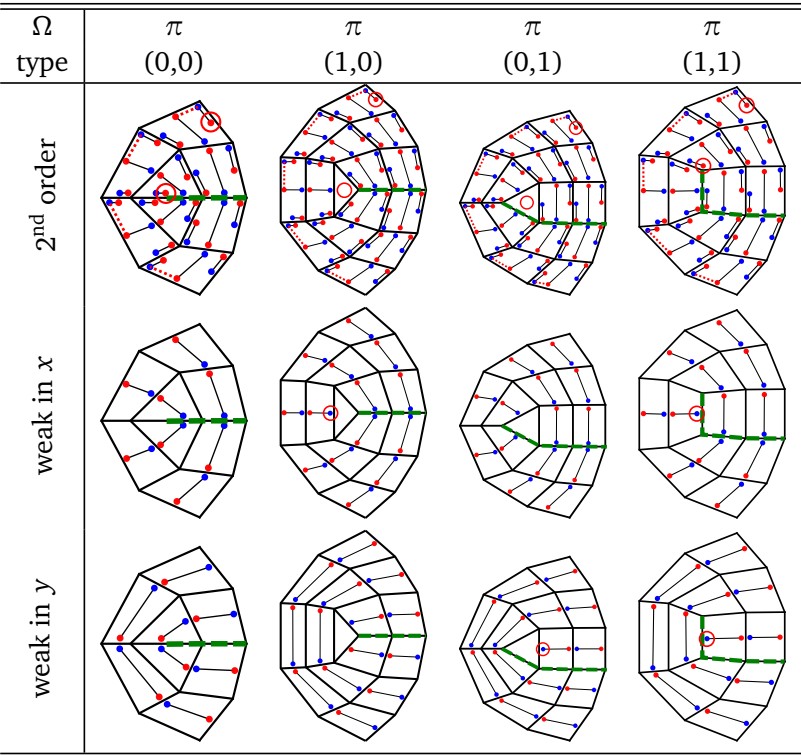

Figure C.7: Decorations of a twofold rotation symmetric lattice with $\pi$ disclinations of all types with second-order topological phases protected by twofold rotation symmetry and weak phase as a stack in $x$, $y$ direction. Red and blue dots denote $d-2$ dimensional anomalous states with $\mathbb{Z}$ topological charge $\pm 1$. The red dashed lines denote a possible hybridization of the anomalous states on the surface. The green dashed line denotes the cut line.

rotation centers is decorated with a topological phase. The corresponding decorations on a lattice with $\pi/3$, $2\pi/3$ or $\pi$ disclination is shown in Fig. C.6. For a lattice with $\pi/3$ or $\pi$ disclination, a decoration of the cut line with $\mathbb{Z}$ topological phases that respects the sixfold rotation symmetry all along the cut line is not possible. A symmetric decoration of a lattice with $2\pi/3$ disclination is possible. In this case the disclination does not host an anomalous state.

*Twofold rotation symmetry.* The generators of $d-2$ cell decorations of a twofold rotation symmetric lattice with $\pi$ disclinations of all types are shown in Fig. C.7. As before, rotation symmetry is broken along the cut line. For weak topological phases, if the disclination is created by folding the gapless surface then there is an array of anomalous states with the same topological charge along the cut line which cannot be gapped. If the disclination is created by folding the gapped surfaces then the cut line in the folded lattice with $\pi$ disclination is gapped. In this case the weak topological phase that corresponds to a stack of lower dimensional topological phases in the $x$, $y$ direction hosts anomalous disclination states at disclination of type (1,0), (0,1), respectively. The anomalous states at the boundary switch their topological charge at the intersection of the cut line with the boundary.

### C.3.1 Relation to the no-go theorem from Sec. B.4

In two dimensions, the local configuration of $\mathbb{Z}$ topological charges is expressed by the representation of chiral antisymmetry. In case the representations of chiral antisymmetry and rotation symmetry $\mathcal{R}_{2\pi/n}$ anticommute, a $2\pi/n$ rotation without applying the internal action of rotation symmetry exchanges the topological charges in the system. This implies that the

pattern of $\mathbb{Z}$ topological charges in two unit cells adjacent over the cut line connected to a $2\pi/n$ disclination needs to be distinguishable from the pattern in the bulk, which is the obstruction found from the topological crystal decoration of lattices with disclination as shown in Figs. C.5, C.6, C.7.

The anticommutativity between the representations of chiral antisymmetry and rotation symmetry is necessary for the existence of the second-order and weak topological crystal as the anomaly cancellation criterion (6) needs to be satisfied at each rotation axis in the unit cell. This shows that the obstruction to decorating a lattice with disclination such that it is locally indistinguishable from the bulk everywhere except at the disclination is related to the obstruction of applying the bulk hopping term due to the algebraic relations of the symmetry elements as discussed in Appendix B.4.

### C.4 Validity of the topological-crystal construction for inhomogeneous and finite size systems

In this subsection, we briefly comment on the validity of the topological-crystal construction for samples of finite size or with spatial inhomogeneities as necessarily present due to the strain around disclinations.

For a system with finite sample geometry and characteristic sample length scale $L$ (such as the distance between defects and/or boundaries), one needs to require that $L$ is much larger than any characteristic correlation length or entanglement length $\xi$ of the system. This requirement is necessary as our derivation for the existence of anomalous disclination states was performed in the topological crystal limit. Ref. [22] argues that the topological crystal limit is an element of the stable topological equivalence class of a topological crystalline phase if there exists an adiabatic process – while allowing to add an arbitrary fine mesh of trivial degrees of freedom – such that all characteristic correlation length or entanglement length scales $\xi$ can be reduced to be much smaller than the thickness $w$ of the topological phases on the lower-dimensional cells of the cell decomposition. Here, $w$ should be much smaller than the lattice spacing $a$. Ref. [22] argues that strictly speaking, the existence of such an adiabatic deformation for all elements in the stable topological equivalence class should be treated as a conjecture. If the conjecture does not hold, the topological crystal construction applies only to those systems for which the adiabatic deformation exists.

The argumentation of Ref. [22] was performed for systems with periodic boundary conditions. Therefore, we expect that this deformation also holds in the bulk if the real-space variation of the Hamiltonian is slow on the scale of all internal length scales of the Hamiltonian $\xi$, such that one can regard each point in space to a good approximation as locally translation symmetric. Then, for sufficiently large and slowly varying systems, the topological crystal construction applies in the bulk and guarantees the existence of anomalous defect and boundary states – independently on whether the system directly at the defect or boundary can be deformed into the topological crystal limit.

# D First order topology and point or line defects

Strong first order topological phases are classified in the tenfold way [1,31] and do not require any crystalline symmetries for their protection. They may be realized in systems without a lattice structure where the dimensionality is only enforced by the locality of the Hamiltonian. Crystalline symmetries however may prohibit the existence of first order topological phases. A typical example is that the presence of a mirror symmetry requires the Chern number in a plane perpendicular to the mirror plane to vanish.

In Sec. D.1 below we show that the independence on any type of underlying lattice of strong first order topological phases implies that there can be no term $\propto \Omega \cdot \nu_1$ linking the rotation holonomy $\Omega$ of disclinations with the first order topological invariant $\nu_1$ contributing to the number of anomalous disclination states. However, tenfold-way first order topological phases respond to $\pi$-fluxes, which may be bound to point or line defects. In Sec. D.2 we list the cases in which they host anomalous states at point or line defects binding $\pi$-fluxes.

## D.1 First order topology and disclinations

In two dimensional space without an underlying lattice one can define disclinations with arbitrary Frank angle $\Omega$ as point defects such that any coordinate system that is parallel transported in a closed loop enclosing the disclination (and no additional disclinations) is rotated by $\Omega$. The rotation holonomy $\Omega_{\text{tot}}$ of a coordinate system parallel transported around several disclinations $j$ is given by the sum of their Frank angles $\Omega_{\text{tot}} = \sum_j \Omega_j$. Now suppose that a disclination with Frank angle $\Omega_0$ would host an anomalous disclination state in the strong first order topological phase. As the existence of disclination states in a topological phase has to be a property of the topological bulk, their existence may only depend on the rotation holonomy of a closed loop that may be deformed to be arbitrarily far away from the disclination (as long as the deformation of the loop does not cross any other disclinations). Thus a disclination with Frank angle $2\Omega_0$ can be constructed by moving two disclinations with Frank angle $\Omega_0$ to the same point in space. As a consequence, the topological charge at a disclination with Frank angle $2\Omega_0$ is twice the topological charge of a disclination with Frank angle $\Omega_0$. Furthermore, if a disclination with Frank angle $\Omega_0$ hosts an anomalous state with topological charge $Q = 1$, the topological charge at a disclination with Frank angle $\Omega_0/2$ has to depend on microscopic details of the disclination and cannot be a property of the topological bulk as the topological charge is quantized to integers.

As first order topological phases do not require any rotation symmetry for their topological protection that would single out a specific Frank angle $\Omega_0$, all disclinations independent on their Frank angle should have the same properties under topological deformations perseving the bulk gap of the first order topological phase. This is only possible if first order topological phases do not host anomalous states at disclinations.

These general arguments can be confirmed by combining the exact diagonalization results from Ref. [54] with the classification and corresponding topological invariants from Ref. [87] for twofold and fourfold rotation symmetric systems. In Ref. [54] models for topological superconductors with odd Chern number and trivial and nontrival weak invariants have been defined and systematically exactly diagonalized on lattices with disclinations. Applying the topological invariants from Ref. [87] to their corresponding models with twofold and fourfold rotation symmetry shows that these models are not simultaneously in a second-order topological phase. Thus the absence of Majorana bound states at disclinations of the odd Chern number model with trivial weak invariants confirms our general arguments from this section.

## D.2 First-order topological phases and $\pi$-fluxes

In this section we show that first order topological phases in symmetry classes and dimension $d$ that allow for $d-2$ dimensional anomalous states host anomalous states at $d-2$ dimensional defects with the property that the geometric phase $\alpha$ acquired by parallel transport around a closed a loop around the defect is $\pi$.

In two dimensions, this property corresponds to the familiar statement that p-wave superconductors with odd Chern number host Majorana bound states at vortex cores [63, 70]. By augmenting such a Chern superconductor with its time reversed copy by applying the same procedure as in 6.2, one constructs a two dimensional topological superconductor in class DIII.

This procedure defines a homomorphism $h_{\mathcal{T}} : \mathcal{K}_D(d=2) \to \mathcal{K}_{DIII}(d=2)$ from the classifying group $\mathcal{K}_D(d=2) \simeq \mathbb{Z}$ of two dimensional topological superconducts in class D to the classifying group two dimensional topological superconductors in class DIII $\mathcal{K}_{DIII}(d=2) \simeq \mathbb{Z}_2$. Under this homomorphism, the Majorana bound state at a vortex is mapped to a Kramers pair of Majorana bound states at a vortex. This procedure shows that the two dimensional superconductor in class DIII hosts Kramers pairs of Majorana bound states at vortices, in agreement with Ref. [71].

By applying the dimensional raising maps from Refs. [47, 75, 76] to the two dimensional first order topological superconductors in class D and class DIII, one shows that $d-2$ dimensional defects binding a $\pi$-flux in all related symmetry classes and dimensions host $d-2$ dimensional anomalous states. The dimensional raising maps starting from Cartan classes with a chiral antisymmetry were reviewed and applied in Appendix B.4. The dimensional raising map preserves the existence of $d-2$ dimensional states at point defects as long as all crystalline symmetries, if present, act trivially in the newly added dimensions. This was exemplified in Sec. 6.3 and was shown using a dimensional reduction scheme based on the scattering matrix in Ref. [11].

### D.3 Presence of internal unitary symmetries

First-order topological phases protected by an internal unitary symmetry $\mathcal{U}$ may also host anomalous states at a $\mathcal{U}$-symmetry flux defect [83]. These defects are defined as the end of a branch cut upon which a crossing particle is acted upon by the symmetry $\mathcal{U}$. The presence of the symmetry flux thus is another property of point defects in addition to the $\mathbb{Z}_2$ geometric $\pi$-flux and the rotation and translation holonomies that needs to be specified when constructing a disclination. The construction of a lattice with $\mathcal{U}$-symmetry flux is similar as discussed in Sec. 2.5. Therefore, similar conditions hold for the algebraic relations on the symmetry elements in order to ensure the absence of a domain wall that allows a unique prediction of the existence of anomalous states at the $\mathcal{U}$-symmetry flux defect.

Furthermore, the classification of first-order topological phases in the presence of additional unitary symmetries follows by block diagonalizing the Hamiltonian under the irreducible representations of the unitary symmetries and identifying the Cartan class and relations between each block [67, 69].

An example is a two-dimensional topological superconductor in Cartan class D with a $\mathbb{Z}_2$ unitary internal symmetry $\mathcal{U}$ with $\mathcal{U}^2 = 1$ that commutes with particle-hole conjugation. In this case the Hamiltonian can be block-diagonalized with respect to the two eigenvalues $\pm 1$ of $\mathcal{U}$ and the two blocks individually are in Cartan class D and can be characterized by a Chern number $Ch_{\pm}$ where the subscript $\pm$ denotes the block. One can identify a generator of a $\mathbb{Z}_2$ symmetry enriched topological phase as a Hamiltonian with $Ch_+ = 1$ and $Ch_- = 0$ and a generator of the $\mathbb{Z}_2$-symmetry protected topological phase as a Hamiltonian with $Ch_- = -Ch_+ = 1$.

In this example, one can distinguish a geometric $\pi$-flux from a $\mathcal{U}$-symmetry flux: the geometric $\pi$-flux is defined by introducing a branch cut through the system such that each particle crossing the branch cut acquires a $\pi$ phase shift. In contrast, the $\mathcal{U}$-symmetry flux is defined by introducing a branch cut such that each particle crossing it is acted upon by $\mathcal{U}$. The geometric $\pi$-flux defect hosts a number of Majorana fermions that is given by the parity of the total Chern number $\theta^{flux} = Ch_+ + Ch_-$ mod 2. The $\mathcal{U}$-symmetry flux hosts a number of Majorana fermions $\theta^{\mathcal{U}} = Ch_-$ mod 2 given by the parity of the Chern number in the $-1$ block only. This can be seen as in the block-diagonal basis of $\mathcal{U}$, the $\mathcal{U}$-symmetry flux contributes a $\pi$ phase shift only in the $-1$ subspace while it acts trivially in the $+1$ subspace.

As for our crystalline examples, each property of the defect is associated with exactly one generator of the topological phases that contributes anomalous states to the defect: only the generator of the symmetry-enriched topological phase with $Ch_+ = 1$ and $Ch_- = 0$ contributes a

Majorana bound state to the geometric $\pi$-flux defect, and only the generator of the $\mathbb{Z}_2$ internal-symmetry protected topological phase with $\text{Ch}_- = -\text{Ch}_+ = 1$ contributes a Majorana bound state to the $\mathcal{U}$-symmetry flux defect.

## E Symmetry-based indicators for two dimensional superconductors in class D

Symmetry-based indicators are easy-to-compute topological invariants for topological crystalline insulators and superconductors that are expressed using the matrix-valued single-particle Hamiltonian $H(\boldsymbol{k}_s)$ at a certain set of high-symmetry momenta $\boldsymbol{k}_s$. The symmetry-based indicators for a Cartan class D superconductor with twofold and fourfold rotation symmetry have been derived in Ref. [87]. The symmetry-based indicators of Ref. [87] are in one-to-one correspondence to the rotation invariants of Ref. [54]. Here, we show how the symmetry-based indicators can be used to formulate a criterion on the existence of anomalous disclination states in terms of the bulk topological invariants.

*Fourfold rotation symmetry.* Anomalous states at point defects exist only for pairing symmetries $u(R_{\pi/2})\Delta(R_{\pi/2}\boldsymbol{k})u^\dagger(R_{\pi/2}) = \pm\Delta(\boldsymbol{k})$. For the other pairing symmetries, there are no topological phases that can host zero-dimensional Majorana defect states (see Table 1 and Appendix F.2). Below we present the result for even pairing symmetry $u(R_{\pi/2})\Delta(R_{\pi/2}\boldsymbol{k})u^\dagger(R_{\pi/2}) = \Delta(\boldsymbol{k})$. There are two symmetry-based indicators,

$$z_{1;x,y} = \mathfrak{N}_{\frac{1}{2}}^{(\pi,0)} + \mathfrak{N}_{\frac{1}{2}}^{(\pi,\pi)} + \mathfrak{N}_{\frac{5}{2}}^{(\pi,\pi)} \mod 2 \tag{39}$$

and

$$z_2 = -\mathfrak{N}_{\frac{1}{2}}^{(0,0)} + 3\mathfrak{N}_{\frac{5}{2}}^{(0,0)} - 2\mathfrak{N}_{\frac{1}{2}}^{(\pi,0)} \tag{40}$$
$$+ 3\mathfrak{N}_{\frac{1}{2}}^{(\pi,\pi)} - \mathfrak{N}_{\frac{5}{2}}^{(\pi,\pi)} \mod 8.$$

Here, $\mathfrak{N}_j^{\boldsymbol{k}_s}$ is the number of eigenstates with negative eigenenergy and eigenvalues $e^{ij2\pi/n}$ under rotation symmetry of the Hamiltonian $H(\boldsymbol{k}_s)$ at the high symmetry momentum $\boldsymbol{k}_s$ with $n$-fold rotation symmetry. The symmetry-based indicator $z_{1;x,y}$ detects the weak topological superconductor. The elements $z_2 \mod 8$ correspond to a Chern superconductor. The element "4" $\in \mathbb{Z}_8$ is ambiguous and may either correspond to the second-order topological superconductor or to a Chern superconductor with Chern number $Ch = 4$.

Due to the ambiguity of the second-order topological superconductor with the Chern superconductor, it is impossible to define a necessary criterion on the existence of Majorana bound states at a disclination purely in terms of topological band labels. However, a criterion can be formulated assuming the Chern number $Ch$ can be explicitly determined. The number of Majorana bound states at a disclination with Frank angle $\Omega$ and translation holonomy $\boldsymbol{T}$ is

$$\theta = \frac{\Omega}{2\pi}(z_2 - Ch) + \boldsymbol{T} \cdot \boldsymbol{G}_\nu \mod 2, \tag{41}$$

where $\boldsymbol{G}_\nu = (z_{1;x,y}, z_{1;x,y})^T$ is the weak invariant and $\boldsymbol{T}$ is the translation holonomy of the disclination.

*Twofold rotation symmetry.* Anomalous states at point defects exist only for even pairing symmetry $u(R_\pi)\Delta(R_\pi\boldsymbol{k})u^\dagger(R_\pi) = \Delta(\boldsymbol{k})$, see Table 1 and Appendix F.2. There are three

symmetry-based indicators. The first two,

$$z_{1;x} = \mathfrak{N}_{\frac{1}{2}}^{(\pi,0)} + \mathfrak{N}_{\frac{1}{2}}^{(\pi,\pi)} \mod 2, \tag{42}$$

$$z_{1;y} = \mathfrak{N}_{\frac{1}{2}}^{(0,\pi)} + \mathfrak{N}_{\frac{1}{2}}^{(\pi,\pi)} \mod 2, \tag{43}$$

correspond to weak topological superconductors with topological crystal limits shown in Fig. 6 (a), (b) in the main text, respectively. Furthermore, there is a symmetry-based indicator

$$z_2 = \mathfrak{N}_{\frac{1}{2}}^{(0,0)} - \mathfrak{N}_{\frac{1}{2}}^{(0,\pi)} - \mathfrak{N}_{\frac{1}{2}}^{(\pi,0)} + \mathfrak{N}_{\frac{1}{2}}^{(\pi,\pi)} \mod 4, \tag{44}$$

whose odd elements detect the parity of the Chern number and the value $z_2 = 2$ is ambiguous between a Chern superconductor with even Chern number and a second-order topological superconductor. The number of Majorana bound states at a disclination with Frank angle $\Omega$ and translation holonomy $\boldsymbol{T}$ can be determined as

$$\theta = \frac{\Omega}{2\pi}(z_2 - Ch) + \boldsymbol{T} \cdot \boldsymbol{G}_v \mod 2, \tag{45}$$

where $\boldsymbol{G}_v = (z_{1;x}, z_{1;y})^T$ is the weak invariant.

## F  Derivation of tables 1 and 2

This section shows how the classification of strong first order, strong rotation symmetry protected second-order, and weak topological phases summarized in tables 1 and 2 can be obtained using our results from C.2 and similar arguments as in the examples Sec. 6. In addition, we briefly review the symmetry classification of superconducting order parameters in Sec. F.1 as it determines the topological classification. A complete discussion can be found in Ref. [87].

In Appendix C.2, we derived simple criteria for the existence of rotation symmetry protected second-order and weak topological phases. In particlar, we showed that a sufficient criterion for the existence of a second-order topological phase with fourfold and sixfold rotation symmetry is the existence of a second-order topological phase with twofold rotation symmetry with representation $U(R_\pi) = U(R_{\pi/2})^2$ and $U(R_\pi) = U(R_{\pi/3})^3$, respectively. For sixfold symmetric second-order topological phases on a lattice and for weak topological phases in fourfold symmetric lattices this criterion is also necessary. All entries have been verified by checking that i) the required hybridization terms to gap out the anomalous states in the topological crystal construction exist or ii) a tight binding model realizing the topological crystalline phase in question can be explicitly defined.

### F.1  Symmetry of the superconducting order parameter

The classification of topological crystalline superconductors depends on the symmetry of superconducting order parameter, as the explicit examples in sections F.2 and F.3 below show. The following is a brief summary of the extensive discussion in Ref. [87].

The BdG Hamiltonian describing superconducting systems is of the form

$$H_{BdG}(\boldsymbol{k}) = \begin{pmatrix} h(\boldsymbol{k}) & \Delta(\boldsymbol{k}) \\ \Delta(\boldsymbol{k})^\dagger & -h^*(-\boldsymbol{k}) \end{pmatrix}, \tag{46}$$

where $\Delta(\boldsymbol{k}) = -\Delta^T(-\boldsymbol{k})$ is the superconducting order parameter and $h(\boldsymbol{k})$ is the normal state single particle Hamiltonian. The BdG Hamiltonian satisfies a particle-hole antisymmetry

$H_{\mathrm{BdG}}(\boldsymbol{k}) = -\tau_1 H_{\mathrm{BdG}}(-\boldsymbol{k})^*\tau_1$ where $\tau$ are Pauli matrices in particle-hole space. The symmetry of the superconducting order parameter $\Delta(\boldsymbol{k})$ can be characterized by a one-dimensional representation $\Theta$ of the point group [87, 88] as

$$\Delta(\boldsymbol{k}) = u(g)\Delta(g\boldsymbol{k})u^\dagger(g)\Theta^*(g), \tag{47}$$

where $u(g)$ is the representation of the point group element $g$ on the normal-state Hamiltonian $h(\boldsymbol{k})$. The corresponding representation on the Bogoliubov-de Gennes Hamiltonian is

$$U(g) = \begin{pmatrix} u(g) & 0 \\ 0 & u^*(g)\Theta(g) \end{pmatrix}. \tag{48}$$

With this representation one finds directly the commutation relation between particle-hole antisymmetry and the point group elements

$$g\mathcal{P} = \Theta(g)\mathcal{P}g. \tag{49}$$

For point groups generated by a single $n$-fold rotation symmetry the one dimensional representation $\theta(g)$ is entirely specified by the phase $\phi$ of the generating element $e^{i\phi} = \theta(R_{2\pi/n})$.

## F.2 Two dimensions

### F.2.1 Class D

*Twofold rotation.* The existence of the mass term guaranteeing the existence of weak and second-order topological phases for both pairing symmetries characterized by the phase $\phi = 0$ and $\pi$ follows from results of Refs. [11, 12], as detailed in the last paragraph of Sec. C.2. For concreteness, for $\phi = 0$ we may choose the representations $U(\mathcal{R}_\pi) = i\tau_2$, $\mathcal{P} = K$ where the mass term that gaps a pair of symmetry related Majorana bound states is $\tau_2$. For $\phi = \pi$ we may choose the representations $U(\mathcal{R}_\pi) = i\tau_2$, $\mathcal{P} = \tau_3 K$ where no mass term exists. Refs. [11, 12] also show that a Chern superconductor with odd Chern number exists only for $\phi = 0$ while for $\phi = \pi$, the Chern number is constrained to be even.

*Fourfold rotation.* The topological classification of superconductors with pairing symmetry characterized by the phase $\phi = 0$ and $\pi$ ($\phi = \pi/2$ and $3\pi/2$) are identical as they are related by a multiplication of the representation of rotation symmetry with a phase [87]. For $\phi = 0, \pi$, the weak and second-order phases exist as the Majorana bound states at the rotation axes gap out in twofold rotation symmetry related pairs. For $\phi = \pi/2, 3\pi/2$ Ref. [87] has shown that no strong second-order topological phase and no weak phase exists and the Chern number is constrained to be even.

*Sixfold rotation.* Similar as for fourfold rotation, the topological classification of superconductors with pairing symmetry characterized by the phase $\phi = 0, 2\pi/3$ and $4\pi/3$ ($\phi = \pi, \pi/3$ and $5\pi/3$) are identical. A second-order topological phase exists only for $\phi = 0, 2\pi/3$ and $4\pi/3$. For $\phi = 0$, a superconductor with Chern number $Ch = 1$ is given by the $p_x + ip_y$ superconductor [54],

$$H(\boldsymbol{k}) = \sum_{i=1}^{3} \sin(\boldsymbol{k}\cdot\boldsymbol{a}_1)\boldsymbol{a}_1\cdot\boldsymbol{\tau} + \cos(\boldsymbol{k}\cdot\boldsymbol{a}_1)\tau_3, \tag{50}$$

with $\boldsymbol{\tau} = (\tau_x, \tau_y)^T$ and $\boldsymbol{a}_1 = (1,0)$, $\boldsymbol{a}_2 = (-\frac{1}{2}, \frac{\sqrt{3}}{2})$ and $\boldsymbol{a}_3 = (-\frac{1}{2}, -\frac{\sqrt{3}}{2})$ and representation of sixfold rotation symmetry $U(\mathcal{R}_{\pi/3}) = e^{i\pi\tau_3/6}$ and particle-hole antisymmetry $\mathcal{P} = \tau_1 K$. For $\phi = \pi$ a superconductor with odd Chern number does not exist. In this symmetry class, a model for a superconductor with Chern number $Ch = 2$ can be defined as

$$H(\boldsymbol{k}) = \sum_{i=1}^{3} \sin(\boldsymbol{k}\cdot\boldsymbol{a}_1)\boldsymbol{a}_1\cdot\boldsymbol{\tau}\rho_0 + \cos(\boldsymbol{k}\cdot\boldsymbol{a}_1)\tau_3\rho_0, \tag{51}$$

with $\boldsymbol{\tau} = (\tau_x, \tau_y)^T$ and $a_1 = (1, 0)$, $a_2 = (-\frac{1}{2}, \frac{\sqrt{3}}{2})$ and $a_3 = (-\frac{1}{2}, -\frac{\sqrt{3}}{2})$ and representation of sixfold rotation symmetry $U(\mathcal{R}_{\pi/3}) = e^{i\pi\tau_3/6}\rho_3$ and particle-hole antisymmetry $\mathcal{P} = \tau_1\rho_1 K$.

*Magnetic twofold rotation.* A magnetic rotation symmetry consists of the combined action of rotation and time-reversal symmetry. For spinful fermions, these operators commute. Thus, for magnetic twofold rotation symmetry we have $(U(\mathcal{R}_\pi\mathcal{T})K)^2 = 1$. A pair of Majorana bound states related by $\mathcal{R}_\pi\mathcal{T}$ with $U(\mathcal{R}_\pi\mathcal{T}) = \tau_1$ gaps upon hybridization with the mass term $\tau_2$. As a consequence, the second-order and weak topological phase exist. Twofold magnetic rotation symmetry requires the Chern number to vanish []. 

*Magnetic fourfold rotation.* For magnetic fourfold rotation symmetry, the operator $\left(U(\mathcal{R}_{\pi/2}\mathcal{T})K\right)^2$ is a unitary twofold rotation operator that commutes with particle-hole anti-symmetry. Furthermore, we have $\left(U(\mathcal{R}_{\pi/2}\mathcal{T})K\right)^4 = -1$ for spinful fermions. Here, Majorana bound states hybridize in twofold rotation symmetry related pairs. Thus a weak and second-order topological phase exists. Furthermore, magnetic fourfold rotation prohibits the existence of a Chern number.

*Magnetix sixfold rotation.* A system with magnetic sixfold rotation symmetry also satisfies magnetic twofold rotation symmetry. We have shown that for the latter, Majorana bound states hybridize in pairs and the Chern number needs to vanish. Thus only the second-order topological phase exists.

### F.2.2 Class DIII

*Pairing symmetry $\phi = 0$.* Superconductors in class DIII with $\phi = 0$ can be constructed from the class D superconductors with $\phi = 0$ by taking two time reversed copies as shown in Sec. 6.2. This construction shows the existence of weak and strong first and second-order topological phases for twofold, fourfold and sixfold rotation symmetry.

*Pairing symmetry $\phi = \pi$.* For this pairing symmetry, Kramers pairs of Majorana bound states at rotation axes with representations $\mathcal{T} = i\sigma_2\tau_0 K$, $\mathcal{P} = \sigma_0\tau_3 K$ hybridize in partners related by twofold rotation $U(\mathcal{R}_\pi) = i\sigma_3\tau_1$ as the mass term $\sigma_1\tau_2$ exists. In systems with fourfold rotation symmetry with $\phi = \pi$, the representation of twofold rotation symmetry commutes with particle-hole symmetry. With the results from the previous paragraph on the pairing symmetry $\phi = 0$, also at fourfold rotation axes the Kramers pairs of Majorana bound states hybridize in partners related by twofold rotation symmetry. Thus, weak and second-order phases exist with twofold, fourfold and sixfold rotation symmetry. The first order topological phase is prohibited by $n$-fold rotation symmetry with pairing symmetry $\phi = \pi$.

Furthermore, with $\phi = \pi$ rotation symmetry anticommutes with particle-hole antisymmetry. As argued in Sec. 2.5, in this case it is impossible to construct a finite hopping across the cut that preserves all symmetries and is locally indistinguishable from the bulk. This implies that there is in general no bulk-defect correspondence at disclinations for pairing symmetry $\phi = \pi$.

### F.2.3 Classes AIII and BDI

In class AIII, the zero dimensional anomalous states have $\mathbb{Z}$ topological charge as zero-energy states with the same eigenvalue under chiral antisymmetry $\Gamma$ do not gap out in pairs. There-fore, a set of symmetry related zero-energy states can only gap out if the rotation symmetry anticommutes with chiral antisymmetry such that the eigenvalue of chiral antisymmetry of symmetry related zero-energy states is opposite. In this case, twofold rotation symmetry re-lated states gap out with a mass term $\tau_2$ with representations $U(\mathcal{R}_\pi) = \tau_2$ and $\Gamma = \tau_3$. As a consequence, second-order topological phases exist for twofold, fourfold and sixfold rota-tion symmetric lattices. Weak topological phases exist in twofold rotation symmetric lattices. There does not exist a first order topological phase in class AIII in two dimensions.

The same result holds for class BDI in two dimensions as the mass term also satisfies the antiunitary symmetry $\mathcal{T} = \tau_3 K$.

### F.2.4 Class CII

Class CII has time-reversal symmetry $\mathcal{T}^2 = -1$ and particle-hole antisymmetry $\mathcal{P}^2 = -1$. Similar arguments as in class AIII hold here, except that the zero-energy states appear in Kramers pairs. Here, a mass term gapping twofold rotation symmetry related partners can be chosen as $\sigma_3 \tau_2$ with representations $\mathcal{T} = i\sigma_2 \tau_0 K$, $\mathcal{P} = \sigma_0 \tau_2 K$ and $U(\mathcal{R}_\pi) = \sigma_3 \tau_2$.

## F.3 Three dimensions

In three dimensions, the $d-1$ dimensional anomalous states with $\mathbb{Z}$ topological charge are chiral states. Their topological charge (i.e. their propagation direction) is only inverted by magnetic rotation symmetry. As a consequence, there are no weak or second-order topological phases in classes A, D, C with non-magnetic rotation symmetry. Furthermore, these Cartan classes also do not host first order topological phases.

### F.3.1 Classes A and D

In class A, chiral modes related by twofold magnetic rotation symmetry gap out in pairs, as the mass term $\tau_2$ for two chiral modes related by twofold magnetic rotation symmetry $\mathcal{R}_\pi \mathcal{T} = \tau_1 K$ described by the low energy Hamiltonian $vk_z \tau_3$ exists. For fourfold magnetic rotation symmetry, it has been shown in Sec. B.3 that the mass term $\tau_3 \sigma_2 - \tau_2 \sigma_0$ describing a ring hybridization of rotation symmetry related chiral modes with low energy Hamiltonian $vk_z \tau_3 \sigma_3$ creates a gap in the spectrum. As a consequence, second-order topological phases exist in lattices with twofold, fourfold and sixfold magnetic rotation symmetry. The weak topological phases as shown in Fig. 6 in the main text exist only in lattices with twofold rotation symmetry.

The mass terms and low energy Hamiltonians also satisfy particle-hole antisymmetry $\mathcal{P} = K$. Thus the results apply also to class D.

The weak phases corresponding to stacks of Chern insulators with stacking direction parallel to the rotation axis exist in class A with non-magnetic rotation symmetry as the Chern number is consistent with rotation symmetry. Magnetic rotation symmetry requires the Chern number to vanish. In class D, the results from section F.2 apply.

### F.3.2 Class C

We regard physical systems in class C as superconductors in the presence of spin rotation symmetry. In this case, magnetic twofold rotation symmetry still satisfies $(\mathcal{R}_\pi \mathcal{T})^2 = 1$. With $\mathcal{R}_\pi \mathcal{T} = \tau_0 \rho_1 K$ and $\mathcal{P} = \tau_2 \rho_0 K$, the mass term $\tau_0 \rho_2$ gaps out the low energy Hamiltonian $vk_z \tau_0 \rho_3$ describing the minimal number of chiral modes related by magnetic twofold rotation symmetry. In class C, fourfold magnetic rotation symmetry satisfies $(\mathcal{R}_{\pi/2} \mathcal{T})^4 = 1$. Here a ring hybridization with real hopping elements gaps symmetry related chiral modes. This shows the existence of the weak and second-order topological phases as shown in Fig. 6 in the main text in lattices with twofold, fourfold and sixfold magnetic rotation symmetry in Cartan class C.

Two dimensional superconductors in class C allow for an even Chern number. The weak phases corresponding to stacks of Chern superconductors with even Chern number with stacking direction parallel to the rotation axis exist with non-magnetic rotation symmetry for any pairing symmetry.

### F.3.3 Class DIII

*Pairing symmetry $\phi = 0$.* Three dimensional superconductors in class DIII with $\phi = 0$ can be constructed from two dimensional class D superconductors with $\phi = 0$ using the dimensional raising map as we also used in Appendix B.4. It has been shown using the reflection matrix dimension reduction scheme in Ref. [11] that the dimensional raising map preserves anomalous states at defects.

Below we illustrate the usage of the dimensional raising map to define a model Hamiltonian for the second-order topological superconductor in class DIII in three dimensions.

In the model described by the Hamiltonian $H_D$ in Eq. (13) the dimerization parameter (or mass) $\delta t$ can be used to tune between the trivial and the topological phase. In particular, the system undergoes a topological phase transition at $\delta t = 0$, where the bulk gap closes. The dimensional raising map requires to replace $\delta t \rightarrow \delta t \cos k_z$, such that the model interpolated between the trivial and the topological phase as a function of the additional momentum parameter $k_z$. In order to ensure that the system remains gapped for all $k_z$ one adds another term $t_z \sin k_z \gamma_z$ to the Hamiltonian, requiring that $\gamma_z$ anticommutes with the Hamiltonian at $k_z = 0, \pi$. Starting from a model without chiral antisymmetry one needs to introduce a new degree of freedom described by Pauli matrices $\sigma$ by taking the original model $H_D \rightarrow H_D s_3$. Then we may choose $\gamma_z = \sigma_3$. Now the model satisfies an additional chiral antisymmetry $\Gamma = \sigma_2$ and, in combination with particle-hole antisymmetry, a time-reversal symmetry $\mathcal{T} = i\sigma_2 K$. Thus we may interpret the $\sigma_3$ degree of freedom as spin.

These arguments are collected in the dimensional raising map, which is expressed as

$$
\begin{aligned}
H_D(k_x, k_y; \delta t) \;\rightarrow\; & H_D(k_x, k_y; \delta t \cos[k_z])\sigma_3 \\
& + t_z \sin[k_z]\tau_0\rho_0\sigma_1.
\end{aligned}
$$

This lifts our two-dimensional model in class D to a three-dimensional model in class DIII realizing a strong second-order topological phase.

*Pairing symmetry $\phi = \pi$.* Similar to two dimensional class DIII superconductors with pairing symmetry $\phi = \pi$, helical Majorana modes $v k_z \sigma_3 \tau_0$ related by twofold rotation $U(\mathcal{R}_\pi) = i\sigma_3 \tau_1$ and representations $\mathcal{T} = i\sigma_2\tau_0 T$, $\mathcal{P} = \sigma_0\tau_3 K$ hybridize as the mass term $\sigma_1\tau_2$ exists. Weak and second-order phases exist with twofold, fourfold and sixfold rotation symmetry. The first order topological phase is prohibited by $n$-fold rotation symmetry with pairing symmetry $\phi = \pi$. These results are consistent with Ref. [67].

The existence of weak phases corresponding to stacks of two-dimensional first order topological superconductors with stacking direction parallel to the rotation axis follows from the results of section F.2.

### F.3.4 Class AII

The classification of time reversal symmetric insulators in class AII is related to class DIII by lifting the particle-hole antisymmetry constraint of class DIII. This construction maps the first order topological phases in $d = 2, 3$ and corresponding anomalous states of class DIII to the corresponding first order topological phases and anomalous states in class AII [31]. Applying this construction to class DIII topological superconductors with pairing symmetry $\phi = 0$ shows the existence of first-order, second-order and weak topological phases in class AII.

Weak phases corresponding to stacks of two-dimensional first order topological insulators with stacking direction parallel to the rotation axis exist with twofold rotation symmetry.

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
