# Peer review of "Bulk-boundary-defect correspondence at disclinations in rotation-symmetric topological insulators and superconductors"

_SciPost Physics, doi:SciPost Phys. 10, 092 (2021)_

## Round 2 · Referee Report · Anonymous (Referee 1) · 2020-9-27

Strengths

1) Adequately written 2) Timely subject 3) Exposed theory matched by quite some intuitive examples

Weaknesses

Retrieves quite some known results affecting the originality/novelty a bit

Report

In this work, Geier et al consider the effect of lattice defects, in particular disinclinations, in rotationally symmetric free fermion topological phases. This is an old subject. Indeed, for the SSH model -although one would call this now a symmetry obstructed band topology- zero modes were directly tied via an index theorem to the underlying topology. This directly links to well known Jackiw-Rebbi solutions. More generally, such index theorems have been considered in the context of various systems that have the possibility of hosting Dirac Fermions. Defects, when assumed to act locally, can have the same type of monodromy effect, mimicking closely the Callan-Harvey mechanism used to describe the “conventional” bulk boundary correspondence in free-fermion topological phases of matter. This is a timely subject due to the active research into crystalline protected topological band theory. In this regard the authors also consider e.g. higher order topological phases.

The paper is nicely written and interesting. Although the authors retrieve some previous results, the paper is self-contained and also has new discussions and examples. Hence, the paper may meet the criteria for publication. Before deciding on this matter, I like to discuss a few points nonetheless.

Some comments/discussions.

a)Although I value the freedom of authors to phrase their text and title to their liking, there are a two aspects I like to discuss in this case;

1] In the first sentence of the abstract the authors say they “establish” a link. As witnessed by the the main text, the authors refer to many papers on the subject. Hence the word establish, having the connotation of being first, is not completely fair in my eyes, affecting especially readership interested only in the abstract. Hence I suggest to use e.g, consider. 2]Related, I urge the authors to change their title. Given the many works and the fact that this focusses on C_n rotation characterized phases, this should be reflected better in the title. As it stands now it appears a general correspondence is found.

b)In sec 2.1 I like to point out that monodromy defects, apart from ref 54 are known to characterize, SPT states generally, see also e.g. https://journals.aps.org/prb/abstract/10.1103/PhysRevB.86.115109

c)In Eq. 1 the holonomy quotes the results of Eq. 81 in ref 54, it is important to note that the translations have been modded out. The authors refer to rotation anomaly above, but considering only Eq.1 and it notation this can be tricky.

d)A main point is the terminology on strong topology. In sec 2.1 the authors state “A strong topological phase remains unaffected when translation symmetries are broken.” In principle one can define of course whatever one wants, but in case of TRS strong means that one does not need a translational symmetry axes to define weak indices. As class AII has a Z_2 invariant in 2D (and) 3D, a layered construction makes sense, i.e. the product over the pfaffian of the sewing matrix is gauge invariant in each plane. But translational symmetry is tricky. For topological (band theory) a lattice is implicitly assumed, c.f. also first sentence of sec 4.1- although here it is clear that its role has not been used. And this implicit presence is even more so assumed for crystalline invariants. Hence strong seems to mean that the topological protection comes from a 2D subgroup in e.g. a plane in 3D, the translations the authors refer to denote the preserved perpendicular direction. I think formulating it from this angle is clearer than stating that translations can be broken as implicitly it can only be perturbatively broken or the crystalline symmetries have to preserved on average. In any case there is an implicit sense of the underlying lattice and hence translation symmetry. In this regard, one should also note that for a weak phase e.g. doubling the unit cell may trivialize the phase, then the translations are altered but there is translational symmetry in every direction. Hence, my request to formulate this slightly different.

e)Purely out of interest. There is isomorphism between class A and AIII, which is particular reflected in the K-theory and according constraints of rotation invariants; https://journals.aps.org/prx/pdf/10.1103/PhysRevX.7.041069. Specifically it relates a piece in class A to a piece in class A \oplus a piece in class AIII. Maybe this is reflected in the relation in adding a dimension and stacking. My question is whether the authors saw some of these relations specifically, see also Appendix A.

f)Sec 5.3 can be improved a bit to my taste. Basically, in the appendix and the examples of sec 6 the relation to the presence of extra symmetries is hinted upon. But apart from enumerating the results in the table, the discussion should be expanded. Relating the different symmetry classes is generally not so straightforward. Indeed spinless 1D E irreps of C_4 or C_6 can e.g. glue together into 2D reps that can give some form of topology, showing the crucial role of spinless TRS in this case and more generally.

Requested changes

see report

  • validity: high
  • significance: good
  • originality: good
  • clarity: high
  • formatting: excellent
  • grammar: excellent

Author:  Max Geier  on 2021-02-12  [id 1232]

(in reply to Report 1 on 2020-09-27)
Category:
remark
answer to question
validation or rederivation

We thank the referee for the positive report and for the constructive criticism. Below we address all comments point by point. We have provided clarifications in the manuscript in response to the comments. We believe that thanks to the referees suggestions, we were able to further improve the quality of our manuscript.

** First referee: **

In this work, Geier et al consider the effect of lattice defects, in particular disinclinations, in rotationally symmetric free fermion topological phases. This is an old subject. Indeed, for the SSH model -although one would call this now a symmetry obstructed band topology- zero modes were directly tied via an index theorem to the underlying topology. This directly links to well known Jackiw-Rebbi solutions. More generally, such index theorems have been considered in the context of various systems that have the possibility of hosting Dirac Fermions. Defects, when assumed to act locally, can have the same type of monodromy effect, mimicking closely the Callan-Harvey mechanism used to describe the “conventional” bulk boundary correspondence in free-fermion topological phases of matter. This is a timely subject due to the active research into crystalline protected topological band theory. In this regard the authors also consider e.g. higher order topological phases.

The paper is nicely written and interesting. Although the authors retrieve some previous results, the paper is self-contained and also has new discussions and examples. Hence, the paper may meet the criteria for publication. Before deciding on this matter, I like to discuss a few points nonetheless.

** Our response: ** We thank the referee for the positive report and for putting our work in context.

** First referee: **

Some comments/discussions. a)Although I value the freedom of authors to phrase their text and title to their liking, there are a two aspects I like to discuss in this case; 1$]$ In the first sentence of the abstract the authors say they “establish” a link. As witnessed by the the main text, the authors refer to many papers on the subject. Hence the word establish, having the connotation of being first, is not completely fair in my eyes, affecting especially readership interested only in the abstract. Hence I suggest to use e.g, consider.

** Our response: ** We agree with the suggestion and have changed the word "establish" to "study" in our abstract.

** First referee: **

2$]$Related, I urge the authors to change their title. Given the many works and the fact that this focusses on $C_n$ rotation characterized phases, this should be reflected better in the title. As it stands now it appears a general correspondence is found.

** Our response: ** We thank the Referee for this suggestion and have accordingly changed our title to "Bulk-boundary-defect correspondence at disclinations in rotation symmetric topological insulators and superconductors". Originally, we had hoped that specifying disclinations in the title would be sufficient to indicate the relevance of rotation symmetry, but we are happy to emphasize the rotation symmetry by including it explicitly in the title.

** First referee: **

b)In sec 2.1 I like to point out that monodromy defects, apart from ref 54 are known to characterize, SPT states generally, see also e.g. https://journals.aps.org/prb/abstract/10.1103/PhysRevB.86.115109

** Our response: ** We thank the referee for pointing us to this reference and added a citation in the introduction of our manuscript.

** First referee: **

c)In Eq. 1 the holonomy quotes the results of Eq. 81 in ref 54, it is important to note that the translations have been modded out. The authors refer to rotation anomaly above, but considering only Eq.1 and it notation this can be tricky.

** Our response: ** We thank the referee for this comment. Also in response to the specific comments 1) and 2) of the third referee, we have included a brief summary of the derivation of the inequivalent holonomy groups as a new section in the Appendix (new Appendix A). We hope that this summary also clarifies the importance of taking the equivalence relation of all starting points that can be reached by a translation of the starting point of the loop or a rotation of the initial coordinate system.

** First referee: **

d)A main point is the terminology on strong topology. In sec 2.1 the authors state “A strong topological phase remains unaffected when translation symmetries are broken.” In principle one can define of course whatever one wants, but in case of TRS strong means that one does not need a translational symmetry axes to define weak indices. As class AII has a $Z_2$ invariant in 2D (and) 3D, a layered construction makes sense, i.e. the product over the pfaffian of the sewing matrix is gauge invariant in each plane. But translational symmetry is tricky. For topological (band theory) a lattice is implicitly assumed, c.f. also first sentence of sec 4.1- although here it is clear that its role has not been used. And this implicit presence is even more so assumed for crystalline invariants. Hence strong seems to mean that the topological protection comes from a 2D subgroup in e.g. a plane in 3D, the translations the authors refer to denote the preserved perpendicular direction. I think formulating it from this angle is clearer than stating that translations can be broken as implicitly it can only be perturbatively broken or the crystalline symmetries have to preserved on average. In any case there is an implicit sense of the underlying lattice and hence translation symmetry. In this regard, one should also note that for a weak phase e.g. doubling the unit cell may trivialize the phase, then the translations are altered but there is translational symmetry in every direction. Hence, my request to formulate this slightly different.

** Our response: ** We thank the referee for these important remarks. We agree with the referees points on the definition of strong topology for topological band theory. We have changed the respective paragraph in section 3.1, and included a footnote elaborating on the relation to topological band theory.

** First referee: **

e)Purely out of interest. There is isomorphism between class A and AIII, which is particular reflected in the K-theory and according constraints of rotation invariants; \url{https://journals.aps.org/prx/pdf/10.1103/PhysRevX.7.041069}. Specifically it relates a piece in class A to a piece in class A $\oplus$ a piece in class AIII. Maybe this is reflected in the relation in adding a dimension and stacking. My question is whether the authors saw some of these relations specifically, see also Appendix A.

** Our response: ** Regarding the constraints on rotation invariants, the discussion in Ref. [PRX 7, 041069] is similar to the derivation of the symmetry-based indicators in Ref. [PRB 101, 245128], which we cited for our formulas for the symmetry-based indicators contained in our manuscript. In particular, Ref. [PRB 101, 245128] contains a discussion on the definition of the rotation invariants in all Cartan classes, and on how to derive all constraints on the rotation invariants in a given symmetry class. For classes A, AIII the discussion of [PRB 101, 245128] is in agreement with [PRX 7, 041069].

In Ref. [PRX 7, 041069], the authors state that "there exists an isomorphism relating the K-theory computation for class A in three dimensions to two pieces in two dimensions, one of which corresponds to class A while the other corresponds to class AIII". Unfortunately, we did not find a precise definition of the isomorphism or a corresponding reference in Ref. [PRX 7, 041069], such that we are not sure how the suggested isomorphism in Ref. [PRX 7, 041069] would be defined. However, this isomorphism seems to be very similar to the isomorphism we present in App. B (former App. A), relating 2d class A/AIII and 3d class AIII/A. The difference between our isomorphism and the suggested isomorphism of Ref. [PRX 7, 041069] is that our isomorphism maps a 3d system in Cartan class A or AIII to a 2d system in Cartan class AIII or A, but no "two pieces in two dimensions, one of which corresponds to class A while the other corresponds to class AIII" are obtained.

** First referee: **

f)Sec 5.3 can be improved a bit to my taste. Basically, in the appendix and the examples of sec 6 the relation to the presence of extra symmetries is hinted upon. But apart from enumerating the results in the table, the discussion should be expanded. Relating the different symmetry classes is generally not so straightforward. Indeed spinless 1D E irreps of $C_4$ or $C_6$ can e.g. glue together into 2D reps that can give some form of topology, showing the crucial role of spinless TRS in this case and more generally.

** Our response: ** We thank the referee for the suggestion. We agree that it is not trivial to obtain the results presented in the table. Therefore, we included a short discussion for each entry in the table in Appendix F (former Appendix E). We believe it is sufficient to leave this detailed discussion to the appendix, because the derivation of the topological classification for all symmetry classes is quite technical as it relies on the results derived in the appendices and is only a side point of the paper. Also, the topological classification has been performed previously and more extensively in the literature, see for example [Shiozaki, arxiv:1907.09354]. The agreement of our result with the literature confirms our classification approach in terms of topological crystals.

It is correct that in the presence of time reversal symmetry, the superconducting order parameter can transform under the 2D irrep of $C_4$ or $C_6$ that is formed by gluing together the two complex conjugate 1D $E$ and $E^*$ irreps of $C_4$ or $C_6$. However, if the superconducting order parameter transforms under a 2D irrep, then the resulting BdG Hamiltonian must break the $C_4$ or $C_6$ symmetry. These 2D irreps consequently do not appear in our table, because we restrict ourselves to systems whose Hamiltonian is symmetric under a rotation symmetry. Note that only for 1D irreps, one can include a $U(1)$ gauge transformation in the representation of the crystalline symmetries on Nambu space that compensates the phase factor that the superconducting order parameter obtains under the crystalline symmetry operations (see for example Ref. [Geier et al., PRB 101, 245128 (2020)] for a detailed discussion).

Sincerely, Max Geier, Ion Cosma Fulga, and Alexander Lau

---

## Round 2 · Referee Report · Frank Schindler (Referee 2) · 2020-10-3

Strengths

1- Comprehensive 2- Detailed 3- Pedagogical 4- Great figures

Weaknesses

1- Often not rigorous 2- Presentation is somewhat disorganized

Report

In this manuscript, the authors derive a correspondence between bulk crystal topology and disclination bound states. They focus on second-order topological insulators with protected corner states. While the disclination response of such phases has already been partially discussed in the literature, a systematic treatment of all symmetry classes has so far been missing. The present manuscript fills this gap in a comprehensive manner by classifying the disclination response for all internal and point group symmetries. It also provides useful topological indicators that predict bound states.

I am therefore in favor of publication, but not before the following concerns are adequately addressed (see also additional requested changes below):

1- I do not understand the derivation of bound states for twofold symmetry. It is a general problem of the paper that the arguments made are often of a very qualitative character. This can be good -- it is pedagogical and physically intuitive -- but already in the case of twofold rotation symmetry, the dangers of oversimplification become evident: In the "Twofold rotation symmetry" part of section 3.4, it is stated that "In the resulting sample, the bulk and all boundaries are gapped by construction". Firstly, this is incorrect, as there is one remaining corner state at the boundary as required by anomaly cancellation [it is in fact shown in Fig 4(a)]. Secondly, what is the precise argument for why the corner state cannot remain on the boundary (while its partner does stay there)? The authors should argue this case more compellingly.

2- The derivation of disclination bound states based on topological crystals is neat. However, the manuscript glosses over the distinction between two very different length scales: (a) The extent $a$ of the physical unit cell. (b) The extent $A$ of the topological crystal unit cell, which has to satisfy $A \gg a$ in order for each of the topological crystal unit cells to accommodate subdimensional topological phases of its own. The derivation in section 4.4 implicitly assumes that the translational part of disclinations is of size $A$, but in realistic materials it will rather be of size $a$. It should be argued why ignoring this mismatch nevertheless produces physically meaningful results.

3- In section 5.3 it is stated that "In symmetry classes for which the bulk-boundary-defect correspondence holds, the direct sum of a first-order topological phase with itself cannot lead to a second-order topological phase". What about, for instance, class AII? Combining two copies of a time-reversal symmetric 3D topological insulator in a C2/C4/C6-symmetric fashion gives a second-order topological insulator, see e.g. section VIIB in the supplementary material of Science Advances Vol. 4, no. 6, eaat0346.

Requested changes

1- There's a consistent miss-spelling: "Franck" should really be the last name of Frederick Charles Frank 2- Typo in the caption of Fig. 2: "The unit cells of three- and sixfold symmetric lattice are parallelograms" -- isn't this only true for the threefold case? 3- inconsistency in and around Eq (2): the vector r sometimes has an index i and sometimes not 4- In section 2.5, the difference between "quantized magnetic flux" and "fluxoid quantization" should be explained, if any 5- In Fig. 4(a), why is it necessary to split the system in half in the first step? The top half seems to not play any role in the cutting-and-gluing procedure that follows (this may be related to me not understanding the argument for bound states here, see point 1 above) 6- In Fig. 7, some disclination bound states seem to be unpaired (see e.g. 2nd row, 2nd column). This is impossible by the anomaly cancellation requirement. Indeed, upon closer inspection, in these cases the boundary always hosts an odd number of gapless states and so necessarily remains gapless even when translational symmetry is relaxed. This consistency should be pointed out explicitly, right now it is confusing that only one state is circled in green without further comment. 7- At the bottom of page 17, the $\nu$ invariants need to be defined before they are used (or their definition referenced in case I missed it) 8- The notion of a "geometric $\pi$-flux quantum" needs to be defined precisely. What's the difference to a magnetic $\pi$-flux? 9- In Eq. (11), there's a "mod 2" missing on the right-hand side 10- typo in Table 2, row "AII": the column belonging to $\phi$ should be empty 11- Caption of Fig. 10: "lowest eigenstates with $E \geq 0$" -- these should be at exactly $E = 0$ due to the particle hole symmetry 12- Typo in the conclusion "as construced"

  • validity: high
  • significance: high
  • originality: high
  • clarity: good
  • formatting: excellent
  • grammar: perfect

Author:  Max Geier  on 2021-02-12  [id 1233]

(in reply to Report 2 by Frank Schindler on 2020-10-03)

We thank the referee for the positive report and for the constructive criticism. Below we address all comments point by point. We have provided clarifications in the manuscript in response to the comments. We believe that thanks to the referees suggestions, we were able to further improve the quality of our manuscript.

** Second referee: **

In this manuscript, the authors derive a correspondence between bulk crystal topology and disclination bound states. They focus on second-order topological insulators with protected corner states. While the disclination response of such phases has already been partially discussed in the literature, a systematic treatment of all symmetry classes has so far been missing. The present manuscript fills this gap in a comprehensive manner by classifying the disclination response for all internal and point group symmetries. It also provides useful topological indicators that predict bound states.

I am therefore in favor of publication, but not before the following concerns are adequately addressed (see also additional requested changes below):

** Our response: ** We thank the referee for the positive review of our manuscript, for the constructive comments and for the recommendation for publication.

** Second referee: **

1- I do not understand the derivation of bound states for twofold symmetry. It is a general problem of the paper that the arguments made are often of a very qualitative character. This can be good -- it is pedagogical and physically intuitive -- but already in the case of twofold rotation symmetry, the dangers of oversimplification become evident: In the "Twofold rotation symmetry" part of section 3.4, it is stated that "In the resulting sample, the bulk and all boundaries are gapped by construction". Firstly, this is incorrect, as there is one remaining corner state at the boundary as required by anomaly cancellation [it is in fact shown in Fig 4(a)]. Secondly, what is the precise argument for why the corner state cannot remain on the boundary (while its partner does stay there)? The authors should argue this case more compellingly.

** Our response: ** We thank the referee for this comment and apologize for the confusion. In our argument, we meant to distinguish between $(d-1)$-dimensional boundaries and $(d-2)$-dimensional corners or hinges of the crystal. More clearly, the mentioned sentence in the "Twofold rotation symmetry" part of section 3.4 should read "In the resulting sample, the bulk and all $(d-1)$-dimensional boundaries are gapped by construction" in order to distinguish the manifolds of different dimension on the boundary. We have clarified this in the revised manuscript. This more restrictive statement, as it was meant to be, then distinguishes between $(d-1)$-dimensional gapless modes localized to $(d-1)$-dimensional boundaries and $(d-2)$-dimensional gapless or ingap modes localized to $(d-2)$-dimensional corners or hinges. As the system in consideration exhibits only a corner/hinge state at a $(d-2)$-dimensional corner / hinge, this clarified statement is correct.

The argument for why the corner state at the folded corner cannot remain on the boundary is the following. Requiring that the cut line along which we connect the folded boundaries becomes indistinguishable from the bulk of the system, also the intersection of the cut line with the $(d-1)$-dimensional boundary in the folded system [c.f. the rightmost depiction in Fig.~4(a)] is a point which is indistinguishable from any other point on the $(d-1)$-dimensional boundary. Therefore, it cannot host a $(d-2)$-dimensional anomalous state, because if it would, this point would be distinguishable. Also, upon connecting the boundaries on the cut line, the $(d-2)$-dimensional anomalous state that is localized to one of the corners and will be connected upon folding cannot propagate to the other corner not involved in the folding process. This is because the $(d-1)$-dimensional boundaries are gapped. Therefore, the only possibility is that, upon connecting the boundaries in the folding process, this $(d-2)$-dimensional anomalous state moves to the disclination, which is itself a $(d-2)$-dimensional defect in the crystal.

We have implemented some minor changes in the respective paragraph to clarify those issues.

** Second referee: **

2- The derivation of disclination bound states based on topological crystals is neat. However, the manuscript glosses over the distinction between two very different length scales: (a) The extent $a$ of the physical unit cell. (b) The extent $A$ of the topological crystal unit cell, which has to satisfy $A \gg a$ in order for each of the topological crystal unit cells to accommodate subdimensional topological phases of its own. The derivation in section 4.4 implicitly assumes that the translational part of disclinations is of size $A$, but in realistic materials it will rather be of size $a$. It should be argued why ignoring this mismatch nevertheless produces physically meaningful results.

** Our response: ** As argued in the original work on topological crystals (Ref. [Song et al., Science Advances 5(12), eaax2007 (2019)]), the topological crystal construction describes a real-space limit for topological crystalline phases that lies within the stable homotopy equivalence class of the topological crystalline phase. More precisely, the authors of Ref. [Song et al., Science Advances 5(12), eaax2007 (2019)] require that there exists an adiabatic process that deforms a given system within the stable homotopy equivalence class into the topological crystal limit while adding a sufficiently fine mesh of trivial degrees of freedom. This requires that any characteristic correlation length or entanglement length at the end of the process $\xi_f$ is much smaller than the unit cell size. If one considers a disclination in a finite system, one therefore needs to require that any length scale of the sample size should be much larger than any characteristic correlation length or entanglement length $\xi$ such that the bulk of the system can be adiabatically deformed into its topological crystal limit. When we perform the Volterra process for the topological crystal limits, we consider finite, but arbitrarily large systems. Therefore, our results obtained from the topological crystal limit should apply for the system in question within the stable homotopy equivalence class as long as any length scale of the sample size is much larger than any characteristic correlation length or entanglement length $\xi$ of the system. Notice that a similar requirement is made when showing the existence of defects states using K-theory as in Ref. [Teo, Kane PRB 115120 (2010)]: There, one requires translational invariance locally for each point in space, resulting in an effective Hamiltonian defined over reciprocal and real space coordinates. The requirement of local translational invariance is only reasonable if the real-space variation of the Hamiltonian is slow on any of its correlation or entanglement length scales.

The authors of Ref. [Song et al., Science Advances 5(12), eaax2007 (2019)] further argue that the existence of such an adiabatic process as discussed above is not proven in general. Therefore, their construction applies only to crystalline topological phases for which such an adiabatic process exists. The authors of Ref. [Song et al., Science Advances 5(12), eaax2007 (2019)] conjecture that such an adiabatic process should exist for all crystalline topological phases. For our case of rotation-symmetric systems, since the classification results from K-theory and from topological crystals agree, we expect that every rotation-symmetric topological phase contains the topological crystal limit within its stable homotopy equivalence class.

Along the lines of this discussion, we have added an explanatory note at the beginning of Sec. 4 and a new subsection Appendix C.4.

** Second referee: **

3- In section 5.3 it is stated that "In symmetry classes for which the bulk-boundary-defect correspondence holds, the direct sum of a first-order topological phase with itself cannot lead to a second-order topological phase". What about, for instance, class AII? Combining two copies of a time-reversal symmetric 3D topological insulator in a C2/C4/C6-symmetric fashion gives a second-order topological insulator, see e.g. section VIIB in the supplementary material of Science Advances Vol. 4, no. 6, eaat0346.

** Our response: ** In the mentioned example for $d=3$ dimensions in class AII, the Hamiltonian $H_h$ for the higher order phase in Eq. (S41) of Ref. [Science Advances Vol. 4, no. 6, eaat0346] in the limit $\Delta_2 = 0$ indeed corresponds to a direct sum of two Hamiltonians of first order topological phases $H_1$, $H_h(\Delta_2 = 0) = H_1 \oplus_\rho H_1$. However, the representation of fourfold rotation symmetry as given in Eq. (S43) exchanges the two Hamiltonians in the direct sum (as can be seen as $C_4$ is proportional to the Pauli matrix $\rho_1$). Therefore, the full system defined by the Hamiltonian together with the representations of its symmetries is not equivalent to the direct sum of a first order topological phase with itself (where both Hamiltonian and representations would be proportional to the $2 \times 2$ identity matrix $\rho_0$).

The statement in our work relates to the abelian group structure of the classifying groups, where addition is defined in terms of the direct sum of the full system including Hamiltonian and representations. We have added a footnote at the end of Sec. 5 to emphasize that the representations are included in the direct sum.

** Second referee: **

1- There's a consistent miss-spelling: "Franck" should really be the last name of Frederick Charles Frank

** Our response: ** We thank the referee for pointing this out. We have corrected this throughout the manuscript.

** Second referee: **

2- Typo in the caption of Fig. 2: "The unit cells of three- and sixfold symmetric lattice are parallelograms" -- isn't this only true for the threefold case?

** Our response: ** Correct -- we have corrected the caption.

** Second referee: **

3- inconsistency in and around Eq (2): the vector r sometimes has an index i and sometimes not

** Our response: ** Around Eq. (2), we use $\vec{r}$ to denote the positions close to the disclination and $\vec{r}_i$ to denote a position in the translation symmetric bulk. For clarification, we have included a footnote below Eq. (2):

"For concreteness, the element $H_{\vec{r}_i, \vec{r}_i + \vec{a}_n}$ can be taken from the translation symmetric system without disclination and without boundary whose lattice positions are denoted by $\vec{r}_i$."

** Second referee: **

4- In section 2.5, the difference between "quantized magnetic flux" and "fluxoid quantization" should be explained, if any

** Our response: ** We have clarified this in the revised version. We now refer directly to the geometric phase acquired by a particle as it is parallel transported along a closed path. This phase is quantized to multiples of $\pi$ both by time-reversal as well as particle-hole symmetry.

** Second referee: **

5- In Fig. 4(a), why is it necessary to split the system in half in the first step? The top half seems to not play any role in the cutting-and-gluing procedure that follows (this may be related to me not understanding the argument for bound states here, see point 1 above)

** Our response: ** Cutting the sample in the first step has a conceptual advantage: one creates a sample with boundaries that are mapped onto each other by the rotation symmetry. This in turn allows to fold the sample and connect the boundaries by including the representation of twofold rotation, with the prescription of Eq. (2). In particular in the absence of translation symmetry, if the system has a continuous large scale order parameter consistent only with the rotation symmetry in the original sample before cutting -- such as a symmetric variation of the charge density -- the cutting in the first step ensures that the this large scale order parameter is continuous after gluing the system to form the disclination in the second step.

However, in our argumentation in the main text, we assumed that the system realizes a second-order topological phase for any sufficiently large cut-out region. This implies a sense of translation symmetry in the bulk of the topological region and allows to symmetrically deform the separated regions. As a result, it is indeed not necessary to perform the cutting procedure in the first step.

** Second referee: **

6- In Fig. 7, some disclination bound states seem to be unpaired (see e.g. 2nd row, 2nd column). This is impossible by the anomaly cancellation requirement. Indeed, upon closer inspection, in these cases the boundary always hosts an odd number of gapless states and so necessarily remains gapless even when translational symmetry is relaxed. This consistency should be pointed out explicitly, right now it is confusing that only one state is circled in green without further comment.

** Our response: ** We thank the referee for pointing this out and have included a note in the caption of the figure.

** Second referee: **

7- At the bottom of page 17, the $\nu$ invariants need to be defined before they are used (or their definition referenced in case I missed it)

** Our response: ** We have included a footnote on how the invariants are abstractly defined. In this footnote, we also point to the examples where we include explicit expressions for the topological invariants.

** Second referee: **

8- The notion of a "geometric $\pi$-flux quantum" needs to be defined precisely. What's the difference to a magnetic $\pi$-flux?

** Our response: ** We have introduced the notion of a geometric $\pi$-flux to refer to a situation in which the geometric phase acquired by a particle when it is parallel transported along a closed loop is quantized to $\pi$. In a time-reversal symmetric, normal state system, this is identical to the magnetic $\pi$-flux. In a particle-hole symmetric system, for instance in a superconductor, this geometric phase counts the net number of superconducting vortices encircled by the particle. We wanted to avoid referring to it only as a magnetic flux, since in superconductors this $\pi$ shift occurs in the full, gauge invariant phase difference, which contains contributions both from the vector potential, as well as from the pairing potential. We have better explained this in the Section 2.6 of our revised version.

** Second referee: **

9- In Eq. (11), there's a "mod 2" missing on the right-hand side

** Our response: ** Thanks! Corrected.

** Second referee: **

10- typo in Table 2, row "AII": the column belonging to $\phi$ should be empty

** Our response: ** Indeed. We have corrected that.

** Second referee: **

11- Caption of Fig. 10: "lowest eigenstates with $E \geq 0$" -- these should be at exactly $E=0$ due to the particle hole symmetry

** Our response: ** Due to the finite size of our sample, all eigenstates acquire a finite energy -- including the Majorana bound states whose energy is exponentially small in the system size due to their finite overlaps. For clarification, we have changed the caption to "... two eigenstates with lowest absolute energy" and checked that the displayed wavefunction weights match this label.

** Second referee: **

12- Typo in the conclusion "as construced"

** Our response: ** Thanks! Typo corrected.

Sincerely, Max Geier, Ion Cosma Fulga, and Alexander Lau

---

## Round 2 · Referee Report · Anonymous (Referee 3) · 2020-10-28

Strengths

1- novel and of immediate relevance 2- exhaustive classification results 3- clear illustrations 4- simple explicit models

Weaknesses

1- structure 2- should better discriminate original results from prior ones 3- some argument not clear

Report

In this work, the authors establish a correspondence between bulk band topology and anomalous states bound to disclination defects. The disclinations are constructed via the Volterra processes, and both the role of the Franck angle and of the translational holonomy are investigated. The appearance of disclination-bound states is considered both for strong topological insulators possibly breaking translational symmetry as well as for crystalline topological insulators characterized using the technique of topological crystals. The theoretical arguments of this work are supported by clear and very helpful illustrations. The conclusions obtained by this work provide an important extension of prior works on the closely related topic of bound states at dislocations, and their appearance is very timely. The results of the analysis appear very exhaustive, original, and of immediate scientific value.
In summary, I am strongly inclined to recommend this work for publication.

On the other hand, I have to criticize the overall organization of the manuscript, which has resulted in the manuscript being a slow and cumbersome read. Studying this work costs an unexpectedly long time – not because of the length or difficulty, but rather because of the often sloppy structure. The authors have appropriately moved some technical parts of the argumentation into appendices, but I found their overall structure rather disorganized – e.g. when certain section of the main text defers detailed discussion to an appendix, whereas the corresponding appendix retrieves results derived in later sections of the text. At other occasions, a paragraph of discussion is included in the text, which does not bear a clear relation to the prior/previous text and lacks a clear motivation (few examples listed further below).

Concerning the structure, I think the manuscript would be readily improved if the authors could do the following two straightforward amendments:

1.) Begin each section with a short summary of its role: How does it connect to the previous sections? How is it structured into subsections? How are the results obtained here needed in the following sections, and how are they relevant for the overarching goal of the paper? How is this section supplemented by Appendices? Similar (perhaps shorter) summary could be placed at the beginning of most subsections (especially the longer ones). Although such summaries would admittedly further increase the length of the manuscript, it would at the same time greatly help the readers navigate through the arguments and discussions.

2.) Place a summary of the main results somewhere: either just after introduction, or in the beginning of the conclusions. It seems to me that the main results are scattered on pages 18—20, especially in Eqs. (7—11) (Remark: Note that the notation “$\mu$” for topological invariants is not even introduced until one gets to read these equations), in the following characterization of systems with unitary rotation symmetry whose representation commutes/does not commute with all internal unitary symmetries/antisymmetries + in Tables 1 and 2. In the present version of the manuscript, it is difficult to locate the results of the analysis are without immersing deep into reading.

Note that I am not asking for new results or for restructuring of the manuscript, so this should not be difficult to achieve.

Furthermore, I think the authors should do a better job distinguishing in their presentation the original results from a review of prior works. At a few occasions, I found the presented arguments not very rigorous (some specific cases are listed below), although I understand that the authors had to find a balance between rigor and clarity. Finally, while the explicit model examples discussed in the last Section are clear and helpful, I was a bit disappointed not to find any motivating example even mentioned until finally getting to apply the derived machinery in the last Section.

I think the authors should keep in mind the above-listed points in mind when preparing their manuscript for resubmission.

Besides these general remarks and comments on the structure, I also have the following more specific comments that the authors should consider:

1.) The work provides a great review of the classification of disclination defects, part of which I have not known before. In contrast, the notion $Hol(\Omega)$ is introduced without a clear mathematical formulation. Do I understand correctly that one (1) constructs an “order-parameter space” M=E/G (E = Euclidean group, G = space group), (2) uses fundamental group $\pi_1(M)$ to describe codimension-2 defects, and finally (3) partitions the fundamental group into equivalence classes $Hol(\Omega)$? A brief construction of the holonomy classes would be helpful (even if only as a footnote).

2.) Related to the previous comment: Do I understand correctly that the listed holonomies in Eq. (1) apply only to Bravais lattices? This is not clearly specified, and it seems to me that more complicated space groups decorated with multiple sites and/or with nonsymmorphic symmetries are not captured by this equation. (Or are these more complicated cases somehow covered by the discussion in Appendices A.1 and A.2?) In either case, the authors should make sure the assumptions for this equation are stated clearly.

3.) In Fig. 1(d—e), shouldn’t the blue dashed Volterra cut go through mid-bonds and mid-cells, rather than through the vertices?

4.) Section 2.5 about spinful fermions begins with an argument that I did not understand, even after several attempts. It says that “when transporting a half-integer spinful particle around a $2\pi$ disclination, there are two effects contributing a $\pi$ phase to its wavefunction: (i) the rotation of the real space coordinate system, and (ii) the basis rotation of the local degrees of freedom”. But (i) and (ii) just look like the passive vs. the active description of the same thing (rotating the coordinates in which to express the degrees of freedom, vs. rotating the degrees of freedom with respect to the coordinates). What do I miss? Could the authors make this argument more explicit?

5.) When building up the topological crystal for space group p2 in Fig. 5(a) and in Appendix B.1, the quarter unit cell (i.e. the 2-cell) is decorated by three 1-cells. The fourth one, which would be (x,a/2) with x in range [0,a/2], is missing. Therefore, it is not true, as written in the Appendix, that the three 1-cells “together cover the complete boundary of the 2-cell upon translating [+rotating?] them with the crystalline symmetry”. Am I right? Then why is this fourth 1-cell dropped? Is it because one thinks about the cells in homological terms, i.e. the boundary of such a cell would simply be the boundary of the sum of the listed three 1-cells, thus making it redundant?

6.) It is not completely clear how the results in Eqs. (7—10) should be read. For example for the pi/2-disclination in Eq. (9), the holonomy has been claimed to be $Z_2$-valued. Then what are the possible values of the two-component vector T in (9)? Do T-vectors (1,0) and (0,1) (which should be equivalent to each other after local rearrangement of atoms) lead to the same number of bound states? It might be worth elaborating a bit on this result, e.g. how the bound state is moved if a disclination with non-zero translational holonomy is split into a pure disclination and a pure dislocation.

7.) When discussing the classification result on page 20, the authors write “(ii) In the remaining subset of symmetry classes where strong, rotation-symmetry protected second-order topological phases are forbidden, the anomaly at the disclination may still be determined from the bulk topology alone.” However, assuming I read Tables 1 and 2 correctly, rather then this “might” being the case, it actually “always is” the case. Or do I misunderstand how to read the tables?

8.) In Fig. 12 for the sixfold-symmetric class D second-order TSC, it might be illuminating to also explicitly show the results for a 2pi/3 (i.e. “doubled”) disclination, when the theory predicts the absence of a disclination-bound state.

9.) Why do the authors speak of “bulk-boundary-defect correspondence” instead of just “bulk-defect correspondence”?

10.) The authors add a short paragraph on disclination dipole in App. A.2, but I didn’t understand the purpose. It seems to relate to nothing else in the rest of the manuscript. Similarly, there is a paragraph on first-order topology near the end of App. B.2 which appears to be misplaced, and probably should be a part of App. C. I also didn’t understand how the paragraph on the relation to Ref. [46] on page 17 fits in the text. (There were a few more such paragraphs, but these three I found particularly confusing.)

11.) Could the authors more clearly explain why Fig. B.4(d) corresponds to a block-off-diagonal coupling with the x-Pauli matrix (cf. text on p.48)?

12.) In the discussion of fluxes bound to defects under Eq. (11), the charge “2e” is set in the magnetic flux quantum. Is this because this discussion only applies to superconductors? Or does this discussion also apply to Gedankenexperiments with topological insulators?

Some typos:
• In Table 2, class AII: The column for phase phi should be empty(?)
• Caption to Fig. A.3 should refer to Eq. (19) [instead of (20)]
• Extra “the” on p. 47 in “upon translating the them with”
• On page (50), there should be “The topological crystal shown in Fig. B.4 (c)” [rather than (d)].
• Two paragraphs further, in sentence “that (a) is a weak topological phase and (b) is a strong” replace by (d) and (e).

Assuming all the listed issues are properly resolved, I would most likely recommend the work for publications after the authors’ resubmission.

Requested changes

(see report)

  • validity: high
  • significance: top
  • originality: high
  • clarity: good
  • formatting: good
  • grammar: excellent

Author:  Max Geier  on 2021-02-12  [id 1234]

(in reply to Report 3 on 2020-10-28)
Category:
remark
answer to question
validation or rederivation
pointer to related literature

We thank the referee for the positive report and for the constructive criticism. Below we address all comments point by point. We have provided clarifications in the manuscript in response to the comments. We have also extended the overview of our article in our introduction, the summary of our main results in the conclusion, and added section introduction and summaries to further improve the readability of our manuscript. We believe that thanks to the referees suggestions, we were able to further improve the quality of our manuscript.

** Third referee: **

In this work, the authors establish a correspondence between bulk band topology and anomalous states bound to disclination defects. The disclinations are constructed via the Volterra processes, and both the role of the Franck angle and of the translational holonomy are investigated. The appearance of disclination-bound states is considered both for strong topological insulators possibly breaking translational symmetry as well as for crystalline topological insulators characterized using the technique of topological crystals. The theoretical arguments of this work are supported by clear and very helpful illustrations. The conclusions obtained by this work provide an important extension of prior works on the closely related topic of bound states at dislocations, and their appearance is very timely. The results of the analysis appear very exhaustive, original, and of immediate scientific value. In summary, I am strongly inclined to recommend this work for publication.

** Our response: ** We thank the referee for highlighting the main results of our manuscript and for their positive words. We also appreciate their inclination towards recommending our manuscript for publication.

** Third referee: **

On the other hand, I have to criticize the overall organization of the manuscript, which has resulted in the manuscript being a slow and cumbersome read. Studying this work costs an unexpectedly long time – not because of the length or difficulty, but rather because of the often sloppy structure. The authors have appropriately moved some technical parts of the argumentation into appendices, but I found their overall structure rather disorganized – e.g. when certain section of the main text defers detailed discussion to an appendix, whereas the corresponding appendix retrieves results derived in later sections of the text. At other occasions, a paragraph of discussion is included in the text, which does not bear a clear relation to the prior/previous text and lacks a clear motivation (few examples listed further below).

** Our response: ** We understand the general criticism of the referee regarding the readability of our manuscript. We have therefore put additional effort into improving the organization and readability of the different sections according to the referee's suggestions, as detailed below.

** Third referee: **

Concerning the structure, I think the manuscript would be readily improved if the authors could do the following two straightforward amendments:

1.) Begin each section with a short summary of its role: How does it connect to the previous sections? How is it structured into subsections? How are the results obtained here needed in the following sections, and how are they relevant for the overarching goal of the paper? How is this section supplemented by Appendices? Similar (perhaps shorter) summary could be placed at the beginning of most subsections (especially the longer ones). Although such summaries would admittedly further increase the length of the manuscript, it would at the same time greatly help the readers navigate through the arguments and discussions.

** Our response: ** We thank the referee for the useful suggestions. For the revised version of the manuscript, we have improved the section and subsection introductions throughout the text to clarify their role and contribution to the overall narrative. We have added short summaries at the end of the sections to highlight the main results of each section and their importance for the course of the paper. We have also restructured some of the sections, improved old subsection titles and introduced new subsection titles. In this way, the overall structure of our paper becomes also much clearer just by looking at the table of contents. Finally, we have also tried to better integrate the references to the Appendices.

** Third referee: **

2.) Place a summary of the main results somewhere: either just after introduction, or in the beginning of the conclusions. It seems to me that the main results are scattered on pages 18—20, especially in Eqs. (7—11) (Remark: Note that the notation “$\nu$” for topological invariants is not even introduced until one gets to read these equations), in the following characterization of systems with unitary rotation symmetry whose representation commutes/does not commute with all internal unitary symmetries/antisymmetries + in Tables 1 and 2. In the present version of the manuscript, it is difficult to locate the results of the analysis are without immersing deep into reading.

Note that I am not asking for new results or for restructuring of the manuscript, so this should not be difficult to achieve.

** Our response: ** We thank the referee for their important criticism and suggestions. We have included a summary of our main results in the beginning of the conclusions.

** Third referee: **

Furthermore, I think the authors should do a better job distinguishing in their presentation the original results from a review of prior works. At a few occasions, I found the presented arguments not very rigorous (some specific cases are listed below), although I understand that the authors had to find a balance between rigor and clarity.

** Our response: ** We thank the referee for pointing us to this important point. We have extended the introduction in order to point to the main results of our work more clearly. In particular, we have highlighted that our work is a generalization of the many previous results for individual cases in literature. Furthermore, we have included a separate results section in the Conclusions and added section summaries throughout the manuscript. We hope that these amendments make our results stand out more clearly.

** Third referee: **

Finally, while the explicit model examples discussed in the last Section are clear and helpful, I was a bit disappointed not to find any motivating example even mentioned until finally getting to apply the derived machinery in the last Section.

** Our response: ** We thank the Referee for this suggestion. Giving a motivating example is clearly one possible way to introduce the reader to a topic. However, we think that this is also a matter of personal taste and we, instead, found a linear structure to be more suitable for the presentation of our results. We would therefore prefer to keep the example section as it is and hope for the Referee's understanding.

As a small compromise from our side, we now explicitly point the reader to the example section in the summary of the Sec. 3. In this way, the reader might already have a glimpse of some exemplary models before proceeding with the more technical sections 4 and 5.

** Third referee: **

I think the authors should keep in mind the above-listed points in mind when preparing their manuscript for resubmission.

** Our response: ** We have carefully considered the Referee's comments and suggestions and substantially improved the organization of the text.

** Third referee: **

Besides these general remarks and comments on the structure, I also have the following more specific comments that the authors should consider:

1.) The work provides a great review of the classification of disclination defects, part of which I have not known before. In contrast, the notion $\text{Hol}(\Omega)$ is introduced without a clear mathematical formulation. Do I understand correctly that one (1) constructs an ''order-parameter space'' $M=E/G$ ($E$ = Euclidean group, $G$ = space group), (2) uses fundamental group $\pi_1(M)$ to describe codimension-2 defects, and finally (3) partitions the fundamental group into equivalence classes $\text{Hol}(\Omega)$? A brief construction of the holonomy classes would be helpful (even if only as a footnote).

** Our response: ** Indeed, this is a way how the equivalence classes $\text{Hol}(\Omega)$ can be derived. We have included a summary of the derivation as a new appendix (new Appendix A).

** Third referee: **

2.) Related to the previous comment: Do I understand correctly that the listed holonomies in Eq. (1) apply only to Bravais lattices? This is not clearly specified, and it seems to me that more complicated space groups decorated with multiple sites and/or with nonsymmorphic symmetries are not captured by this equation. (Or are these more complicated cases somehow covered by the discussion in Appendices A.1 and A.2?) In either case, the authors should make sure the assumptions for this equation are stated clearly.

** Our response: ** In response to this comment, we have updated Sec.~2 of our manuscript to include a more precise definition of the lattice in terms of the charge density. Also in response to the first comment 1.), we hope that the discussion in the appendix clarifies that (i) the classification of holonomies applies for any rotation symmetric charge density in the primitive unit cell, which captures also multiple sites (and therefore also non-Bravais lattices) and which may also satisfy additional symmetries, and (ii) the holonomy classification scheme does not include however additional symmetries. By including additional symmetries as allowed operations in the holonomy classification, we expect that one may find a larger, but not a smaller, set of distinct holonomies, because one can always restrict to using only symmetry operators of the smaller, original space group to recover its corresponding holonomies.

** Third referee: **

3.) In Fig. 1(d—e), shouldn’t the blue dashed Volterra cut go through mid-bonds and mid-cells, rather than through the vertices?

** Our response: ** In the caption of Fig. 1 we now specify that the solid lines denote the boundary of the primitive unit cell. We furthermore clarify in the text that, in our construction, we require that the cuts are performed along boundaries of unit cells in order to ensure that the lattice of the glued system is locally indistinguishable from a defect-free lattice along the cut line.

** Third referee: **

4.) Section 2.5 about spinful fermions begins with an argument that I did not understand, even after several attempts. It says that “when transporting a half-integer spinful particle around a $2\pi$ disclination, there are two effects contributing a $\pi$ phase to its wavefunction: (i) the rotation of the real space coordinate system, and (ii) the basis rotation of the local degrees of freedom”. But (i) and (ii) just look like the passive vs. the active description of the same thing (rotating the coordinates in which to express the degrees of freedom, vs. rotating the degrees of freedom with respect to the coordinates). What do I miss? Could the authors make this argument more explicit?

** Our response: ** The first contribution (i) is the $\pi$ phase acquired by a spinful fermion upon $2\pi$ rotation in real space. The second contribution (ii) denotes the $\pi$ phase factor obtained by the application of the gluing construction in Eq. (2), which includes an action of the representation $U(\mathcal{R}_{2\pi}) = -1$.

One may ask if it is necessary to include the second contribution, i.e. whether it is necessary to insist on the gluing construction in Eq. (2) also for $2\pi$ disclinations. We argue that one should do so, because only then is the distinction between disclinations and geometric $\pi$ fluxes clear and only then one can split a $2 \pi$ disclination into separate disclinations with smaller Frank angle without introducing spurious geometric $\pi$ fluxes.

We have clarified the formulation in the main text and included an explanatory footnote.

** Third referee: **

5.) When building up the topological crystal for space group p2 in Fig. 5(a) and in Appendix B.1, the quarter unit cell (i.e. the 2-cell) is decorated by three 1-cells. The fourth one, which would be (x,a/2) with x in range [0,a/2], is missing. Therefore, it is not true, as written in the Appendix, that the three 1-cells “together cover the complete boundary of the 2-cell upon translating [+rotating?] them with the crystalline symmetry”. Am I right? Then why is this fourth 1-cell dropped? Is it because one thinks about the cells in homological terms, i.e. the boundary of such a cell would simply be the boundary of the sum of the listed three 1-cells, thus making it redundant?

** Our response: ** For space group $p2$, we choose the asymmetric unit to extend over two adjacent unit cells, with boundary as given in Appendix C.1 (previously Appendix B.1). With this choice, the line (x,a/2) with x in range [0,a/2] is not a boundary of the asymmetric unit -- even though it lies at the boundary of the unit cell. Therefore, there is no 1-cell at this line, because it does not appear as a boundary of our asymmetric unit. We used this freedom of choosing an asymmetric unit extending beyond the unit cell such that the corner of the asymmetric unit is at the unit cell center.

It is an interesting observation that the additional 1-cell at the line (x,a/2) with x in range [0,a/2] would be redundant. This redundancy can also be shown with an explicit deformation: A decoration with this "additional" 1-cell is equivalent to a decoration with the three "original" 1-cells (the ones shown in Fig. 5(a)). This can be seen as one can symmetrically deform the additional 1-cell by pulling at its center until the deformed additional 1-cell coincides with the three original 1-cells. This process can also be reversed. During this deformation, we kept the boundary of the 1-cell(s) fixed. It is very interesting that the existence of such a deformation (and the resulting redundancy) already seems to follow form the fact that the boundaries of the related 1-cells are identical, as the referee suggested (even though we are currently not aware of a general proof).

We apologize that our formulation in the text was not clear and have extended the discussion to point to the subtlety.

** Third referee: **

6.) It is not completely clear how the results in Eqs. (7—10) should be read. For example for the pi/2-disclination in Eq. (9), the holonomy has been claimed to be $\mathbb{Z}_2$-valued. Then what are the possible values of the two-component vector T in (9)? Do T-vectors (1,0) and (0,1) (which should be equivalent to each other after a local rearrangement of atoms) lead to the same number of bound states? It might be worth elaborating a bit on this result, e.g. how the bound state is moved if a disclination with non-zero translational holonomy is split into a pure disclination and a pure dislocation.

** Our response: ** In fact, Eqs.~(7) - (10) apply to any value of the two-component vector T. To elaborate on this, we have included the following paragraph in the main text:

"For symmetry classes whose $(d-2)$-dimensional anomalous states have $\mathbb{Z}_2$ topological charge, these equations not only predict the total parity of anomalous states at disclinations, but also at dislocations and at collections thereof. First, Equations (7) to (10) are also valid for dislocations, because for zero Frank angle, the equations agree with the familiar result for dislocations [Teo and Hughes, Annual Review of Condensed Matter Physics, 8(1), 211, (2017)]. Second, Equations (7) to (10) depend only on the holonomical quantities of a loop around the defect. This allows to perform local rearrangements of the lattice at the defect, which in particular allows to split the defect into multiple defects. By regarding the holonomy of a loop around each of these defects, one can apply Eqs. (7) to (10) to each defect individually. This shows that Eqs. (7) to (10) determine the parity of anomalous states of collections of defects from the holonomy of an enclosing loop. As a consequence, Eqs. (7) to (10) also determine the fate of the defect anomalies upon splitting and, conversely, fusion of lattice defects."

Regarding the fourfold rotation symmetry, we have added the following sentence:

"Furthermore, recall that in Eq. (9), fourfold rotation symmetry requires that $\nu_x = \nu_y$ such that for a $\pi/2$ disclination of type $1$, the two equivalent translation holonomies $\vec{T}=(0,1)$ and $\vec{T}=(1,0)$ always yield the same result."

We hope that these additional explanations provide a sufficient answer to the Referee's questions.

** Third referee: **

7.) When discussing the classification result on page 20, the authors write “(ii) In the remaining subset of symmetry classes where strong, rotation-symmetry protected second-order topological phases are forbidden, the anomaly at the disclination may still be determined from the bulk topology alone.” However, assuming I read Tables 1 and 2 correctly, rather then this “might” being the case, it actually “always is” the case. Or do I misunderstand how to read the tables?

** Our response: ** Indeed for all cases mentioned in the table, this always is the case. When deriving the tables, we made specific assumptions on how the symmetry classes are realized in a physical system (i.e. spinful/spinless fermions, spin rotation symmetry, et cetera). However, there are other, mathematically allowed combinations of symmetry class and dimension where no second order phases exist (i.e. they fall in category (ii)), yet the combinations of symmetries are such that a domain wall must exist.

For example, in two dimensions, class BDI with a twofold rotation symmetry that squares to 1 which anticommutes with time-reversal symmetry $\mathcal{T}$ and commutes with particle-hole antisymmetry $\mathcal{P}$. Therefore the rotation symmetry also anticommutes with the unitary antisymmetry $\mathcal{TP}$. This combination of symmetries requires that a $\pi$ disclination must be bound to a domain wall. Also, the topological classification in this symmetry class for two dimensions reveals that there is no second order phase but there is a weak topological phase (as tabulated in Table IV of Ref. [PRX, 9, 011012] under the entry $\text{BDI}^{\mathcal{S}_{-+}}$ for $d=2$ and $\mathcal{S} = \mathcal{R}$ for the second-order phase and $d=2$ and $\mathcal{S} = \mathcal{M}$ for the weak phase). This is an example where disclinations including a translation holonomy may host anomalous states, however the disclination is also bound to a domain wall for which its anomaly can not be determined from the bulk topology alone.

As a side remark to this example, note that the weak phase is constrained to even values of the classifying group of class BDI, i.e. the topological crystal of the weak phase requires a decoration with pairs of one-dimensional class BDI generators. This constraint is required by the similar topological crystal constructions of the weak and second-order topological phases.

** Third referee: **

8.) In Fig. 12 for the sixfold-symmetric class D second-order TSC, it might be illuminating to also explicitly show the results for a 2pi/3 (i.e. “doubled”) disclination, when the theory predicts the absence of a disclination-bound state.

** Our response: ** We thank the Referee for this suggestion. We have added a new figure (Fig. 13) to show that a 2pi/3 disclination does not lead to anomalous disclination states.

** Third referee: **

Why do the authors speak of “bulk-boundary-defect correspondence” instead of just “bulk-defect correspondence”?

** Our response: ** We use the term “bulk-boundary-defect correspondence to highlight the connection between the bulk topology and the corresponding appearance of anomalous states on the boundary -- the defining feature of higher-order topological phases -- and at disclinations as lattice defects. The term "bulk-defect correspondence" highlights only a connection between bulk topology and defect anomaly, leaving open the question of whether anomalies might exist on the boundary. We believe it is an important result that the appearance of higher-order boundary anomalies and defect anomalies are tied to the same bulk topological phase. Therefore, we would like to highlight this fact in the title of our work.

** Third referee: **

10.) The authors add a short paragraph on disclination dipole in App. A.2, but I didn’t understand the purpose. It seems to relate to nothing else in the rest of the manuscript. Similarly, there is a paragraph on first-order topology near the end of App. B.2 which appears to be misplaced, and probably should be a part of App. C. I also didn’t understand how the paragraph on the relation to Ref. [46] on page 17 fits in the text. (There were a few more such paragraphs, but these three I found particularly confusing.)

** Our response: ** In our construction of single disclinations, the domain wall along the cut line must connect to the boundary. However, in the paragraph in Appendix~B.2 (previously Appendix A.2), we show that for a disclination dipole the domain wall rather connects the two disclinations. We have added a motivating phrase at the beginning of the paragraph, and referenced it in the main text.

The paragraph on first-order topology near the end of Appendix C.2 (previously Appendix B.2) contains a small result on the consistency of a first-order topological phase with rotation symmetry. Because this result was derived from the topological crystal construction, and because that Appendix contains many similar conditions on the existence of also weak and second order phases, we believe that this is the best place for this paragraph within our manuscript. We used these conditions in our exhaustive classification of topological phases in Appendix F (previously Appendix E).

The purpose of the paragraph on page 17 relating to Ref. [46] is to provide an argument why screw dislocations host anomalous states parallel to the screw axis. It is important to capture also screw disclinations within our formalism, in particular to argue that our results (7) to (10) also apply to screw disclinations and screw dislocations.

** Third referee: **

11.) Could the authors more clearly explain why Fig. B.4(d) corresponds to a block-off-diagonal coupling with the x-Pauli matrix (cf. text on p.48)?

** Our response: ** As an answer to this question, we have included the following footnote in the respective paragraph in Appendix C.2 (previously Appendix B.2):

"The block-off-diagonal coupling $h \otimes \mu_1$ is required for the following reason. Twofold rotation symmetry exchanges the $(d-1)$-cell decorations within each block. Upon symmetrically moving the decorations away from the $(d-1)$-cells at the center of the unit cell, any diagonal hopping $h \otimes \mu_0$ or $h \otimes \mu_3$ would become non-local because the twofold rotation symmetry related decorations within a single block move in opposite directions. Only a block-off-diagonal coupling $h \otimes \mu_1$ remains local upon moving the decorations of each block symmetrically in opposite directions, as shown in Fig. C.4 (d) [previously Fig. B.4 (d)]."

** Third referee: **

12.) In the discussion of fluxes bound to defects under Eq. (11), the charge “2e” is set in the magnetic flux quantum. Is this because this discussion only applies to superconductors? Or does this discussion also apply to Gedankenexperiments with topological insulators?

** Our response: ** This discussion also applies to the following Gedankenexperiment with insulators: A system that hosts a magnetic flux $n \frac{hc}{2e}$ in a unit cell, with $n$ integer, still satisfies time-reversal symmetry. In this case, any particle transported around the unit cell with flux acquires a geometric phase equal to $n \pi$. For odd $n$, it has been shown that the three dimensional first order strong topological insulator hosts a helical mode at the magnetic $\pi$ flux defect (see for example Ref. [Hong et al., Nano Lett. 2014, 14, 5, 2815–2821 (2014)]). This result also follows from our mathematical arguments of App D.2 as in the resubmitted version of our article (App. C.2 in the original version).

We have included a footnote below Eq. (11) explaining the Gedankenexperiment that leads to the flux quantization in time-reversal symmetric insulators.

Sincerely, Max Geier, Ion Cosma Fulga, and Alexander Lau

---

## Round 3 · Referee Report · Frank Schindler (Referee 2) · 2021-3-1

Report

In their resubmission, the authors have addressed all of my requests and comments in a satisfactory manner. I am therefore in favor of publication of the manuscript in its present form.

---

## Round 3 · Referee Report · Anonymous (Referee 1) · 2021-3-30

Report

I am satisfied by the changes made, which I have considered in detail, and recommend publication.

---

## Round 3 · Author Response

We would like to thank all referees for their positive reports and for thoroughly studying our paper. We are very thankful for the constructive criticism and have modified our manuscript accordingly. We hope that the modifications provide further clarification, improve the readability, and make our results stand out more clearly. We believe that thanks to the referees suggestions we were able to further improve the quality of the manuscript.

We have included our responses to the individual referee reports as replies on the submissions page. For convenience, we provide in a separate email to the editor our manual "difference file" of our manuscript where all our new text insertions are marked in red and relevant deletions are in gray and crossed out.

Sincerely,
Max Geier, Ion Cosma Fulga, Alexander Lau

---

## Round 3 · List of Changes

General: * changed manuscript title slightly (crystalline -> rotation symmetric) * expanded introduction and conclusion to emphasize the central results * improved readability by including section summaries and introductions as well as minor adjustments and cross-references throughout the text * included several footnotes throughout the text providing detailed information * added Refs. 60,61,62,64,65,66,68,73 * new Appendix A about holonomy equivalence classes of disclinations * new subsection Appendix C.4 on the validity of the topological crystal construction for finite size samples * other small changes and clarifications throughout the Appendix * corrected typos throughout the text

Section 1:

  • Added references 60,61,62 and sentence in the introduction in the paragraph on the literature
  • Extended the summary of the manuscript to also contain the main results

Section 2:

  • Changed section title to "Disclinations"
  • added section introduction
  • added more precise definition of the lattice in terms of charge densities
  • added a footnote in Sec. 2.1
  • added Ref. 64 on definition of primitive unit cell
  • clarified caption of Fig. 1
  • added a reference to Appendix B.2 on disclination dipoles
  • in Sec. 2.3, point to new Appendix A for details on holonomy classes
  • added Refs. 35,50,53,65,66 in Sec. 2.3
  • added footnote in Sec. 2.5 to clarify the notation in Eq.(2)
  • in Sec. 2.6: changed text on geometric pi fluxes for clarification
  • added footnote in Sec. 2.6 to support the discussion
  • minor changes in the text and in the subsection titles to improve readability

Section 3:

  • added "Strong" to the section title
  • Changed section introduction
  • clarification regarding strong topological phases in Sec. 3.1 and added footnote
  • clarification regarding two-fold symmetric case in Sec. 3.4 and added footnote
  • added section summary
  • minor modifications to text and titles to improve readability

Section 4:

  • Changed section title to "Disclinations in topological crystals"
  • small changes in subsection titles
  • small changes in the text to improve readability
  • the section introduction points to Appendix C.4 which contains a discussion on the validity of the approach for finite size samples
  • added section summary
  • added a note in the caption of Fig. 7
  • corrected a typo in Eqs. (7) to (10)
  • additional footnote in Sec. 4.4 clarifying the definition of the topological invariants used in Eqs. (7) to (10)
  • additional paragraph at the end of Sec. 4.4 elaborating on the translation holonomy vector T

Section 5:

  • added section outline to the section introduction
  • re-arranged the text into new subsections to improve clarity
  • added footnote in Sec. 5.1 on flux quantization
  • clarified interpretation of Cartan class BDI
  • added footnote in Sec. 5.5 to clarify direct sums

Section 6:

  • added reference to kwant software package
  • minor clarification in the caption of Fig. 10
  • added new Fig. 13 for 2pi/3 disclination and corresponding sentence in the text.

Section 7: * split the Conclusions into "Results" and "Discussion" * added additional paragraphs summarizing the main results of the paper

Appendix

Appendix A: * new appendix discussion the construction of the holonomy equivalence classes of disclinations

Appendix C.2: * added footnote explaining the necessity of the block-offdiagonal coupling

Appendix C.4: * new appendix discussing the validity of the topological crystal approach for finite size samples

---

## Editorial Decision

published